# Targeting dependency on a paralog pair of CBP/p300 against de-repression of KREMEN2 in SMARCB1-deficient cancers

**Mariko Sasaki** [1], **Daiki Kato**[2], **Karin Murakami**[2], **Hiroshi Yoshida** [3], **Shohei Takase** [1], **Tsuguteru Otsubo**[2] & **Hideaki Ogiwara** [1] ✉

SMARCB1, a subunit of the SWI/SNF chromatin remodeling complex, is the causative gene of rhabdoid tumors and epithelioid sarcomas. Here, we identify a paralog pair of CBP and p300 as a synthetic lethal target in SMARCB1-deficient cancers by using a dual siRNA screening method based on the "simultaneous inhibition of a paralog pair" concept. Treatment with CBP/p300 dual inhibitors suppresses growth of cell lines and tumor xenografts derived from SMARCB1-deficient cells but not from SMARCB1-proficient cells. SMARCB1-containing SWI/SNF complexes localize with H3K27me3 and its methyltransferase EZH2 at the promotor region of the *KREMEN2* locus, resulting in transcriptional downregulation of *KREMEN2*. By contrast, SMARCB1 deficiency leads to localization of H3K27ac, and recruitment of its acetyltransferases CBP and p300, at the *KREMEN2* locus, resulting in transcriptional upregulation of *KREMEN2*, which cooperates with the SMARCA1 chromatin remodeling complex. Simultaneous inhibition of CBP/p300 leads to transcriptional downregulation of *KREMEN2*, followed by apoptosis induction via monomerization of KREMEN1 due to a failure to interact with KREMEN2, which suppresses anti-apoptotic signaling pathways. Taken together, our findings indicate that simultaneous inhibitors of CBP/p300 could be promising therapeutic agents for SMARCB1-deficient cancers.

Cancer genomic medicine is a type of cancer therapy that focuses on targeting gene mutations in cancer cells. Future development of multigene panel testing is expected to promote application of cancer genomic medicine[1]. Currently, cancer genomic medicine is applied mainly to cancers with gain-of-function (GOF) mutations in oncogenes[2]. Other mutations include loss-of-function (LOF) mutations in tumor suppressor genes, although genes harboring LOF mutations cannot be a therapeutic target. Synthetic lethality is defined as cell death caused by simultaneous suppression of two genes rather than suppression of a single gene[3]. Synthetic lethal agents that target a synthetic lethal factor in cancers with LOF mutations are available as a cancer therapy.

Mutation of genes encoding components of the SWI/SNF (SWItch/ Sucrose Non-Fermentable) chromatin remodeling complex are detected in ~20% of all cancer patients[4,5]. The SWI/SNF chromatin remodeling complex comprises about 15 subunits, and is classified into three complexes: the BRG1/BRM-associated factor (BAF) complex, the polybromo-associated BAF (PBAF) complex, and the noncanonical BAF (ncBAF) complex[6]. Most SWI/SNF-related genes cause LOF genetic aberrations in cancer cells; therefore, development of therapies based on synthetic lethality is a promising therapeutic strategy. Almost all rhabdoid tumors and epithelioid sarcomas are deficient in SMARCB1 (SWI/SNF-related, matrix-associated actin-dependent regulator of

[1]Division of Cancer Therapeutics, National Cancer Center Research Institute, 5-1-1, Tsukiji, Chuo-ku, Tokyo 104-0045, Japan. [2]Cancer Research Unit, Sumitomo Pharma Co., Ltd, 3-1-98 Kasugade-naka, Konohana-ku, Osaka 554-0022, Japan. [3]Department of Diagnostic Pathology, National Cancer Center Hospital, 5-1-1, Tsukiji, Chuo-ku, Tokyo 104-0045, Japan. ✉e-mail: hogiwara@ncc.go.jp

chromatin, subfamily B, member 1), a subunit of the SWI/SNF chromatin remodeling complex[7,8]. Tazemetostat, an inhibitor of EZH2 (enhancer of zeste homolog 2), is approved as a treatment for SMARCB1-deficient epithelioid sarcomas (a rare type of cancer), even though it shows limited clinical efficacy[9]. The SWI/SNF chromatin remodeling complex regulates cellular functions such as transcription by opening or closing the chromatin structure. Transcription is regulated by promoting or repressing gene expression via various chromatin-regulating factors[3]; therefore, cancers deficient in the SWI/SNF chromatin remodeling complex are vulnerable because the balance between promotion and suppression is disrupted. In general, the SWI/SNF complex promotes gene transcription; however, it can also do the opposite[10]. Therefore, it may go unnoticed that aberration of the repressive function of the SWI/SNF complex generates vulnerabilities that promote transcription.

CBP (CREB-binding protein [*CREBBP*]) and its paralog p300 (E1A binding protein p300 [*EP300*]) act redundantly to acetylate histone H3K27 and promote transcription by opening the chromatin structure and recruiting other transcriptional regulators[11,12]. CBP and p300 recruit components of the RNA Polymerase II machinery and act as adapters for recruitment of other transcriptional cofactors[13]. Two domains of CBP/p300, the catalytic histone acetylation (HAT) domain and the bromo domain (BRD) (which binds chromatin through the acetylated histone) could be therapeutic targets. Therefore, development of small-molecule inhibitors of CBP/p300 is an active area of drug discovery for diverse human diseases, including cancer. The HAT inhibitor A-485 and the BRD inhibitor inobrodib (CCS1477) are potent and promising inhibitors of CBP/p300[14,15]. The CBP and p300 proteins are a paralog pair that share high sequence homology and functional similarity. Basically, existing inhibitors of CBP/p300, including A-485 and inobrodib, selectively inhibit the function of CBP and p300 simultaneously[14,15].

Recent advances in technologies such as next generation sequencing and the CRISPR/Cas9 system have led to identification of many novel synthetic lethal targets through comprehensive genomic screening[16]. In principle, a synthetic lethal target can be a single determining factor for a LOF mutation gene. Current synthetic lethal screening technology can identify a single factor as a synthetic lethal target; however, next generation synthetic lethal screening can be expected to identify a set of two factors as a synthetic lethal target for a gene harboring a LOF mutation. Indeed, simultaneous inhibition of two factors could lead to synthetic lethality in cancer cells harboring a gene with a LOF mutation. Paralog proteins are two very similar proteins that, in general, have a redundant function[17,18]. Because paralog pair proteins have a very similar structure, inhibiting either one can be difficult; rather, inhibiting both simultaneously is more realistic. Indeed, most inhibitors of a protein paralog pair inhibit both proteins. Previous screenings for synthetic lethality searched for a single factor[16]. These screenings could only identify agents that target one of the proteins in a paralog pair[19]. Thus, conventional screening methods may overlook paralog pairs as a synthetic lethal target. Here, we hypothesizes that screening for synthetic lethal paralog pairs may be a promising approach to identifying novel synthetic lethal targets. The "simultaneous inhibition of a paralog pair" method may therefore be an advanced strategy for therapies based on a single agent that can target two proteins. Here, we aim to identify paralog pairs as a synthetic lethal target for SMARCB1-deficient cancers.

## Results

### Simultaneous inhibition of CBP/p300 causes synthetic lethality in SMARCB1-deficient cancer cells

To identify a novel synthetic lethal target in SMARCB1-deficient cancers, we first established an isogenic cell line model by introducing *SMARCB1* cDNA into SMARCB1-deficient JMU-RTK-2 rhabdoid tumor cells (Fig. 1a). We observed that expression of SMARCB1 protein in

JMU-RTK-2 + SMARCB1 cells was comparable with that in SMARCB1-proficient cell lines HEK293T and 786-O (Supplementary Fig. 1a). Therefore, we considered expression of SMARCB1 protein in the JMU-RTK-2 + SMARCB1 cells to be at the wild-type level rather than being overexpressed. Use of the SMARCB1 isogenic cell line model enabled us to perform synthetic lethal screening to identify paralog pairs as a synthetic lethal target for SMARCB1-deficient cells. SMARCB1, a member of the SWI/SNF chromatin remodeling complex, is involved in chromatin regulation. Synthetic lethality is often caused by similar functional partners[3]. In addition, some chromatin regulator proteins have a paralog[20]. In this study, we searched for a promising synthetic lethal target for SMARCB1-deficient cancers among chromatin regulators; this is because SMARCB1 is involved in chromatin regulation and there are cases in which functionally related chromatin regulators show synthetically lethal properties. We selected 30 pairs based on published molecular phylogenetic trees and known protein structures of chromatin regulators such as histone acetyltransferase, histone methyltransferase, histone demethylase, and chromatin remodeling factor[20–24]. Depleting these paralog pairs did not affect growth of HEK293T immortalized normal cell lines, suggesting that simultaneous inhibition of paralog pairs would not be toxic to normal cells (Fig. 1b). Therefore, we next screened paralog pairs of chromatin regulators as a possible synthetic lethal target in SMARCB1-deficient cells. The siRNA screening shown in Fig. 1b identified CREBBP + EP300, KDM3A + KDM3B, KMT2C + KMT2D, BRPF1 + BRPF3, KDM6A + KDM6B, and PRDM8 + PRDM13 as previously unidentified paralog pair candidates that are synthetic lethal to SMARCB1-deficient cells. EZH2 + EZH1 was also identified as an existing synthetic lethal target for SMARCB1-deficient cancers[25,26] during screening (Fig. 1b). In addition, it has been reported that KDM6A and KDM6B are synthetic lethal targets for cancers deficient in SMARCA4, which is another subunit of the SWI/SNF complex[27]. We revalidated these candidates using other SMARCB1+ cells (786-O, H460) and SMARCB1- cells (G402, HS-ES-2R) to narrow them down (Fig. 1c). The data showed that the siRNA pair si*CREBBP*+si*EP300* had little effect on the viability of SMARCB1-proficient cells but was lethal to SMARCB1-deficient cells, i.e., it was synthetically lethal. In addition, we used six SMARCB1-proficient cell lines (SMARCB1+) and six SMARCB1-deficient cell lines (SMARCB1-) (Fig. 1d) to confirm that simultaneous depletion of paralog pair CREBBP + EP300 (Supplementary Fig. 1b–d) decreased the viability of SMARCB1-deficient cell lines, but not that of SMARCB1-proficient cell lines (Fig. 1e). Therefore, we focused on CREBBP + EP300 as a promising synthetic lethal target pair in SMARCB1-deficient cells.

Next, we asked whether dual suppression of CREBBP and EP300, but not suppression of either alone, causes synthetic lethality. Depletion of either CREBBP or EP300 (Supplementary Fig. 1e–g) partially suppressed growth of JMU-RTK-2 -SMARCB1 cells, but not that of JMU-RTK-2 +SMARCB1 cells (Fig. 1f). Moreover, simultaneous depletion of both paralogs (Supplementary Fig. 1e–g) led to significantly greater growth suppression than depletion of either paralog alone (Fig. 1f). In addition, simultaneous depletion of CREBBP and EP300 in SMARCB1-deficient cell lines (Supplementary Fig. 1h–j) led to significantly greater growth suppression than depletion of either paralog alone (Fig. 1g), but single or dual depletion of CREBBP and EP300 (Supplementary Fig. 1h–j) did not affect the growth of SMARCB1-proficient cell lines (Fig. 1g). Thus, simultaneous dual suppression of CREBBP and EP300, but not single suppression, causes synthetic lethality in SMARCB1-deficient cancers. Therefore, in this study, we focused on the paralog pair CREBBP and EP300 as a synthetic lethal target for SMARCB1-deficient cancers.

It is reported that restoring SMARCB1 genes in SMARCB1-deficient cells slows cell proliferation[28]. Indeed, growth of JMU-RTK-2 +SMARCB1 cell lines was slower than that of JMU-RTK-2 -SMARCB1 cell lines (Supplementary Fig. 1k). Therefore, to consider the impact of differences in proliferation of these cell lines on the results of siRNA

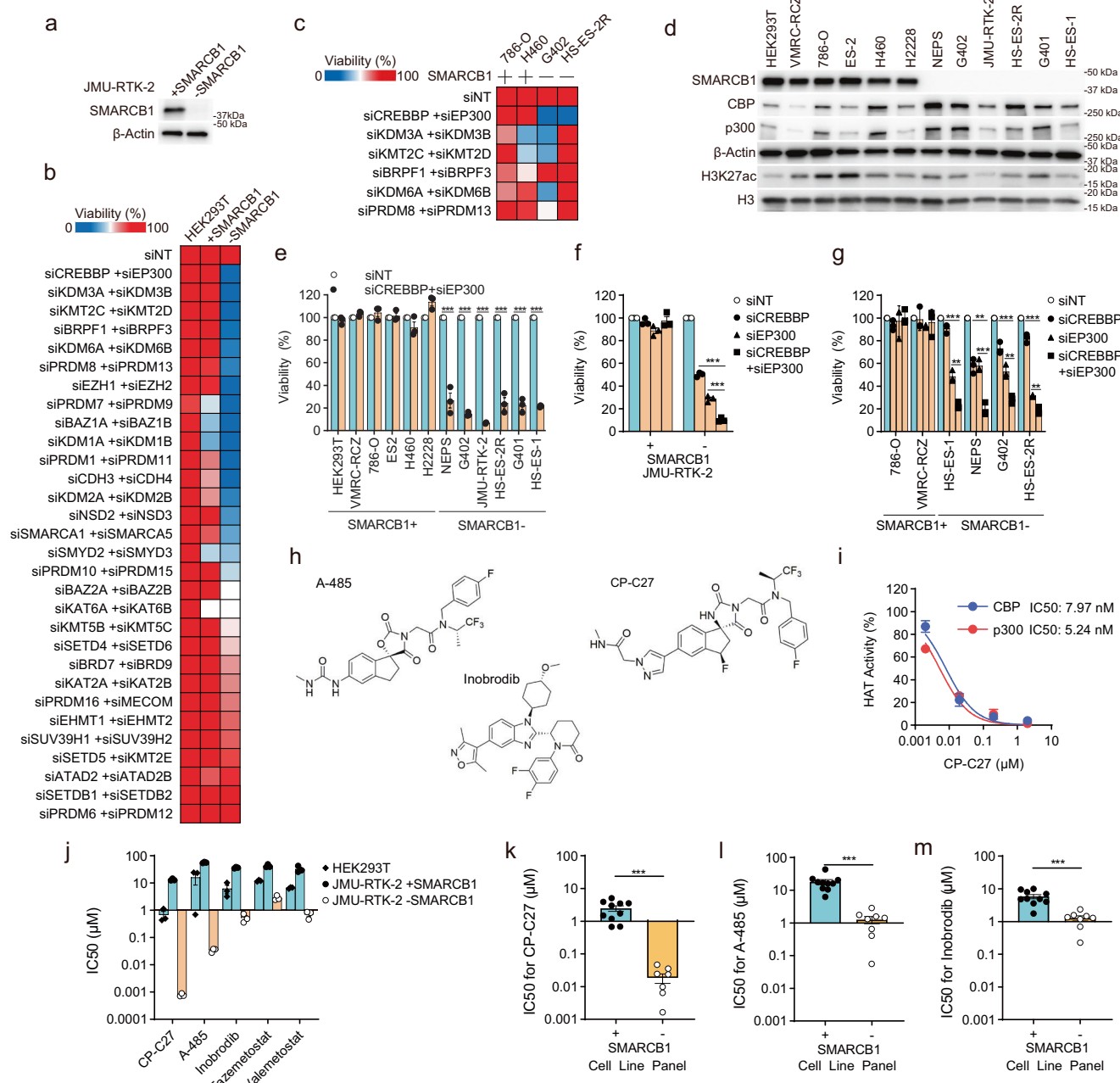

screening, we also examined SMARCB1-proficient HEK293T immortalized cell lines and five SMARCB1-proficient cancer cell lines. We confirmed that the siRNA targeting CREBBP and EP300 that we used for screening had almost no effect on cell proliferation (Fig. 1b, e). In addition, we examined synthetic lethality in six SMARCB1-proficient cell lines and six SMARCB1-deficient cell lines and found little difference in the degree of cell proliferation overall (Supplementary Fig. 1l). This suggests that not only isogenic cells harboring SMARCB1 (i.e., JMU-RTK-2 +SMARCB1 cells), but also the six SMARCB1-proficient cell lines, did not have an effect on CBP/p300 inhibition due to differences in cell proliferation.

To conduct drug susceptibility screening using inhibitors of chromatin regulators, we selected existing inhibitors from among the chromatin regulator paralog pair factors selected by siRNA screening. Drug susceptibility screening using these inhibitors revealed that the CBP/p300 inhibitor A-485 was the most selective for SMARCB1-deficient cancers (Supplementary Fig. 1m). A-485 acts as a dual inhibitor of CBP and p300 by targeting the HAT domain (Fig. 1h)[14].

By contrast, inobrodib (CCS-1477) acts as a dual inhibitor of CBP/p300 by targeting the BRD (Fig. 1h), but does not inhibit at least 30 other proteins with a BRD, including SMARCA4 (SWI/SNF-related, matrix-associated, actin-dependent regulator of chromatin, subfamily a, member 4), SMARCA2 (SWI/SNF-related, matrix-associated, actin-dependent regulator of chromatin, subfamily a, member 2), and PBRM1 (polybromo-1), which harbor the bromodomains within SWI/SNF complex subunits[15]. Inobrodib is now in a phase I clinical trial for hematological cancers and solid tumors (NCT04068597, NCT03568656)[15]. Recently, the potent and orally available CBP and p300 HAT inhibitor CP-C27 (CBP and p300 inhibitor-compound 27) was generated as an optimized A-485 analog (Fig. 1h)[29]. We found that CP-C27 selectively inhibited the HAT activity of both CBP and p300, but had no effect on other HATs (Fig. 1i, Supplementary Fig. 1n). Next, we investigated the selectivity of existing representative CBP/p300 specific inhibitors CP-C27, A-485, and inobrodib in SMARCB1-deficient cells using the SMARCB1 isogenic model. The IC$_{50}$ (50% inhibitory concentration) values derived from JMU-RTK-2

**Fig. 1 | Simultaneous inhibition of CBP/p300 causes synthetic lethality in SMARCB1-deficient cancer cells. a** Immunoblot analysis of SMARCB1 and β-actin expression in JMU-RTK-2 + SMARCB1 and JMU-RTK-2 -SMARCB1 cells. The experiments were repeated twice independently with similar results. **b** Heatmap showing the viability of HEK293T, JMU-RTK-2 + SMARCB1, and JMU-RTK-2 -SMARCB1 cells transfected with siRNAs targeting 30 paralog pairs. Cells were transfected for 48 h with the indicated siRNAs. The cells were then reseeded and transfected repeatedly with the indicated siRNAs for 48 h. The cells were then reseeded and incubated for 7 days. Cell viability is shown as a heatmap. **c** Heatmap showing the viability of SMARCB1-proficient (786-O and H460) and SMARCB1-deficient (G402 and HS-ES-2R) cell lines transfected with the indicated paralog pairs of siRNAs. Cells were transfected for 48 h with the indicated siRNAs. The cells were then reseeded and transfected repeatedly with the indicated siRNAs for 48 h. The cells were then reseeded and incubated for 7 days. Cell viability is shown as a heatmap. **d** Immunoblot analysis of SMARCB1, CBP, p300, β-actin, and histone H3 and H3K27ac expression in SMARCB1-proficient and SMARCB1-deficient cell lines. **e** Viability of SMARCB1-proficient (786-O, VMRC-RCZ, 786-O, ES2, H460, and H2228) and SMARCB1-deficient (NEPS, G402, JMU-RKT-2, HS-ES-2R, G401, and HS-ES-1) cell lines transfected with siRNAs specific for *CREBBP + EP300*, or with NT (non-targeting) siRNA. Cells were transfected for 48 h with the indicated siRNAs. The cells were then reseeded and transfected repeatedly with the indicated siRNAs for 48 h. The cells were then reseeded and incubated for 7 days. Data are presented as the mean ± SEM (standard error of the mean), $n = 3$ independent experiments. **f** Viability of JMU-RTK-2 +SMARCB1 and JMU-RTK-2 -SMARCB1 cells transfected with siRNAs specific for *CREBBP* and/or *EP300*, or with NT siRNA. Cells were transfected for 48 h with the indicated siRNAs. The cells were then reseeded and

transfected repeatedly with the indicated siRNAs for 48 h. The cells were then reseeded and incubated for 7 days. Data are presented as the mean ± SEM, $n = 3$ independent experiments. **g** Viability of SMARCB1-proficient (786-O and VMRC-RCZ) and SMARCB1-deficient (HS-ES-1, NEPS, G402, and HS-ES-2R) cell lines transfected with siRNAs specific for *CREBBP* and/or *EP300*, or with NT siRNA. Cells were transfected for 48 h with the indicated siRNAs. The cells were then reseeded and transfected repeatedly with the indicated siRNAs for 48 h. The cells were then reseeded and incubated for 7 days. Data are presented as the mean ± SEM, $n = 3$ independent experiments. **h** Chemical structures of CBP/p300 dual inhibitors A-485, inobrodib, and CP-C27. **i** Histone acetylation (HAT) activity of CBP and p300 in vitro. The $IC_{50}$ (50% inhibitory concentration) values denoting the inhibitory effects of CP-C27 are shown. Data are presented as the mean ± SD (standard deviation), $n = 2$ independent experiments. **j** $IC_{50}$ values of the CBP/p300 inhibitors CP-C27, A-485, and inobrodib, the EZH2 inhibitor tazemetostat, and the EZH1/EZH2 inhibitor valemetostat in HEK293T, JMU-RTK-2 +SMARCB1, and JMU-RTK-2 -SMARCB1 cells. Cells were treated with inhibitors for 6 days and $IC_{50}$ values were calculated based on cell viability. Data are presented as the mean ± SEM, $n = 3$ independent experiments. **k−m** $IC_{50}$ values for CBP/p300 inhibitors CP-C27 (**k**), A-485 (**l**), and inobrodib (**m**) in SMARCB1-proficient (H460, H1048, H2009, H2228, 786-O, H358, Caki-1, HEK293T, VMRC-RCZ, and ES2) and SMARCB1-deficient (HS-ES-2M, HS-ES-2R, A-204, NEPS, G401, G402, HS-ES-1, and JMU-RTK-2) cells (corresponding to Supplementary Fig. 1o−q). Cells were treated with inhibitors for 6 days and $IC_{50}$ values were calculated based on cell viability. Data are presented as the mean ± SEM, (SMARCB1+; $n = 10$ biological independent cell lines, SMARCB1-; $n = 8$ biological independent cell lines). For all experiments, $p$-values were determined by an unpaired two-tailed Student's $t$-test. $*p < 0.05$, $**p < 0.01$, $***p < 0.001$.

---

-SMARCB1 cells treated with CBP/p300-specific inhibitors CP-C27, A-485, and inobrodib were markedly lower than those from JMU-RTK-2 +SMARCB1 cells and HEK293T non-cancer cells (Fig. 1j). In particular, CP-C27 was more selective for SMARCB1-deficient cancer cells than A-485 or inobrodib (Fig. 1j). Importantly, CP-C27 was more selective than the EZH2 inhibitor tazemetostat (EPZ-6438), which is approved for treatment of SMARCB1-deficient epithelioid sarcomas[30], and it was more selective than the EZH2/EZH1 dual inhibitor valemetostat (DS-3201) (Fig. 1j). In addition to the isogenic SMARCB1-deficient model, these CBP/p300 dual inhibitors, especially CP-C27, selectively sensitized SMARCB1-deficient cell lines in the cancer cell line panel (Fig. 1k, Supplementary Fig. 1o) to a greater extent than A-485 (Fig. 1l, Supplementary Fig. 1p) and inobrodib (Fig. 1m, Supplementary Fig. 1q). These results indicate that CBP/p300 dual inhibitors are promising therapeutic agents for SMARCB1-deficient cancers.

## SMARCB1 deficiency leads to upregulation of the *KREMEN2* gene

CBP and p300 promote transcription of various genes through histone acetylation at transcriptional regions such as the promotor and enhancer regions[31]. By contrast, the SWI/SNF chromatin remodeling complex is involved not only in transcriptional promotion, but also in transcriptional suppression, of various genes through regulation of the chromatin structure at transcriptional regions[10,32,33]. Based on the differences in these transcriptional roles, we hypothesized that SMARCB1 and CBP/p300 act either co-operatively or competitively to regulate expression of certain genes that could be key factors in determining synthetic lethality induced by simultaneous inhibition of CBP/p300 in SMARCB1-deficient cells. To investigate this hypothesis, we preformed gene expression analyses using RNA-seq. First, we identified a set of 471 genes that were concordantly upregulated in SMARCB1-deficient cells (JMU-RTK-2, HS-ES-2R) but not in SMARCB1-proficient (JMU-RTK-2 + SMARCB1, 786-O) cells (Fig. 2a). Next, we identified a set of 50 genes that were concordantly downregulated in SMARCB1-deficient cells (JMU-RTK-2, HS-ES-2R) treated with A-485, but not in SMARCB1-proficient cells (JMU-RTK-2 + SMARCB1, 786-O) treated with A-485 (Fig. 2a). Then, we identified a set of 22 genes that showed overlap between these two gene sets (Fig. 2a). To further narrow down these genes, we identified a set of 54 genes that were concordantly downregulated in two other SMARCB1-deficient cell lines (G402, NEPS)

treated with A-485 (Fig. 2a). Finally, we identified only the *KREMEN2* (Kringle containing transmembrane protein 2) as a gene overlapping between the 22 gene and the 54 gene sets (Fig. 2a). The *KREMEN2* gene, which is coding a single-pass transmembrane protein that plays dual roles in cells (suppression of the Wnt/β catenin pathway and the apoptosis pathway)[34,35], was upregulated specifically in SMARCB1-deficient cells and downregulated specifically in SMARCB1-deficient cells treated with CBP/p300 inhibitors (Fig. 2a), indicating that *KREMEN2* is a candidate gene that determines synthetic lethality. We were unable to identify a gene downregulated specifically in SMARCB1-deficient cells and downregulated specifically in SMARCB1-deficient cells treated with CBP/p300 inhibitors (Supplementary Fig. 2a).

To confirm that the *KREMEN2* gene is a determinant of synthetic lethality, we investigated whether depleting *KREMEN2* affects cell viability. Depletion of *KREMEN2* (Supplementary Fig. 2b) reduced the viability of JMU-RTK-2 -SMARCB1 cells, but not that of JMU-RTK-2 +SMARCB1 cells, in the SMARCB1 isogenic model (Fig. 2b). In addition, depletion of *KREMEN2* (Supplementary Fig. 2c) reduced the viability of SMARCB1-deficient cell lines, but not that of SMARCB1-proficient cell lines, in the cancer cell line panel (Fig. 2c). Moreover, loss of viability upon depletion of *KREMEN2* from SMARCB1-deficient cells was rescued by overexpression of *KREMEN2* cDNA (Fig. 2d, Supplementary Fig. 2d, e). Thus, SMARCB1-deficient cancer cells are dependent on KREMEN2 expression. Expression of *KREMEN2* mRNA in SMARCB1-deficient cells was higher than that in SMARCB1-proficient cells (Fig. 2e, f). To further examine expression levels of the *KREMEN2* gene among cell lines with different genetic abnormalities, we used mutation, copy number and gene expression data from the CCLE (Cancer Cell Line Encyclopedia) database in DepMap (data version 23Q2). Expression of *KREMEN2* mRNA was compared with gene expression data from *SMARCB1/SMARCA4*-proficient, *SMARCB1*-deficient, and SMARCA4-deficient cell lines. The results showed that expression of *KREMEN2* in not only *SMARCB1*-deficient cell lines, but also in *SMARCA4*-deficient cell lines, was significantly higher than that in *SMARCB1/SMARCA4*-proficient cell lines (Supplementary Fig. 2f). Moreover, we used a published data set (GSE11482) to confirm that expression of *KREMEN2* mRNA in kidney rhabdoid tumors, which are deficient in the *SMARCB1* gene, was significantly higher than that in

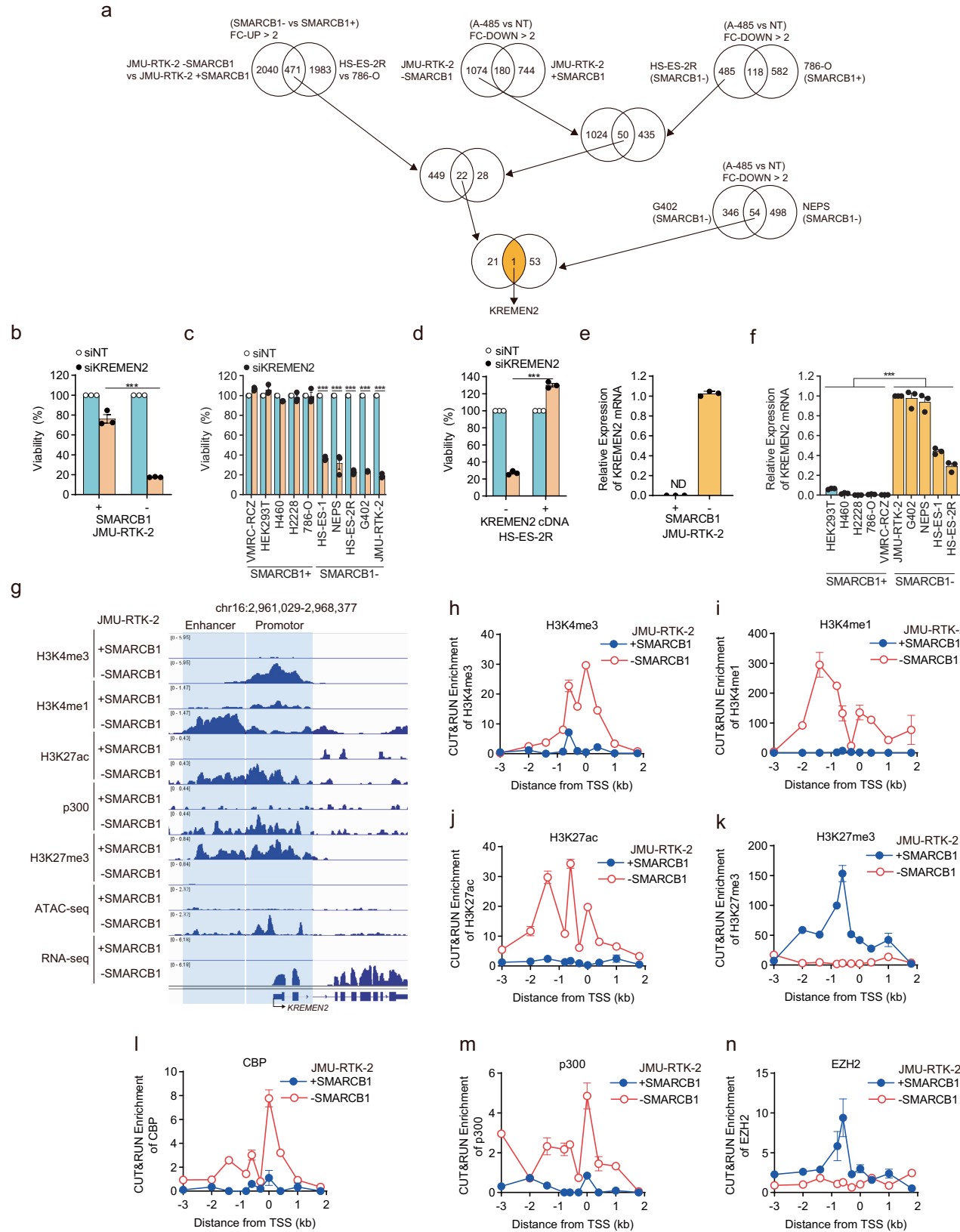

other types of kidney tumor (Supplementary Fig. 2g). To further obtain additional evidence to support involvement of SMARCB1 in transcriptional repression of the *KREMEN2* gene, we examined expression of *KREMEN2* pre-mRNA in the *SMARCB1* isogenic model. Expression of *KREMEN2* pre-mRNA in JMU-RTK-2 -SMARCB1 cells was higher than that in JMU-RTK-2 +SMARCB1 cells (Supplementary Fig. 2h). In

addition, expression levels of *KREMEN2* pre-mRNA in SMARCB1-deficient cell lines was higher than that in SMARCB1-proficient cell lines in the cancer cell line model (Supplementary Fig. 2i). These results indicate that SMARCB1 is required for transcriptional repression of *KREMEN2*, and that SMARCB1-deficiency increases expression of *KREMEN2* mRNA due to de-repression.

**Fig. 2 | SMARCB1 deficiency upregulates expression of the *KREMEN2* gene.**
**a** Schematic flow illustrating the method used to identify KREMEN2 as a determinant for synthetic lethality in SMARCB1-deficient cells treated with a CBP/p300 inhibitor. *KREMEN2* was identified and selected as a gene that is upregulated in SMARCB1-deficient cells, and downregulated in A-485 treated SMARCB1-deficient cells but not in SMARCB1-proficient cells. A set of 471 genes concordantly upregulated in SMARCB1-deficient cells (JMU-RTK-2, HS-ES-2R), but not SMARCB1-proficient cells (JMU-RTK-2 + SMARCB1, 786-O), and a set of 50 genes that were concordantly downregulated in SMARCB1-deficient cells (JMU-RTK-2, HS-ES-2R) treated with A-485, but not in SMARCB1-proficient cells (JMU-RTK-2 + SMARCB1, 786-O), were isolated. Next, a set of 22 genes that overlapped these two gene sets was isolated. In addition, a set of 54 genes that was concordantly downregulated in two other SMARCB1-deficient cells (G402, NEPS) after treatment with A-485 was identified. The *KREMEN2* gene was identified from the 22 and 54 overlapping genes. **b** Viability of JMU-RTK-2 +SMARCB1 and JMU-RTK-2 -SMARCB1 cells transfected with the indicated siRNAs. Cells were transfected for 48 h with the indicated siRNAs. The cells were reseeded and transfected repeatedly with the indicated siRNAs for 48 h. The cells were then reseeded and incubated for 7 days. Data are presented as the mean ± SEM (standard error of the mean), $n = 3$ independent experiments. **c** Viability of SMARCB1-proficient (VMRC-RCZ, HEK293T, H460, H2228, and 786-O) and SMARCB1-deficient (HS-ES-1, NEPS, HS-ES-2R, G402, and JMU-RTK-2) cell lines transfected with the indicated siRNAs. Cells were transfected for 48 h with the indicated siRNAs. The cells were reseeded and transfected repeatedly with the indicated siRNAs for 48 h. The cells were then reseeded and incubated for 7 days. Data are presented as the mean ± SEM, $n = 3$ independent experiments. **d** Viability

of HS-ES-2R mock cells and HS-ES-2R +KREMEN2 cells transfected with the indicated siRNAs. Cells were transfected for 48 h with the indicated siRNAs. The cells were reseeded and transfected repeatedly with the indicated siRNAs for 48 h. The cells were then reseeded and incubated for 7 days. Data are presented as the mean ± SEM, $n = 3$ independent experiments. **e** Expression of *KREMEN2* mRNA in JMU-RTK-2 +SMARCB1 and JMU-RTK-2 -SMARCB1 cells (relative to that in JMU-RTK-2 -SMARCB1 cells). *KREMEN2* mRNA was not detected (ND) in JMU-RTK-2 + SMARCB1. Data are presented as the mean ± SD (standard deviation), $n = 3$ independent experiments. **f** Expression of *KREMEN2* mRNA in SMARCB1-proficient (HEK293T, H460, 786-O, VMRC-RCZ, and H2228) and SMARCB1-deficient (JMU-RTK-2, G402, NEPS, HS-ES-1, and HS-ES-2R) cell lines (relative to that in JMU-RTK-2 cells). Data are presented as the mean ± SD, $n = 3$ independent experiments. **g** Localization signals generated by H3K4me3, H3K4me1, H3K27ac, H3K27me3, CUT&RUN-seq, p300 ChIP-seq, ATAC-seq, and RNA-seq around the KREMEN2 locus in JMU-RTK-2 +SMARCB1 and JMU-RTK-2 -SMARCB1 cells. H3K4me3-localized regions denote the promotor regions of the KREMEN2 locus. H3K4me1-localized regions denote the enhancer regions of the *KREMEN2* locus. **h–n** Enrichment of CUT&RUN signals for H3K4me3 (**h**), H3K4me1 (**i**), H3K27ac (**j**), H3K27me3 (**k**), CBP (**l**), p300 (**m**), and EZH2 (**n**) (calculated relative to the CUT&RUN signal for normal IgG) at the indicated regions distant from the transcription start site (TSS) of the *KREMEN2* gene in JMU-RTK-2 + SMARCB1 and JMU-RTK-2 -SMARCB1 cells. Data are presented as the mean ± SD, $n = 3$ independent experiments. For all experiments, $p$-values were determined by an unpaired two-tailed Student's $t$-test. *$p < 0.05$, **$p < 0.01$, ***$p < 0.001$.

We were unable to detect intracellular KREMEN2 protein using a commercially available anti-KREMEN2 antibody (Supplementary Fig. 2j–l). Transfection of *KREMEN2* siRNA into several SMARCB1-deficient cell lines reduced expression of *KREMEN2* mRNA, as detected by qPCR (Supplementary Fig. 2j). The commercially available anti-KREMEN2 antibody did bind to several protein bands on western blots, but the intensity of these bands was not reduced after transfection of *KREMEN2* siRNA (Supplementary Fig. 2k). However, the KREMEN2 antibody did detect ectopically overexpressed KREMEN2 because the intensity of a band representing overexpressed KREMEN2 was reduced after transfection of KREMEN2 siRNA (Supplementary Fig. 2l). Although expression of *KREMEN2* mRNA in SMARCB1-deficient cells was higher than that in SMARCB1-proficient cells, expression of endogenous KREMEN2 proteins was not detected by commercially available antibodies. However, expression of overexpressed KREMEN2 proteins was detected by introducing cDNA encoding exogenous KREMEN2. Since this overexpressed KREMEN2 protein was knocked down by siKREMEN2, it suggests that the antibody does recognize KREMEN2. Therefore, although the currently available antibodies were not able to detect expression of endogenous KREMEN2 protein, the lethality induced by KREMEN2 knockdown can be avoided by overexpression of KREMEN2, as shown in Fig. 2d. Thus, CBP/p300 inhibitors may attenuate gene expression of *KREMEN2* and induce cell death. However, we have not been able to prove that endogenous KREMEN2 proteins are expressed and functional. In the future, this issue will be resolved by inserting an epitope tag into endogenous KREMEN2 or by creating anti-KREMEN2 antibodies with high specificity and affinity.

To further investigate transcriptional regulation at the *KREMEN2* gene locus, we characterized the chromatin state at the locus by using CUT&RUN (Cleavage Under Targets and. Release Using Nuclease)-seq or ChIP (Chromatin immunoprecipitation)-seq (Fig. 2g). H3K4me3, a marker of the promotor region, localized proximal to the transcription start site (TSS) of the *KREMEN2* gene in JMU-RTK-2 SMARCB1-deficient cells (JMU-RTK-2 -SMARCB1), but not in JMU-RTK-2 SMARCB1-rescued cells (JMU-RTK-2 +SMARCB1) (Fig. 2g). By contrast, H3K4me1, a marker of the enhancer region, localized to sites distal to the TSS in JMU-RTK-2 -SMARCB1 cells, but not in JMU-RTK-2 +SMARCB1 cells (Fig. 2g). H3K27ac, a histone acetylated by CBP and p300, and a marker of transcriptional activated regions, localized with p300, and also localized broadly across both regions bound by H3K4me1 and H3K4me3, in

JMU-RTK-2 -SMARCB1 cells but not in JMU-RTK-2 +SMARCB1 cells (Fig. 2g). Conversely, in JMU-RTK-2 +SMARCB1 cells, H3K27me3, a marker of transcriptionally inactive regions, localized broadly across almost the same regions as H3K27ac and p300 in JMU-RTK-2 -SMARCB1 cells (Fig. 2g). In addition, the ATAC-seq (Assay for Transposase-Accessible Chromatin with sequencing) signal, which is a marker of open chromatin, was detected in the promotor region of the *KREMEN2* locus in JMU-RTK-2 -SMARCB1 cells, but not in JMU-RTK-2 +SMARCB1 cells (Fig. 2g). This was supported by detection of the RNA-seq signal across the exon regions of the *KREMEN2* gene locus in JMU-RTK-2 -SMARCB1 cells, but not in JMU-RTK-2 +SMARCB1 cells (Fig. 2g). This result was also confirmed by published ATAC-seq (GSE124903) and RNA-seq (GSE124903) data derived from another *SMARCB1* isogenic cell line model (SMARCB1-deficient TTC1240 cells), which showed that the ATAC-seq and RNA-seq signals in TTC1240 -SMARCB1 cells were higher than those in TTC1240 +SMARCB1 cells (Supplementary Fig. 2m).

To further confirm localization of histone markers and CBP/p300 at the *KREMEN2* locus, we conducted CUT&RUN-qPCR. H3K4me3, a marker of transcriptional promotion, localized in the vicinity of the TSS in JMU-RTK-2 -SMARCB1 cells, but not in JMU-RTK-2 +SMARCB1-cells (Fig. 2h). H3K4me1, a marker of transcriptional enhancement, localized in a region distant from the TSS in JMU-RTK-2 -SMARCB1 cells, but not in JMU-RTK-2 +SMARCB1 cells (Fig. 2i). H3K27ac, a marker of both the promoter and enhancer regions, localized across the regions that were also localized by H3K4me3 and H3K4me1 in JMU-RTK-2 -SMARCB1 cells, but not in JMU-RTK-2 +SMARCB1 cells (Fig. 2j). In contrast to H3K27ac, H3K27me3 (a marker of transcriptional repression) localized in the region intermediate between the promoter and enhancer regions in JMU-RTK-2 +SMARCB1 cells, but not in JMU-RTK-2 -SMARCB1 cells (Fig. 2k). Moreover, in JMU-RTK-2 -SMARCB1 cells, the acetyltransferases CBP and p300 of H3K27ac also localized across regions that were localized by H3K27ac (Fig. 2l, m). Conversely, in JMU-RTK-2 +SMARCB1 cells, the methyltransferase EZH2, a component of PRC2 methyltransferase complex, of H3K27me3 was also localized across regions that were localized by H3K27me3 (Fig. 2n). This result also confirmed published ChIP-seq data (GSE90634) derived from the TTC1240 *SMARCB1* isogenic cell line model showing that H3K27me3 co-localized with SUZ12, another component of the PRC2 complex (Supplementary Fig. 2n).

Thus, CUT&RUN-qPCR yielded results similar to those of CUT&RUN-seq and ChIP-seq.

CUT&RUN-seq and ChIP-seq tend to generate non-specific background signals. Therefore, we added snapshots of ChIP-seq and CUT&RUN-seq signals at the *ANKRD1* and *CDKN1A* regions as "positive regions" in which SMARCB1 binding to loci outside the *KREMEN2* locus has been reported[28,36]. The localization signals of H3K4me3, H3K4me1, H3K27ac, and p300 ChIP-seq, in addition to ATAC-seq and RNA-seq, were detected in the *ANKRD1* gene region in JMU-RTK-2 +SMARCB1 cells, whereas that of H3K27me3 was detected in JMU-RTK-2 -SMARCB1 cells (Supplementary Fig. 2o). The localization signals of H3K4me3, H3K4me1, H3K27ac, p300 ChIP-seq, ATAC-seq, and RNA-seq were detected in the *CDKN1A* region in both JMU-RTK-2 +SMARCB1 cells and JMU-RTK-2 -SMARCB1 cells, but that of H3K27me3 ChIP-seq was not detected in either JMU-RTK-2 + SMARCB1 or JMU-RTK-2 -SMARCB1 (Supplementary Fig. 2p). These snapshots support the data showing that the histone markers and p300 localize at the *KREMEN2* locus. Taken together, these data suggest that SMARCB1 proficiency underlies the transcriptionally repressed state of the *KREMEN2* gene. Thus, SMARCB1 deficiency leads to upregulation of *KREMEN2* expression.

## SMARCB1-containing SWI/SNF complexes localize at the upstream region of the *KREMEN2* locus to repress transcription

To further investigate transcriptional repression of the *KREMEN2* gene by SMARCB1, we used CUT&RUN-seq to examine localization of SMARCB1 at the *KREMEN2* locus. SMARCB1 co-localized to the pro-motor regions of the *KREMEN2* locus occupied by H3K4me3 and H3K27ac in JMU-RTK-2 +SMARCB1 cells, but not in JMU-RTK-2 -SMARCB1 cells (Fig. 3a). Similar results were obtained by analyzing published ChIP-seq data (GSE90634, GSE124903) derived from another *SMARCB1* isogenic (TTC1240) cell line model (Supplementary Fig. 3a)[28,36]. In addition, SMARCB1 localized to the upstream regions of the *KREMEN2* locus in SMARCB1-proficient 786-O cells, but not in SMARCB1-deficient NEPS cells (Supplementary Fig. 3b). It is common for epigenetic data obtained by NGS technologies such as ChIP-seq and CUT&RUN-seq to exhibit some non-specific background signals. To address this, we incorporated positive control snapshots of the *ANKRD1* and *CDKN1A* loci, both of which recruit the SMARCB1-containing SWI/SNF complex[28,36]. At both the *ANKRD1* and *CDKN1A* gene loci, SMARCB1 co-localized with H3K27ac and H3K4me3 at regions upstream of the TSS in JMU-RTK-2 +SMARCB1 cells, but not JMU-RTK-2 -SMARCB1 cells (Supplementary Fig. 3c, d). Similar results were obtained for another *SMARCB1* isogenic (TTC1240) cell line model (Supplementary Fig. 3e, f)[28,36,37], and for a pair of SMARCB1-proficient 786-O cell lines and SMARCB1-deficient NEPS cell lines (Supplementary Fig. 3g, h). These snapshots support the data showing that SMARCB1 localizes at the *KREMEN2* locus. To further investigate transcriptional regulation of the *KREMEN2* gene region of the SWI/SNF complex containing SMARCB1, we examined localization of SWI/SNF factors using CUT&RUN-qPCR. We confirmed that SMARCB1 binds to the region between the promoter region and the enhancer region in JMU-RTK-2 +SMARCB1 cells (Fig. 3b), as did H3K27me3 (Fig. 2k) and EZH2 (Fig. 2n); however, localization was absent from JMU-RTK-2 -SMARCB1 cells, indicating that SMARCB1 localizes directly at the promotor region of the *KREMEN2* locus and plays a role in transcriptional repression of the *KREMEN2* gene. Therefore, upregulation of *KREMEN2* gene expression due to a deficiency of SMARCB1 may underlie the KREMEN2-dependency of SMARCB1-deficient cells.

The SWI/SNF chromatin remodeling complex comprises BAF, PBAF, and ncBAF complexes[6]. To investigate transcriptional regulation of the *KREMEN2* gene by the SWI/SNF complex containing SMARCB1, we examined localization of SWI/SNF factors using CUT&RUN-qPCR. The ATPase factor SMARCA4 of the SWI/SNF complex localized at regions upstream of the *KREMEN2* locus in JMU-RTK-2 +SMARCB1 cells

(Fig. 3c). However, SMARCA4 was still present in JMU-RTK-2 -SMARCB1 cells, and was newly localized to the TSS (Fig. 3c). In addition, the constituents of SWI/SNF subtype complexes cBAF (ARID1A) and PBAF (PBRM1), both of which include SMARCB1, localized at regions upstream of the *KREMEN2* locus in JMU-RTK-2 +SMARCB1 cells (Fig. 3d, e). However, in JMU-RTK-2 -SMARCB1 cells, ARID1A (cBAF) and PBRM1 (PBAF) shifted to sites proximal to the TSS (Fig. 3d, e). These results also confirmed the similar localization patterns of SMARCB1, SMARCA4, DPF2 (cBAF), and ARID2 (PBAF) in the published ChIP-seq data (GSE90634, GSE124903) derived from the TTC1240 *SMARCB1* isogenic cell line model (Supplementary Fig. 3i)[28,36]. In addition, SS18, a constituent of cBAF and ncBAF, also localized at the region upstream of the *KREMEN2* locus in JMU-RTK-2 +SMARCB1 cells (Fig. 3f). In JMU-RTK-2 -SMARCB1 cells, SS18 shifted to sites proximal to the TSS (Fig. 3f), as in ARID1A (cBAF) (Fig. 3d). In SMARCB1-deficient cell lines, the ncBAF complex is substantially more localized to promoter-proximal sites[28,37,38]. Therefore, we examined localization of the GLTSCR1, a constituent of ncBAF. In JMU-RTK-2 +SMARCB1 cells, GLTSCR1 did not localize to the *KREMEN2* locus, but it did in JMU-RTK-2 -SMARCB1 cells (Fig. 3g), in which it was newly localized at sites proximal to the TSS. Thus, as observed for the SWI/SNF complex containing SMARCB1, cBAF and PBAF (as well as the transcriptional repressor EZH2) localized at regions upstream of the *KREMEN2* locus, and are thus considered to repress transcription of *KREMEN2*.

It is reported that even in cells with SMARCB1 deficiency, the cBAF and PBAF complexes have no effect on the formation of these complexes[6]. Our data suggest that residual cBAF and PBAF deficient in SMARCB1 migrate to the TSS of the *KREMEN2* gene (Fig. 3c–f). In addition, we thought that ncBAF would also be newly recruited to the *KREMEN2* gene regions (Fig. 3g). Therefore, we examined whether residual cBAF and PBAF deficient in SMARCB1, or newly recruited ncBAF, promote transcription of *KREMEN2*. Unexpectedly, suppression of cBAF (ARID1A), PBAF (ARID2), and ncBAF (BRD9) did not attenuate expression of *KREMEN2* (Fig. 3h). Simultaneous suppression of SMARCA4 and SMARCA2, both of which are essential for SWI/SNF function, did not reduce expression of *KREMEN2* (Fig. 3h). Therefore, we assumed that none of the SMARCB1-deficient residual SWI/SNF complexes were involved in promoting expression of *KREMEN2*. Apart from the SWI/SNF complex, other chromatin remodeling complexes such as ISWI, CHD, and INO80 family complexes are classified as sub-complexes[24]. Therefore, we investigated whether chromatin remodeling complexes other than SWI/SNF complexes are involved in promotion of *KREMEN2* transcription. Suppression of SMARCA1, but not that of other complexes, attenuated expression of *KREMEN2* (Fig. 3i). In addition, we confirmed that depletion of SMARCA1 from SMARCB1-deficient cell lines (Supplementary Fig. 3j, k) reduced expression of *KREMEN2* mRNA (Fig. 3i). Therefore, we considered that the SMARCA1 complex was involved in promotion of *KREMEN2* transcription in SMARCB1-deficient cells. SMARCA1 is an ATPase and a subunit of the ISWI family complex, which is involved in transcription when recruited to target gene loci[24]. In fact, SMARCA1 was localized widely in the region upstream of the *KREMEN2* locus due to a lack of SMARCB1 (Fig. 3k). In addition, treatment of JMU-RTK-2 -SMARCB1-cells with the CBP/p300 inhibitor CP-C27 attenuated localization of H3K27ac across the upstream regions of the *KREMEN2* locus (Fig. 3l), as well as localization of SMARCA1 around the TSS site (Fig. 3m). Therefore, in SMARCB1-deficient cells, the SMARCA1 complex may function as a chromatin remodeling complex involved in promoting transcription of *KREMEN2*.

## Simultaneous inhibition of CBP/p300 in SMARCB1-deficient cells induces synthetic lethality by downregulating *KREMEN2*

Next, we investigated whether *KREMEN2* expression upregulated due to de-repression caused by SMARCB1-deficiency is dependent on CBP/p300. Treatment of SMARCB1-deficient cells with CBP/p300 dual

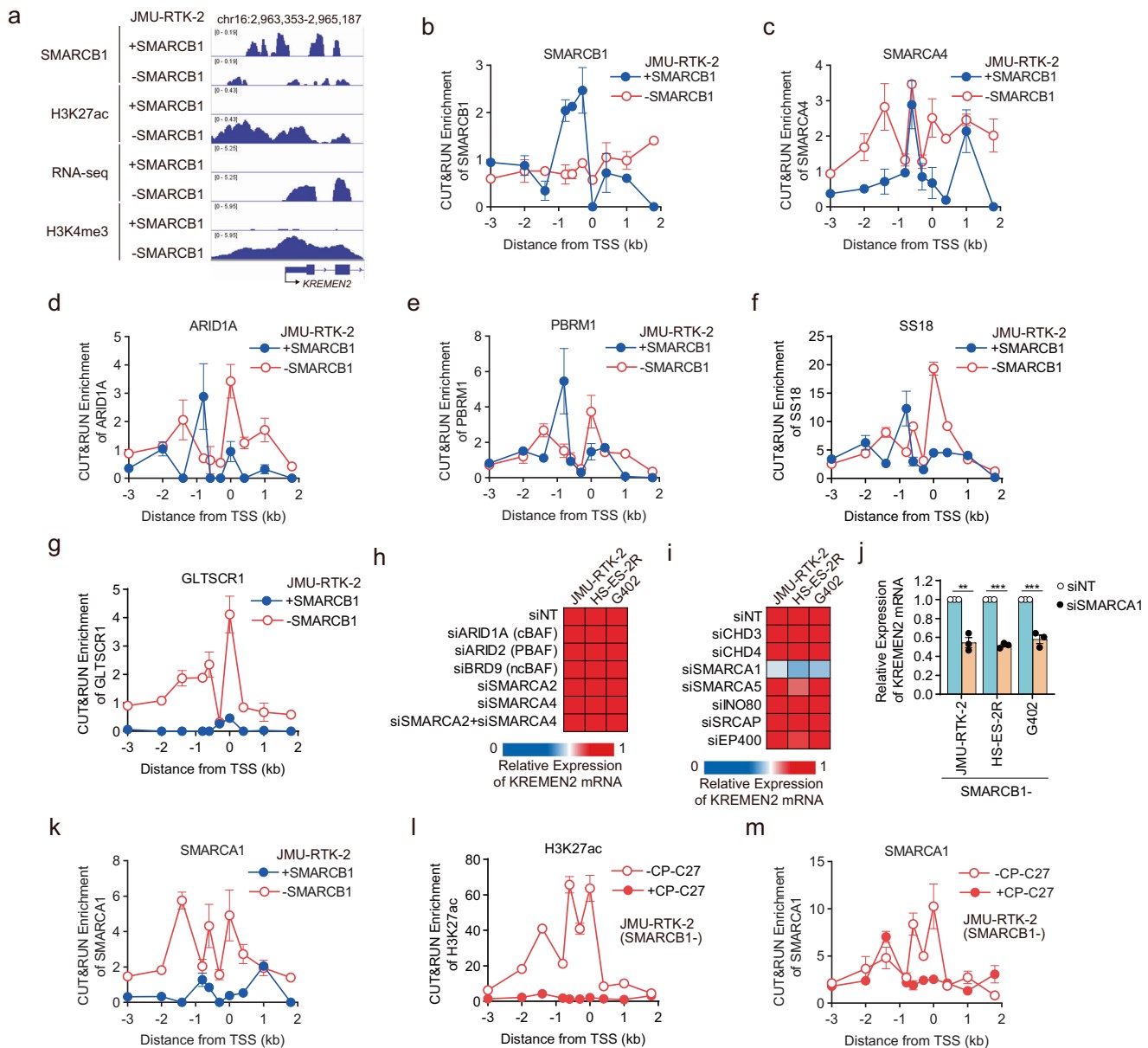

**Fig. 3 | SMARCB1-containing SWI/SNF complexes localize at the region upstream of the *KREMEN2* locus to repress transcription. a** Localization of signals generated by SMARCB1, H3K27ac, and H3K4me3 CUT&RUN-seq, and RNA-seq around the *KREMEN2* locus in JMU-RTK-2 + SMARCB1 and JMU-RTK-2 -SMARCB1 cells. **b**–**g** Enrichment of CUT&RUN signals for SMARCB1 (**b**), SMARCA4 (**c**), ARID1A (**d**), PBRM1 (**e**), SS18 (**f**), and GLTSCR1 (**g**) (relative to that of normal IgG) at the indicated regions distant from the transcription start site (TSS) of the *KRE-MEN2* locus in JMU-RTK-2 +SMARCB1 and JMU-RTK-2 -SMARCB1 cells. Data are presented as the mean ± SD (standard deviation), *n* = 3 independent experiments. **h** Heatmap of *KREMEN2* mRNA expression in SMARCB1-deficient cells (JMU-RTK-2, HS-ES-2R, and G402) transfected for 48 h with the indicated siRNAs. **i** Heatmap of *KREMEN2* mRNA expression in SMARCB1- (JMU-RTK-2, HS-ES-2R, and G402) cells

transfected for 48 h with the indicated siRNAs. **j** Expression of *KREMEN2* mRNA (relative to that in siNT-transfected cells) in SMARCB1-deficient cell lines (JMU-RTK-2, HS-ES-2R, and G402) transfected for 48 h with the indicated siRNAs. Data are presented as the mean ± SD, *n* = 3 independent experiments. **k** Enrichment of CUT&RUN signals for the SMARCA1 (relative to that of normal IgG signal) at the indicated regions distant from the TSS of the *KREMEN2* locus in JMU-RTK-2 +SMARCB1 and JMU-RTK-2 -SMARCB1 cells. Data are presented as the mean ± SD, *n* = 3 independent experiments. **l, m** Enrichment of CUT&RUN signals for the H3K27ac (**l**) and SMARCA1 (**m**) signals (relative to that of normal IgG signal) at the indicated regions distant from the TSS of the *KREMEN2* locus in SMARCB1-deficient JMU-RTK-2 cells treated without or with 2 µM CP-C27 for 24 h. Data are presented as the mean ± SD, *n* = 3 independent experiments.

inhibitors A-485 and CP-C27 decreased acetylation of H3K27ac (Supplementary Fig. 4a, b), and led to downregulation of not only *KREMEN2* mRNA (matured) (Fig. 4a, b) but also *KREMEN2* pre-mRNA (Supplementary Fig. 4c, d). In addition, depletion of both *CREBBP* and *EP3OO* led to downregulation of *KREMEN2* mRNA expression in SMARCB1-deficient cells (Fig. 4c). Moreover, depletion of both *CREBBP* and *EP3OO* downregulated *KREMEN2* gene expression to a significantly greater extent than depletion of either *CREBBP* or *EP3OO* (Fig. 4d).

These results indicate that CBP and p300 are redundantly involved in transcriptional upregulation of the *KREMEN2* gene in SMARCB1-deficient cells.

Next, we characterized the effects of chromatin structure and localization of transcription factors by inhibiting CBP and p300 in SMARCB1-deficient JMU-RTK-2 cells. Treatment with CBP/p300 dual inhibitors A-485 and CP-C27 reduced the ATAC-seq signal at the promotor region, as well as the RNA-seq signal across the exon region, of

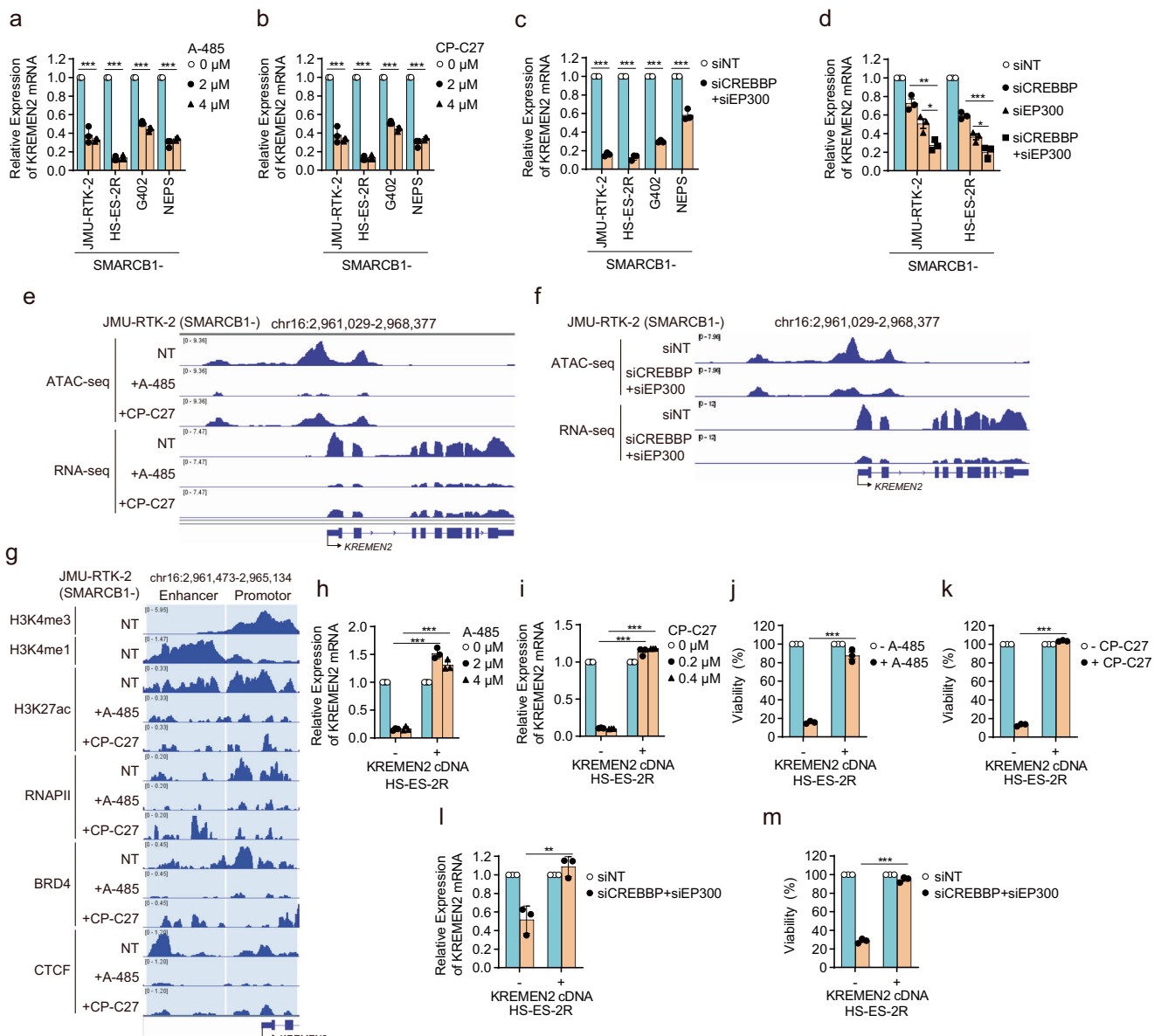

**Fig. 4 | Simultaneous inhibition of CBP/p300 in SMARCB1-deficient cells induces synthetic lethality by downregulating *KREMEN2*. a, b** Expression of *KREMEN2* mRNA (relative to that in siNT-transfected cells) in SMARCB1-deficient cell lines (JMU-RTK-2, HS-ES-2R, G402, and NEPS) treated with the indicated concentrations of A-485 (**a**) and CP-C27 (**b**) for 24 h. Data are presented as the mean ± SD (standard deviation), *n* = 3 independent experiments. **c** Expression of *KREMEN2* mRNA (relative to that in siNT-transfected cells) in SMARCB1-deficient cell lines (JMU-RTK-2, HS-ES-2R, G402, and NEPS) transfected for 48 h with the indicated siRNAs. Data are presented as the mean ± SD, *n* = 3 independent experiments. **d** Expression of *KREMEN2* mRNA (relative to that in siNT-transfected cells) in SMARCB1-deficient cell lines (JMU-RTK-2 and HS-ES-2R) transfected for 48 h with the indicated siRNAs. Data are presented as the mean ± SEM (standard error of the mean), *n* = 3 independent experiments. **e** Localization signals generated by ATAC-seq and RNA-seq around the *KREMEN2* locus in SMARCB1-deficient JMU-RTK-2 cells treated without or with 2 µM A-485 or 0.2 µM CP-C27 for 24 h. Non-treated = NT. **f** Localization signals generated by ATAC-seq, and RNA-seq around the *KREMEN2* locus in SMARCB1-deficient JMU-RTK-2 cell line transfected for 48 h with the indicated siRNAs. **g** Localization signals generated by H3K4me, H3K4me1, H3K27ac, RNAPII, BRD4, and CTCF CUT&RUN-seq at the regions upstream of the *KREMEN2*

locus in SMARCB1-deficient JMU-RTK-2 cells treated without or with 2 µM A-485 or 0.2 µM CP-C27 for 24 h. H3K4me3-localized regions corresponds to the promotor regions of the *KREMEN2* locus. H3K4me1-localized regions correspond to the enhancer regions of the *KREMEN2* locus. **h, i** Expression of *KREMEN2* mRNA (relative to that in NT (non-treated) cells) in HS-ES-2R mock and HS-ES-2R +KREMEN2 cells treated without or with 2 or 4 µM A-485 (**h**) or 0.2 or 0.4 µM CP-C27 (**i**) for 24 h. Data are presented as the mean ± SD, *n* = 3 independent experiments. **j, k** Viability of HS-ES-2R mock and HS-ES-2R + KREMEN2 cells treated with 3 µM A-485 (**j**) or 0.04 µM CP-C27 (**k**) for 6 days. Data are presented as the mean ± SEM, *n* = 3 independent experiments. **l** Expression of *KREMEN2* mRNA (relative to that in siNT-transfected cells) in HS-ES-2R mock and HS-ES-2R + KREMEN2 cells transfected with the indicated siRNAs for 48 h. Data are presented as the mean ± SD, *n* = 3 independent experiments. **m** Viability of HS-ES-2R mock and HS-ES-2R +KREMEN2 cells transfected with the indicated siRNAs. Cells were transfected for 48 h with the indicated siRNAs. The cells were then reseeded and transfected repeatedly with the indicated siRNAs for 48 h. The cells were then reseeded and incubated for 7 days. Data are presented as the mean ± SEM, *n* = 3 independent experiments. For all experiments, *p*-values were determined by an unpaired two-tailed Student's *t*-test. *$p < 0.05$, **$p < 0.01$, ***$p < 0.001$.

the *KREMEN2* locus (Fig. 4e). Similarly, simultaneous depletion of CBP and p300 reduced the ATAC-seq signal and the RNA-seq signal at the *KREMEN2* locus (Fig. 4f). These results suggest that simultaneous inhibition of CBP/p300 leads to chromatin compaction at the promotor region, followed by transcriptional repression. This was supported by the finding that H3K27ac signals across the upstream regions were attenuated by treatment with the CBP/p300 dual inhibitors (Fig. 4g). In addition, localization of transcriptional RNA polymerase II (RNAPII), which colocalizes with H3K4me3, at the promotor regions decreased upon treatment with the CBP/p300 dual inhibitors (Fig. 4g). Localization of BRD4 (bromodomain-containing protein 4), a member of the bromo and extra-terminal (BET) family, to the *KREMEN2* locus was investigated because BRD4 is a transcription factor that binds to acetylated histones and may be involved in transcriptional regulation of the localized region of H3K27ac, which is targeted by CBP/p300[39]. CTCF (CCCTC-binding factor) is a transcription factor that localizes upstream of its target gene loci[40]. In addition, CTCF colocalizes with the residual SWI/SNF complex in the absence of SMARCB1[37], suggesting that it may be involved in transcription of *KREMEN2*, expression of which is promoted by SMARCB1 deficiency. Therefore, we examined whether BRD4 and CTCF localize to the *KREMEN2* locus in SMARCB1-deficient cell lines, and found that BRD4 and CTCF did localize to the *KREMEN2* regions (Fig. 4g). However, localization of BRD4 and CTCF was attenuated by inhibition of CBP/p300 (Fig. 4g). To clarify that CBP and p300 are required for transcription of *KREMEN2*, we next examined the effect of depleting either BRD4 or CTCF on expression of *KREMEN2*. Unexpectedly, depletion of BRD4 or CTCF alone (Supplementary Fig. 4e, f) had almost no effect on expression of *KREMEN2* (Supplementary Fig. 4g, h). These results indicate that histone acetylation by CBP and p300 acts as a foundation for the transcriptional machinery at the *KREMEN2* locus, and is required for recruitment of a set of transcription factors, including BRD4 and CTCF; this then facilitates transcription of the *KREMEN2* gene via RNA polymerase II. In SMARCB1-deficient cells treated with the CBP/p300 inhibitor, histone acetylation reduced by CBP/p300-mediated inhibition leads to failure to recruit transcriptional factors to promotor and enhancer regions at the *KREMEN2* locus, followed by attenuation of transcription of the *KREMEN2* gene (Supplementary Fig. 4i).

To further investigate whether KREMEN2 is a determining factor with respect to inhibition of CBP/p300 in SMARCB1-deficient cells, we rescued KREMEN2 expression after treatment with CBP/p300 dual inhibitors. Treatment with CBP/p300 dual inhibitors decreased expression of *KREMEN2* (Fig. 4h, i, Supplementary Fig. 4j), followed by a reduction in cell viability (Fig. 4j, k, Supplementary Fig. 4k); however, cell viability was rescued by overexpression of KREMEN2. Correspondingly, simultaneous depletion of CBP and p300 (Supplementary Fig. 4l, m) decreased *KREMEN2* expression (Fig. 4l), followed by a reduction in cell viability (Fig. 4m); however, cell viability was again rescued by overexpression of KREMEN2. Similar results were obtained for five cloned SMARCB1-deficient HS-ES-2R cells stably transfected with a KREMEN2 overexpression vector (Supplementary Fig. 4n−q). Taken together, these data suggest that synthetic lethality caused by simultaneous inhibition of CBP/p300 in SMARCB1-deficient cells is dependent on KREMEN2 expression. Therefore, KREMEN2 is a determining factor involved in synthetic lethality.

### Downregulation of *KREMEN2* in SMARCB1-deficient cells through CBP/p300-mediated inhibition induces apoptosis via monomerization of KREMEN1

To date, only two functions of KREMEN2 have been reported in the literature, one relating to the Wnt/β-catenin pathway[34,41] and the other to the apoptotic pathway[35]. Therefore, we next examined whether synthetic lethality induced by downregulation of *KRENEN2* via CBP/p300 inhibition is associated with these two pathways. KREMEN2 plays

a role in a pathway that suppresses Wnt/β-catenin, leading to degradation of β-catenin protein[34,41]. Therefore, we hypothesized that downregulating KREMEN2 via dual inhibition of CBP/p300 will increase expression of β-catenin. As shown in Fig. 2f, expression of *KREMEN2* in SMARCB1-deficient cell lines was higher than that in SMARCB1-proficient cell lines. Conversely, expression of β-catenin in SMARCB1-deficient cell lines tended to be lower than that in SMARCB1-proficient cell lines (Supplementary Fig. 5a). However, β-catenin expression in JMU-RTK-2 -SMARCB1 cells was the same as that in JMU-RTK-2 +SMARCB1 cells (Supplementary Fig. 5b), even though expression of *KREMEN2* in JMU-RTK-2 -SMARCB1 cells was higher than that in JMU-RTK-2 +SMARCB1 cells (Fig. 2e). In addition, treatment with CBP/p300 dual inhibitors did not increase expression of β-catenin (Supplementary Fig. 5c), even though the inhibitors downregulated expression of *KREMEN2* (Fig. 4a, b). Taken together, downregulation of KREMEN2 in SMARCB1-deficient cells treated with CBP/p300 dual inhibitors may not involve de-suppression of the Wnt/β-catenin pathway.

KREMEN2 also plays a role in suppressing apoptosis[35]; thus downregulation of *KREMEN2* via dual inhibition of CBP/p300 in SMARCB1-deficient cells may induce apoptosis. To examine this possibility, we first investigated whether suppressing CBP/p300 in SMARCB1-deficient cells triggers apoptosis. Treatment with CBP/p300 dual inhibitors induced apoptosis in SMARCB1-deficient cells but not in SMARCB1-proficient cells (Fig. 5a, b), as manifested by an increase in the number of cells that were positive for Annexin V staining (a marker of apoptosis). In addition, depleting CBP/p300 induced apoptosis specifically in SMARCB1-deficient cells (Supplementary Fig. 5d). To further investigate the relationship between CBP/p300 inhibition and apoptosis, we performed gene expression analysis using RNA-seq to isolate apoptotic markers involved in specific alterations in expression of gene sets in SMARCB1-deficient cells treated with CBP/p300 dual inhibitors (Fig. 5c). We isolated 3,135 genes whose expression was upregulated or downregulated in JMU-RTK-2 -SMARCB1 cells but not in JMU-RTK-2 +SMARCB1 cells treated with CBP/p300 dual inhibitors (Fig. 5c). In addition, we isolated 1,163 genes whose expression was upregulated or downregulated specifically in SMARCB1-deficient HS-ES-2R cells but not in SMARCB1-proficient 786-O cells treated with CBP/p300 dual inhibitors (Fig. 5c). Then, we identified a set of 332 genes that showed overlap between these two gene sets (Fig. 5c). WikiPathway analysis identified 112 molecular pathways that were significantly associated with the 332 genes (Fig. 5c). To identify apoptotic markers responding to treatment with CBP/p300 dual inhibitors, we focused on two apoptosis-related pathways: Hs-Apoptosis-WP254-106302 and Hs-Apoptosis-Modulation-and-Signaling-WP1772-107525 (Fig. 5c). We then identified pro-apoptotic marker genes *CASP6* (Caspase-6) and *CASP9* (Caspase-9) as being upregulated specifically in SMARCB1-deficient cells treated with CBP/p300 dual inhibitors (Fig. 5d). Quantitative PCR was used to validate upregulation of the *CASP6* and *CASP9* genes in SMARCB1-deficient cells, but not in SMARCB1-proficient cells, treated with CBP/p300 dual inhibitors (Fig. 5e, Supplementary Fig. 5e−g). To confirm involvement of KREMEN2 in suppression of apoptosis, we investigated whether expression of pro-apoptotic marker genes is altered by knockdown of *KREMEN2*. The *CASP6* gene was upregulated by knockdown of *KREMEN2* in SMARCB1-deficient cells, but not in SMARCB1-proficient cells (Supplementary Fig. 5h), indicating that suppression of KREMEN2 in SMARCB1-deficient cells induces apoptosis. Taken together, these results indicate that inhibition of CBP/p300 in SMARCB1-deficient cells induces apoptosis by downregulating *KREMEN2*.

KREMEN2 and KREMEN1 (Kringle containing transmembrane protein 1) are single-pass transmembrane proteins that interact with each other[35]. KREMEN1 is involved in apoptosis induction[42], whereas KREMEN2 suppresses KREMEN1-mediated apoptosis[35]. Therefore, we hypothesized that downregulating *KREMEN2* would increase

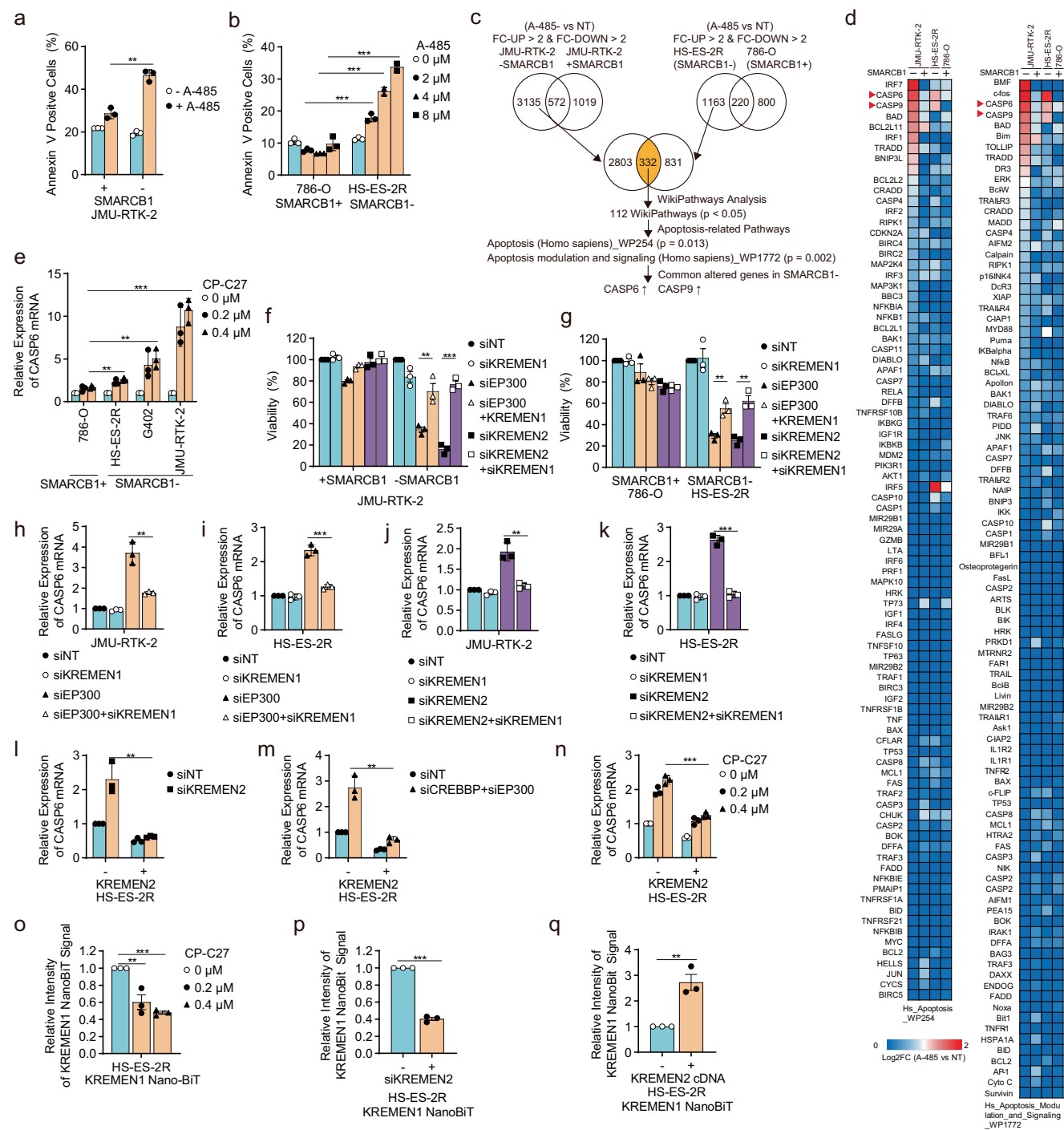

KREMEN1-mediated apoptosis, and then additional depletion of *KREMEN1* would inhibit apoptosis. First, we asked whether synthetic lethality in SMARCB1-deficient cells caused by suppression of CBP/p300 could be prevented by depletion of *KREMEN1*. Depleting p300 (Supplementary Fig. 5j) decreased the viability of SMARCB1-deficient cells (JMU-RTK-2 -SMARCB1, HS-ES-2R, and G402), but not that of SMARCB1-proficient cells (JMU-RTK-2 +SMARCB1, 786-O, and H460) (Fig. 5f, g, Supplementary Fig. 5i); however, additional depletion of *KREMEN1* (Supplementary Fig. 5k) rescued the viability of SMARCB1-deficient cells (Fig. 5f, g, Supplementary Fig. 5i), indicating that the decrease in cell viability induced by depletion of p300 is mediated by KREMEN1. Next, we investigated whether synthetic lethality caused by KREMEN2 depletion in SMARCB1-deficient cells was prevented by additional depletion of *KREMEN1*. Depletion of *KREMEN2*

(Supplementary Fig. 5l) decreased the viability of SMARCB1-deficient cells, but not that of SMARCB1-proficient cells (Fig. 5f, g, Supplementary Fig. 5i); however, additional depletion of *KREMEN1* (Supplementary Fig. 5k) significantly rescued the viability of SMARCB1-deficient cells (Fig. 5f, g, Supplementary Fig. 5i), indicating that the decrease in viability induced by depletion of *KREMEN2* is mediated by KREMEN1. Correspondingly, upregulating *CASP6* by depleting *EP300* (Fig. 5h, i) or *KREMEN2* (Fig. 5j, k) from SMARCB1-deficient cells was prevented by additional depletion of *KREMEN1*. In addition, the sensitivity of SMARCB1-deficient cells to CBP/p300 inhibitors was attenuated by depletion of *KREMEN1* (Supplementary Fig. 5m–p). Correspondingly, upregulation of *CASP6* in SMARCB1-deficient cells by a CBP/p300 inhibitor was prevented by depletion of *KREMEN1* (Supplementary Fig. 5q). These results indicate that downregulating *KREMEN2* by

**Fig. 5 | Downregulation of *KREMEN2* in SMARCB1-deficient cells through CBP/p300 inhibition induces apoptosis via monomerization of KREMEN1.**
**a** Percentage of Annexin V-positive cells within the JMU-RTK-2 +SMARCB1 and JMU-RTK-2 -SMARCB1 cell populations treated with 2 μM A-485 for 96 h. Data are presented as the mean ± SD (standard error), *n* = 3 independent experiments.
**b** Percentage of Annexin V-positive cells within the SMARCB1-proficient (786-O) and SMARCB1-deficient (HS-ES-2R) cell populations treated with the indicated concentrations of A-485 for 6 days. Data are presented as the mean ± SD, *n* = 3 independent experiments. **c** Schematic flow diagram showing identification of apoptotic markers induced by simultaneous inhibition of CBP/p300 specifically in SMARCB1-deficient cells. A set of 3,135 genes upregulated or downregulated by > 2-fold in JMU-RTK-2 -SMARCB1, but not in JMU-RTK-2 +SMARCB1 cells, was identified. In addition, a set of 1,163 genes upregulated or downregulated > 2-fold in SMARCB1-deficient cells (HS-ES-2R) but not in SMARCB1-proficient cells (786-O), was identified. Then, a set of 332 genes that overlapped between these gene sets was isolated. Wikipathway analysis identified 112 molecular pathways that were significantly associated with the 332 genes. To identify apoptotic markers induced by simultaneous inhibition of CBP/p300 specifically in SMARCB1-deficient cells, we focused on two apoptosis pathways. Among the genes related to these apoptosis pathways, *CASP6* and *CASP9* were identified as genes concordantly upregulated specifically in SMARCB1-deficient cells (see also Fig. 5d). **d** Heatmap showing changes (relative to non-treatment) in mRNA levels in apoptosis pathways (Hs-Apoptosis-WP254-106302 and Hs-Apoptosis-Modulation-and-Signaling-WP1772-107525) induced by treatment with 2 μM A-485 for 24 h. The pro-apoptotic marker genes *CASP6* and *CASP9* were identified as genes upregulated specifically in SMARCB1-deficient cells treated with CBP/p300 inhibitor. *CASP6* and *CASP9* are denoted by red arrows.
**e** Expression of *CASP6* mRNA (relative to that in NT (non-treated) cells) in SMARCB1-proficient (786-O) and SMARCB1-deficient (JMU-RTK-2, HS-ES-2R, and G402) cell lines treated with 0.2 or 0.4 μM CP-C27 for 24 h. Data are presented as the mean ± SD, *n* = 3 independent experiments. **f** Viability of JMU-RTK-2 +SMARCB1 and JMU-RTK-2 -SMARCB1 cells transfected with or without siRNAs targeting *EP300*, *KREMEN2*, and/or *KREMEN1*. Cells were transfected with indicated siRNAs for 48 h. The cells were reseeded and incubated for 7 days. Data are presented as

the mean ± SEM (standard error of the mean), *n* = 3 independent experiments.
**g** Viability of SMARCB1-proficient (786-O) and SMARCB1-deficient (HS-ES-2R) cell lines transfected with or without siRNAs targeting *EP300*, *KREMEN2*, and/or *KREMEN1*. Cells were transfected with indicated siRNAs for 48 h. Cells were reseeded and transfected with indicated siRNAs for 48 h. The cells were reseeded and incubated for 7 days. Data are presented as the mean ± SEM, *n* = 3 independent experiments. **h**, **i** Expression of *CASP6* mRNA (relative to that in siNT (non-targeting)-transfected cells) in SMARCB1-deficient JMU-RTK-2 (**h**) and HS-ES-2R (**i**) cells transfected with or without siRNAs targeting *EP300* or *KREMEN1*, for 96 h. Data are presented as the mean ± SD, *n* = 3 independent experiments. **j**, **k** Expression of *CASP6* mRNA (relative to that in siNT-transfected cells) in SMARCB1-deficient JMU-RTK-2 (**j**) and HS-ES-2R (**k**) cells transfected with siNT, or with siRNAs targeting *KREMEN2* or *KREMEN1*, for 24 h. Data are presented as the mean ± SD, *n* = 3 independent experiments. **l**, **m** Expression of *CASP6* mRNA (relative to that in siNT-transfected HS-ES-2R mock cells) in HS-ES-2R mock and HS-ES-2R +KREMEN2 cells transfected with siNT, or with siRNAs targeting *KREMEN2* (**l**) or *CREBBP* + *EP300* (**m**), for 96 h. Data are presented as the mean ± SD, *n* = 3 independent experiments.
**n** Expression of *CASP6* mRNA (relative to that in HS-ES-2R mock cells) in HS-ES-2R mock and HS-ES-2R +KREMEN2 cells treated without or with 0.2 or 0.4 μM CP-C27 for 24 h. Data are presented as the mean ± SD, *n* = 3 independent experiments. **o** NanoBiT activity of KREMEN1 (relative to that in non-treated cells) in HS-ES-2R NanoBiT cells (HS-ES-2R +KREMEN1-SmBiT +KREMEN1-LgBiT) treated with 0.2 or 0.4 μM CP-C27 for 24 h. Data are presented as the mean ± SEM, *n* = 3 independent experiments. **p**, NanoBiT activity of KREMEN1 (relative to that in siNT-transfected cells) in HS-ES-2R NanoBiT cells (HS-ES-2R +KREMEN1-SmBiT +KREMEN1-LgBiT) transfected with siNT (−), or with siRNAs targeting *KREMEN2* (+), for 96 h. Data are presented as the mean ± SEM, *n* = 3 independent experiments. **q** NanoBiT activity of KREMEN1 (relative to that in cells without *KREMEN2* cDNA) in HS-ES-2R NanoBiT cells (HS-ES-2R +KREMEN1-SmBiT +KREMEN1-LgBiT) transduced without (−) or with the *KREMEN2* cDNA vector (+). Data are presented as the mean ± SEM, *n* = 3 independent experiments. For all experiments, *p*-values were determined by an unpaired two-tailed Student's *t*-test. **p* < 0.05, ***p* < 0.01, ****p* < 0.001.

inhibiting CBP/p300 in SMARCB1-deficient cells underlies apoptosis mediated by KREMEN1.

A previous study shows that KREMEN1-mediated apoptosis is suppressed by overexpression of KREMEN2[35]. Correspondingly, upregulation of *CASP6* by depletion of *KREMEN2* from SMARCB1-deficient cells was prevented by overexpression of KREMEN2 (Fig. 5l). In addition, upregulation of *CASP6* in SMARCB1-deficient cells by depletion or inhibition of CBP/p300 was prevented by overexpression of KREMEN2 (Fig. 5m, n). These results indicate that KREMEN2 suppresses KREMEN1-mediated apoptosis. Thus, downregulating *KREMEN2* in SMARCB1-deficient cells by inhibiting CBP/p300 induces apoptosis through de-suppression of KREMEN1.

Next, we tried to examine expression of KREMEN1 protein. Although we could detect KREMEN1 protein expressed via the overexpression vector, we could not detect endogenous KREMEN1 protein using a commercially available anti-KREMEN1 antibody (Supplementary Fig. 5r–t) (as was the case for KREMEN2) (Supplementary Fig. 2j–l). We confirmed knockdown of *KREMEN1* mRNA by *KREMEN1* siRNA (Supplementary Fig. 5r); however, western blot analysis using commercially available KREMEN1 antibodies did not detect KREMEN1 proteins (as was also the case for KREMEN2). Many bands were detected by the anti-KREMEN1 antibodies, but none showed reduced intensity or disappeared after treatment with *KREMEN1* siRNA, even after long exposure to chemical luminescence agents (Supplementary Fig. 5s). In addition, in cells transduced with an KREMEN1 over-expression vector, we detected bands of KREMEN1 that disappeared upon treatment with *KREMEN1* siRNA (Supplementary Fig. 5t). The current commercially available KREMEN1 antibody detected KREMEN1 when expressed at high levels, but not at basal levels. This may have something to do with the specificity and affinity of the KREMEN1 antibody; however, we have not been able to prove that endogenous KREMEN1 proteins are expressed and functional. In the future, this

issue will be resolved by inserting an epitope tag into endogenous KREMEN1, or by creating anti-KREMEN1 antibodies with high specificity and affinity.

KREMEN1 and KREMEN2 are single-transmembrane proteins; however, they have opposite effects on induction of apoptosis: KREMEN1 triggers apoptosis[42], whereas KREMEN2 suppresses it[35]. Co-immunoprecipitation experiments suggest that KREMEN1 forms homodimers; however, formation of these homodimers is suppressed by overexpression of KREMEN2[35]. There are no studies reporting quantitative analyses of the effect of KREMEN2 on homodimer formation by KREMEN1. Therefore, we investigated the relationship between KREMEN1 and KREMEN2 with respect to protein–protein interactions. To do this, we constructed an assay system based on quantitative measurement of homodimer formation by KREMEN1 using the NanoBiT system. This system can analyze protein–protein interactions, such as the KREMEN1-KREMEN1 homo-interaction, in real-time in living cells. It does this by detecting luminescence signals generated when the individual components of NanoLuc luciferase, i.e., the Small BiT (SmBiT) comprising 13 amino acid residues, and the Large BiT (LgBiT) comprising 156 amino acid residues, complement each other to form an active luciferase molecule[43,44]. We constructed vectors in which SmBiT and LgBiT were fused to the C-terminal region of KREMEN1. The NanoBiT system emits luminescence when proteins tagged with SmBiT and LgBiT are co-expressed in cells, and each protein binds in close proximity in the cell. It is possible to measure the amount of binding between proteins based on the amount of luminescence emitted at this time, and express it as an index.

Therefore, we introduced KREMEN1 harboring a SmBiT tag and a LgBiT tag at the C-terminus into HEK293T cells and measured NanoBiT activity as a readout of the KREMEN1 homo-interaction (Supplementary Fig. 5u). No NanoBiT activity was observed when SmBiT alone or LgBiT alone was introduced into cells (Supplementary Fig. 5v).

However, NanoBiT activity was observed when both KREMEN1-SmBiT and KREMEN1-LgBiT were introduced (Supplementary Fig. 5v). Thus, KREMEN1-SmBiT and KREMEN1-LgBiT appear to homo-interact inside the cell. Therefore, we used the SMARCB1-deficient cell line HS-ES-2R to establish a KREMEN1-SmBiT/KREMEN1-LgBiT stably transduced cell line, called HS-ES-2R +KREMEN1-NanoBiT (Supplementary Fig. 5t). We then used this cell line to investigate the effects of KREMEN2 expression on the KREMEN1 homo-interaction. Treatment with CBP/p300 inhibitors attenuated NanoBiT activity in the HS-ES-2R +KREMEN1-NanoBiT cell line (Fig. 5o). In addition, NanoBiT activity was reduced by knockdown of *KREMEN2* (Fig. 5p, Supplementary Fig. 5w). By contrast, NanoBiT activity was increased by overexpression of KREMEN2 (Fig. 5q, Supplementary Fig. 5x). These results indicate that KREMEN2 increases the homo-interaction between KREMEN1 monomers. Thus, decreased expression of KREMEN2 via CBP/p300 inhibition promotes monomerization of KREMEN1, followed by induction of apoptosis.

## Downregulation of *KREMEN2* by CBP/p300 inhibition in SMARCB1-deficient cells suppresses anti-apoptotic signaling pathways

Here, we found that downregulation of *KREMEN2* by CBP/p300 in SMARCB1-deficient cells triggered apoptotic cell death via KREMEN1. A previous report shows that KREMEN1 is involved in induction of apoptosis[35]; however, the molecular pathways downstream of KREMEN1 that trigger apoptosis are still unclear. Therefore, we performed gene expression analysis in two SMARCB1-deficient cell lines (JMU-RTK-2 and HS-ES-2R) using RNA-seq after treatment with the CBP/p300 inhibitor A-485, after *CREBBP/EP300* depletion, and after *KREMEN2* depletion. To better understand the molecular pathways impacted by inhibition of CBP/p300, depletion of *CREBBP/EP300*, and depletion of *KREMEN2*, we carried out Gene Set Enrichment Analysis (GSEA)[45,46], and also made use of the Molecular Signatures Database Hallmark Gene Set collection, each of which can be used to identify a specific biological state or process and to identify genes involved in these signatures[47]. We then isolated significantly enriched signatures (i.e., $p < 0.05$, $q < 0.25$) among downregulated genes affected by treatment with A-485 (Fig. 6a), *CREBBP/EP300* knockdown (Fig. 6b), and *KREMEN2* knockdown (Fig. 6c). By contrast, there were no significantly enriched signatures among upregulated genes. Notably, we identified five overlapping downregulated gene signatures that correlated negatively with CBP/p300 inhibitor treatment, depletion of *CREBBP/EP300*, and depletion of *KREMEN2* (Fig. 6d). Of the five, we focused on two: TNFA-SIGNALING-VIA-NFKB and IL6-JAK-STAT3-SIGNALING (Supplementary Fig. 6a–c). This is because these signatures are associated with regulation of apoptosis[48,49]. Inhibition of the TNFα (tumor necrosis factor-α)/NF-kB (nuclear factor-kappa B) or IL-6 (Interleukin 6)/JAK2 (Janus Kinase 2)/STAT3 (signal transducer and activator of transcription 3) signaling pathways induces apoptosis[48,49], suggesting that downregulation of *KREMEN2* by CBP/p300 inhibition triggers apoptosis by suppressing the TNFα/NF-kB and IL-6/JAK2/STAT3 signaling pathways.

Next, we identified the genes within these signatures that are associated with apoptosis induction. Notably, core enrichment genes, which are the subset of genes that contributes most to the enrichment results, in each signature from TNFA-SIGNALING-VIA-NF-KB or IL6-JAK-STAT3-SIGNALING, overlapped with 11 genes and three genes, respectively, in these pathways (Fig. 6e), indicating that multiple genes in the gene sets of the TNFα/NF-kB or IL-6/JAK2/STAT3 signaling pathways are associated with apoptosis via the CBP/p300-KREMEN2-KREMEN1 axis. We also found that each of the gene sets overlapped with two genes, *CSF1* (colony-stimulating factor-1) and *SOCS3* (suppressors of cytokine signaling 3) (Fig. 6e). Importantly, suppression of CSF1 or SOCS3 is involved in induction of apoptosis[50–52]. CSF1 is a cytokine and binds to its receptor CSF1R (colony-stimulating factor-1 receptor), which is a receptor tyrosine kinase, and then induces

tyrosine phosphorylation CSF1R, leading to activation of RAS (rat sarcoma)-ERK (extracellular signal-regulated kinase), PI3K (phosphatidylinositol 3-kinase)-AKT (protein kinase B), and JAK2-STAT3 phosphorylation signaling[53–55]. By contrast, suppression of SOCS3 induces hyper activation of STAT3 phosphorylation signaling and reduces activation of PI3K-ATK phosphorylation signaling[51]. Thus, downregulation of *KREMEN2* upon inhibition of CBP/p300 in SMARCB1-deficient cells could impact phosphorylation signaling pathways such as RAS-ERK, PI3K-AKT, or JAK-STAT3. Therefore, to investigate the phosphorylation signaling pathways affected by CBP/p300 inhibition, we performed phospho-protein microarray analysis to identify phosphorylation proteins affected by CBP/p300 inhibition in SMARCB1-deficient cells. We found that treatment of JMU-RTK-2 -SMARCB1 cells with a CBP/p300 inhibitor markedly attenuated phosphorylation of AKT and its downstream protein PRAS40 (proline-rich AKT substrate of 40 kDa), but did not affect that of other TNFα/NF-kB or IL-6/JAK2/STAT3 signaling-related proteins such as STATs and ERKs (Fig. 6f, g). Therefore, we focused on the PI3K-AKT phosphorylation signaling pathway as a downstream signaling pathway affected by treatment with CBP/p300 inhibitors. We then confirmed that treatment of SMARCB1-deficient cell lines with CBP/p300 inhibitors A-485 and CP-C27 attenuated acetylation of histone H3K27ac and then reduced phosphorylation of AKT pS473 and PRAS40 pT246 (Fig. 6h, Supplementary Fig. 6d, e). By contrast, no phosphorylation of AKT was observed in the three SMARCB1-proficient cell lines (the exception was 786-O), but phosphorylation of PRAS40 was detected in all cell lines. CBP/p300 inhibition did not affect phosphorylation of AKT or PRAS40 (Supplementary Fig. 6f). In addition, simultaneous depletion of CBP and p300 in SMARCB1-deficient cells attenuated acetylation of H3K27ac and reduced phosphorylation of AKT pS473 and PRAS40 pT246 (Fig. 6i). Moreover, depletion of *KREMEN2* in SMARCB1-deficient cells also reduced phosphorylation of AKT pS473 and PRAS40 pT246 (Fig. 6j). The PI3K-AKT signaling pathway is a well-known oncogenic pathway involved in preventing apoptosis[56]. Therefore, these findings suggest that downregulating *KREMEN2* by inhibiting CBP/p300 in SMARCB1-deficient cells triggers induction of apoptosis by suppressing the PI3K-AKT signaling pathway.

Next, we investigated whether suppression of the AKT signaling pathway by CBP/p300 inhibition is mediated by KREMEN1. Attenuation of ATK phosphorylation by inhibition of CBP/p300 was partially rescued by depletion of *KREMEN1* (Fig. 6k). In addition, attenuation of ATK phosphorylation upon depletion of *KREMEN2* was also partially rescued by depletion of *KREMEN1* (Fig. 6l). These results indicate that downregulation of *KREMEN2* by inhibition of CBP/p300 suppresses the PI3K-AKT signaling pathway mediated by KREMEN1, followed by induction of apoptosis.

Therefore, we propose the following molecular mechanism to explain synthetic lethality upon simultaneous inhibition of CBP and p300 in SMARCB1-deficient cancers: in SMARCB1-proficient cells, the SMARCB1-containing SWI/SNF complex suppresses transcription of *KREMEN2*; this suggests that SMARCB1-proficient cells are not dependent on CBP/p300 and KREMEN2 (Fig. 6m). By contrast, SMARCB1 deficiency increases expression of *KREMEN2* mediated by both CBP and p300 in collaboration with the SMARCA1 chromatin remodeling complex, followed by suppression of KREMEN1 due to homo-dimerization, culminating in activation of anti-apoptotic signaling pathways (Fig. 6n). Thus, downregulation of KREMEN2 through inhibition of CBP/p300 leads to monomerization of KREMEN1, followed by induction of apoptotic cell death via suppression of anti-apoptotic signaling pathways (Fig. 6o).

## Treatment with a CBP/p300 dual inhibitor suppresses growth of tumor xenografts derived from SMARCB1-deficient cancer cells

To investigate the in vivo effect of the CBP/p300 dual inhibitor CP-C27, we used it to treat mice bearing subcutaneous xenografts. Twice daily

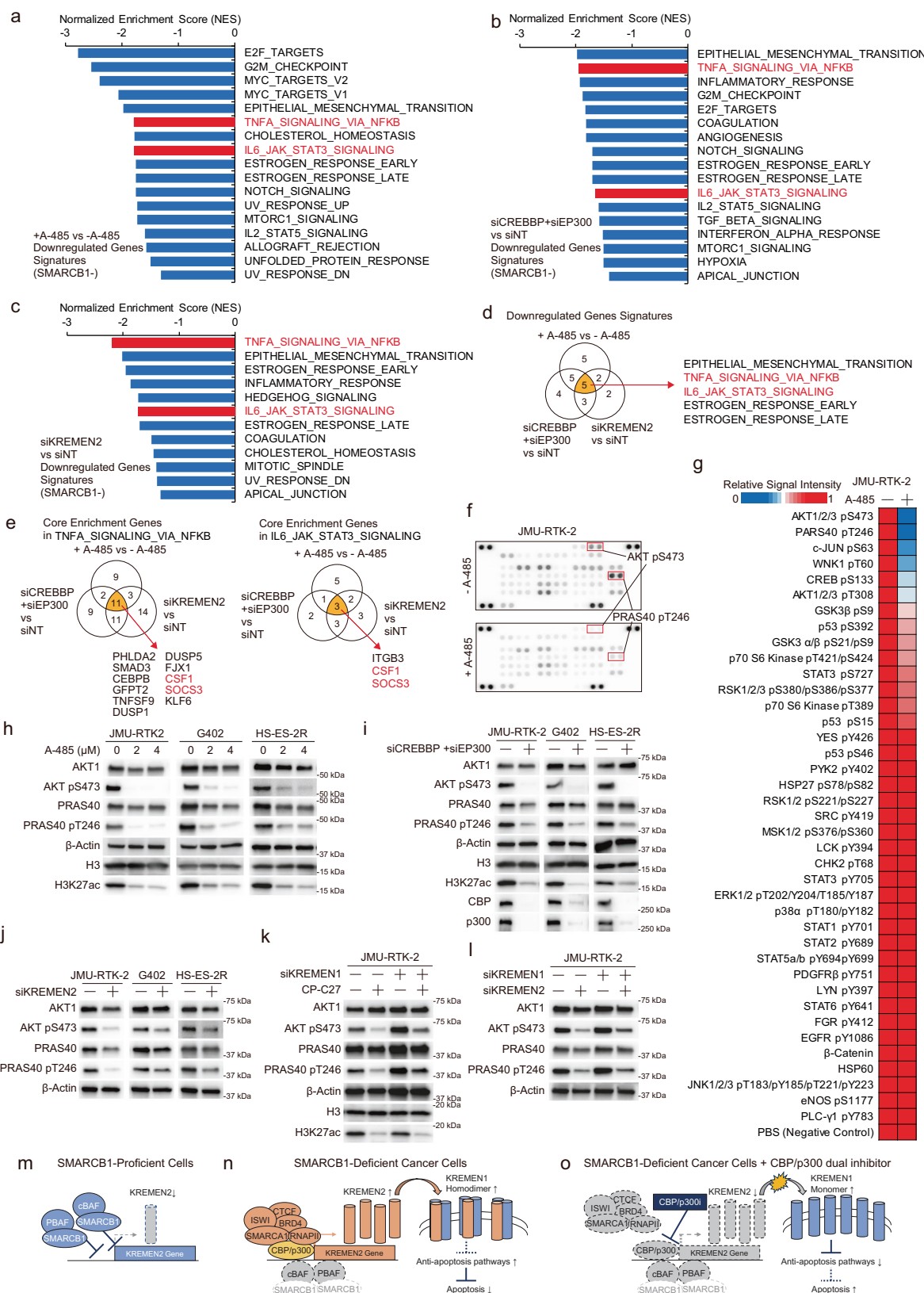

treatment with 3 mg/kg of CP-C27 led to significant suppression of tumors derived from SMARCB1-deificient G401 cells (Supplementary Fig. 7a). Higher doses (10 mg/kg to 30 mg/kg) drastically and significantly suppressed tumor growth by 77% and 97%, respectively (Supplementary Fig. 7b, c). Importantly, none of these doses had any effect on the body weight of mice (Supplementary Fig. 7d). In addition,

the concentrations of CP-C27 in the xenograft tumors, as well as the plasma, were >0.1 μM (Supplementary Fig. 7e), which was sufficient to suppress acetylation of H3K27 in SMARCB1-deficient cell lines (G401 and JMU-RTK-2) (Supplementary Fig. 7f, g). Indeed, we confirmed attenuation of H3K27ac in xenografts (Supplementary Fig. 7h), along with a significant reduction in tumor weight, upon treatment with

**Fig. 6 | Downregulation of *KREMEN2* by simultaneous inhibition of CBP/p300 in SMARCB1-deficient cells suppresses anti-apoptotic signaling pathways.**
**a**, **b**, **c** Normalized enrichment scores (NES) of significantly enriched biological signaling signatures identified among each of the gene sets downregulated upon treatment with 2 μM CBP/p300 inhibitor A-485 for 24 h (**a**) upon depletion of *CREBBP + EP300* for 96 h (**b**) or upon depletion of *KREMEN2* for 96 h (**c**) in SMARCB1-deficient cell lines (JMU-RTK-2 and HS-ES-2R), as determined by the Molecular Signatures Database Hallmark Gene Set collection in Gene Set Enrichment Analysis (GSEA). Gene sets with an FDR (False Discovery Rate) *q*-value < 0.25 and a Normal *p*-value < 0.05 were considered to be enriched significantly. *p*-values were determined by an unpaired two-tailed Student's *t*-test. **d** Identification of five significantly enriched biological signaling signatures that overlapped among gene sets downregulated upon treatment with 2 μM CBP/p300 inhibitor A-485 for 24 h, upon depletion of *CREBBP + EP300* for 96 h, or upon depletion of *KREMEN2* for 96 h. **e** Identification of core enrichment genes among the gene sets comprising TNFα/NF-kB- and IL-6/JAK2/STAT3-related signatures. Core enrichment genes are the subset of genes that contributes most to the enrichment result. Core enrichment genes in each TNFA-SIGNALING-VIA-NFKB or IL6-JAK-STAT3-SIGNALING signature overlapped with 11 genes and three genes, respectively, among core enrichment genes upon treatment with 2 μM CBP/p300 inhibitor A-485 for 24 h, upon depletion of *CREBBP + EP300* for 96 h, or upon depletion of *KREMEN2* for 96 h. **f** Immunoblot analysis of phosphorylated proteins in protein microarrays derived from SMARCB1-deficient JMU-RTK-2 cells treated without or with 2 μM A-485 for 16 h. **g** Heatmap showing the signal intensities of phosphorylated proteins (relative to that in non-treated cells) in SMARCB1-deficient JMU-RTK-2 cells treated without or with 2 μM A-485 for 16 h. **h** Immunoblot analysis of AKT1, AKT pS473, PRAS40, PRAS40 pT246, histone H3, H3K27ac, and β-actin expression in SMARCB1-deficient cell lines (JMU-RTK-2, G402, and HS-ES-2R) treated without or with 2 μM or 4 μM A-485 for 16 h. The experiments were repeated twice independently with similar results. **i** Immunoblot analysis of AKT1, AKT pS473, PRAS40,

PRAS40 pT246, histone H3, H3K27ac, CBP, p300, and β-actin expression in SMARCB1-deficient cell lines (JMU-RTK-2, G402, and HS-ES-2R) transfected with or without siRNAs for *CREBBP + EP300* for 96 h. The experiments were repeated twice independently with similar results. **j** Immunoblot analysis of AKT1, AKT pS473, PRAS40, PRAS40 pT246, and β-actin expression in SMARCB1-deficient cell lines (JMU-RTK-2, G402, and HS-ES-2R) transfected with or without siRNAs for *KREMEN2* for 96 h. The experiments were repeated twice independently with similar results. **k** Immunoblot analysis of AKT1, AKT pS473, PRAS40, PRAS40 pT246, histone H3, H3K27ac, and β-actin levels in SMARCB1-deficient cell lines (JMU-RTK-2) treated without or with CP-C27 after transfection with or without siRNAs for *KREMEN1*. Cells were transfected for 48 h with the indicated siRNAs. The cells were then reseeded and incubated for 24 h. The cells were then treated with 0.2 μM CP-C27 for 16 h. The experiments were repeated twice independently with similar results. **l** Immunoblot analysis of AKT1, AKT pS473, PRAS40, PRAS40 pT246, and β-actin expression in SMARCB1-deficient cell lines (JMU-RTK-2) transfected with or without siRNAs for KREMEN2 and/or KREMEN1, for 96 h. The experiments were repeated twice independently with similar results. **m** Schematic models of the proposed molecular mechanism explaining synthetic lethality upon simultaneous inhibition of CBP and p300 in SMARCB1-deficient cancers. In SMARCB1-proficient cells, the SMARCB1-containing SWI/SNF complex suppresses transcription of *KREMEN2*; this suggests that SMARCB1-proficient cells are not dependent on CBP/p300 and KREMEN2. **n** In SMARCB1-deficient cancer cells, SMARCB1 deficiency increases expression of KREMEN2 mediated by both CBP and p300 in collaboration with the SMARCA1 chromatin remodeling complex and transcription factors, resulting in suppression of KREMEN1 due to homodimerization and culminating in activation of anti-apoptotic signaling pathways. **o** In SMARCB1-deficient cancer cells treated with a CBP/p300 dual inhibitor, downregulation of KREMEN2 via inhibition of CBP/p300 leads to monomerization of KREMEN1, followed by induction of apoptotic cell death via suppression of anti-apoptotic signaling pathways.

10 mg/kg CP-C27 (Supplementary Fig. 7i). Moreover, once-daily treatment with CP-C27 led to marked suppression of tumor growth in another SMARCB1-deficient JMU-RTK-2 xenograft model (Fig. 7a, b), but not in a SMARCB1-proficient H460 xenograft model (Fig. 7c, d). Taken together, these results demonstrate that the CBP/p300 dual inhibitor CP-C27 suppresses growth of SMARCB1-deficient xenografts.

To further investigate the anti-tumor effects of depleting p300 from tumors, we established SMARCB1-proficient H460 cells and SMARCB1-deficient JMU-RTK-2 cells by doxycycline-induced knockdown of *EP300*. We confirmed that doxycycline decreased expression of p300 by JMU-RTK-2 shEP300 and H460 shEP300 cells, but not by JMU-RTK-2 shNT and H460 shNT cells (Supplementary Fig. 7j, k). Under these conditions, SMARCB1-deficient JMU-RTK-2 shEP300 cells lost viability, but SMARCB1-proficient H460 shEP300 cells did not (Supplementary Fig. 7l). Thus, we inoculated these cells into mice. The growth of xenograft tumors derived from SMARCB1-deficient JMU-RTK-2 shEP300 cells (Supplementary Fig. 7m, n), but not JMU-RTK-2 shNT cells (Supplementary Fig. 7o, p), was suppressed by treatment with doxycycline. By contrast, the growth of xenograft tumors derived from SMARCB1-proficient H460 shEP300 cells (Supplementary Fig. 7q, r) and H460 shNT cells (Supplementary Fig. 7s, t) was not suppressed by treatment with doxycycline. These results indicate that depletion of p300 in SMARCB1-deficient tumors suppresses tumor growth.

Next, we conducted immunohistochemical analyses to examine the molecular response of SMARCB1-deficient xenografts to dual CBP/p300 inhibition by CP-C27 (Fig. 7e). Treatment of SMARCB1-deficient xenografts with CP-C27 led to a significant reduction in the number of H3K27ac-positive cells (Fig. 7e, f), and conversely increased the number of cells positive for the apoptotic markers cleaved PARP (Fig. 7e, g) and cleaved caspase 3 (Fig. 7e, h). In addition, CP-C27 treatment attenuated AKT phosphorylation (Fig. 7e, i, j) as well as *KREMEN2* gene expression (Fig. 7k) in SMARCB1-deficient xenografts. These in vivo observations support the results obtained from the cell line models,

and indicate that a CBP/p300 inhibitor could suppress the growth of SMARCB1-deficient tumors.

## Discussion

Here, we identified a paralog pair, CBP/p300, as a promising therapeutic target for SMARCB1-deficient cancers. In principle, synthetic lethality is based on the "one-on-one" relationship between two genes, gene *A* and gene *B*[3,16]. Here, we showed that simultaneous inhibition of a paralog pair (e.g., gene *C1* and its paralog gene *C2*) causes synthetic lethality in cancer cells deficient in gene D. This concept illustrates the "two-on-one" relationship between two genes (*C1 + C2*) and another gene (D). In the present study, we show that simultaneous inhibition of two genes (*CREBBP + EP300*) causes synthetic lethality in cancers with a LOF mutation in another gene (i.e., *SMARCB1*). We focused on targeting a paralog pair rather than two different genes because paralog proteins are so similar, making it difficult to develop a selective inhibitor that targets either one. Indeed, most inhibitors target both proteins in a paralog pair. However, inhibitors of proteins with a paralog can inhibit multiple proteins; this is because some proteins have multiple paralogs and are encoded by multiple paralog genes. To simplify the screening system and subsequent analyses, we focused on paralog pairs encoded by two paralog genes. We found that simultaneous inhibition of paralog pairs, at least with respect to chromatin regulators, did not affect the growth of SMARCB1-proficient cells, even though it induced cell death in SMARCB1-deficient cells. Thus, it should be possible to develop advanced synthetic lethal therapies that inhibit both proteins in a paralog pair within a cancer cell harboring a gene with a LOF mutation.

In general, the SWI/SNF chromatin remodeling complex (which includes SMARCB1) plays a role promoting gene transcription by increasing chromatin accessibility; however, SWI/SNF-deficiency can promote gene expression despite reduced chromatin accessibility[10,32,57,58]. In this case, transcription-promoting factors that compete with the SWI/SNF complex could be a synthetic lethal target in SWI/SNF-deficient cancers. Indeed, we identified transcription-

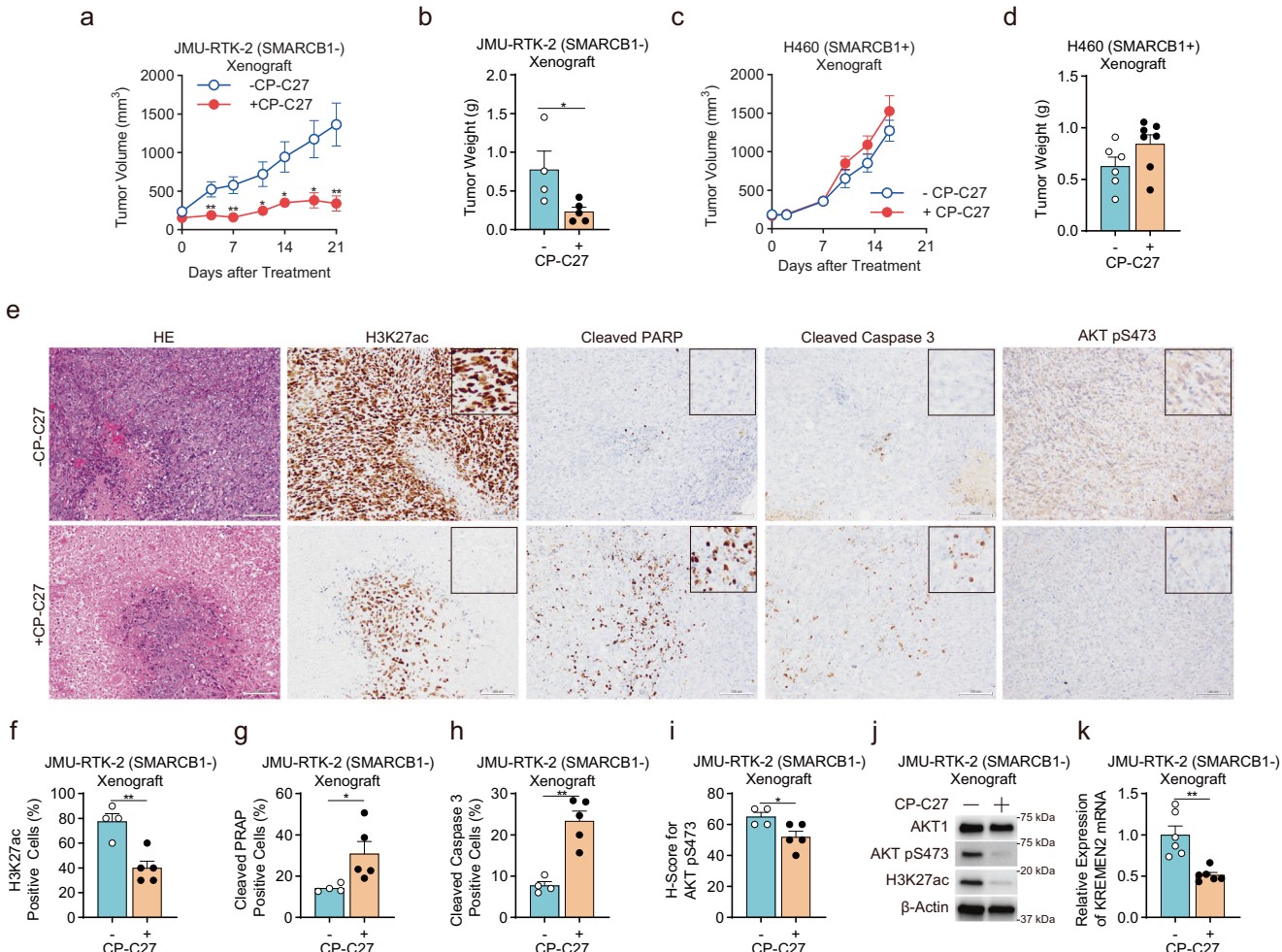

**Fig. 7 | Treatment with a CBP/p300 dual inhibitor suppresses growth of tumor xenografts derived from SMARCB1-deficient cancer cells. a, b,** Volume (**a**) and weight (**b**) of tumor xenografts derived from SMARCB1-deficient JMU-RTK-2 cell lines and harvested from mice treated with 10 mg/kg CP-C27 once a day. Data are presented as the mean ± SEM (standard error of the mean). -CP-C27 ($n = 4$ biologically independent mice per group), +CP-C27 ($n = 5$ biologically independent mice per group). **c, d** Volume (**c**) and weight (**d**) of tumor xenografts derived from SMARCB1-proficient H460 cell lines and harvested from mice treated with 10 mg/kg CP-C27 once a day. Data are presented as the mean ± SEM. -CP-C27 ($n = 6$ biologically independent mice per group) +CP-C27 ($n = 7$ biologically independent mice per group). **e** Representative immunohistochemical staining of H3K27ac, cleaved PARP, cleaved caspase-3, and AKT pS473 in xenografts derived from SMARCB1-deficient JMU-RTK-2 cell line and isolated from mice treated without or with 10 mg/kg CP-C27 once a day. Scale bar, 100 μm. **f, g, h, i** Percentage of cells positive for H3K27ac (**f**), number of cells positive for cleaved PARP (**g**), percentage

of cells positive for cleaved caspase 3 (**h**), and the H-score for cells positive for AKT pS473 (**i**) in xenografts derived from SMARCB1-deficient JMU-RTK-2 cells and isolated from mice treated without or with 10 mg/kg CP-C27 once a day. Data are presented as the mean ± SEM. -CP-C27 ($n = 4$ biologically independent mice per group), +CP-C27 ($n = 5$ biologically independent mice per group). **j** Immunoblot analysis of AKT1, AKTpS473, H3K27ac, and β-actin expression in xenografts derived from SMARCB1-deficient JMU-RTK-2 cells isolated from mice treated without or with 30 mg/kg CP-C27 once a day. **k** Expression of KREMEN2 mRNA (relative to that in -CP-C27 tumors) in xenografts derived from SMARCB1-deficient JMU-RTK-2 cells isolated from mice treated without or with 30 mg/kg CP-C27 once a day. Data are presented as the mean ± SEM. -CP-C27 ($n = 6$ biologically independent mice per group), +CP-C27 ($n = 6$ biologically independent mice per group). For all experiments, $p$-values were determined by an unpaired two-tailed Student's $t$-test. $*p < 0.05$, $**p < 0.01$, $***p < 0.001$.

promoting factors CBP and p300 as synthetic lethal targets in SMARCB1-deficient cells. Moreover, tazemetostat, an inhibitor of EZH2 within the PRC2 complex, has been approved for use against SMARCB1-deficient epithelioid sarcomas[30]. In the present study, we found that the CBP/p300 dual inhibitor CP-C27 showed higher selectivity and potency than tazemetostat in SMARCB1-deficient cells, indicating that CBP/p300 dual inhibitors such as CP-C27 may be promising treatments for SMARCB1-deficient cancers.

With respect to the mechanism underlying induction of synthetic lethality by simultaneous inhibition of CBP/p300 in SMARCB1-deficient cancers, we found that localization of the BAF and PBAF complexes, including SMARCB1, to the promotor region of the KREMEN2 gene locus leads to transcriptional repression of the KREMEN2

gene; this is because deficiency of SMARCB1 leads to an increase in expression of KREMEN2 by allowing CBP/p300 to localize at the promotor region. Inhibiting either CBP or p300 in SMARCB1-deficient cells led to a partial reduction in expression of the KREMEN2 gene, whereas inhibition of both CBP and p300 led to marked repression of KREMEN2. These observations were reflected by a decrease in cell viability. By contrast, SMARCB1 deficiency does not affect formation of the cBAF and PBAF complexes[6]. The residual SMARCB1-deficient cBAF and PBAF complexes remained localized at the TSS site of the KREMEN2 locus. Although the residual SWI/SNF complexes might function at any target gene locus, the residual complexes did not promote transcriptional expression of the KREMEN2 gene. Instead, a SMARCA1 chromatin remodeling complex was newly recruited to the KREMEN2 locus in

SMARCB1-deficient cells. The SMARCA1-NURF complex, a subtype of ISWI complex, plays a role in transcriptional promotion by randomizing nucleosome spacing to assist RNAPII activation[24]. This suggests that expression of the *KREMEN2* gene is required for viability for SMARCB1-deficient cells, and is induced redundantly by CBP and p300 in co-operation with the SMARCA1 chromatin remodeling complex.

Acetylation of H3K27 by CBP/p300 is required for recruitment of transcription factors. During transcription of the *KREMEN2* gene, we found that inhibiting CBP/p300 suppressed H3K27ac and reduced recruitment of not only RNA polymerase II, but also that of transcription factors BRD4 and CTCF, to the promoter and enhancer regions. However, single knockdown of BRD4 or CTCF did not affect expression of the *KREMEN2* gene. BRD4 and CTCF have their own paralogs: BRD2/BRD3 for BRD4, and CTCFL for CTCF. Thus, recruitment of H3K27ac by CBP/p300 to the transcriptional region of the *KREMEN2* gene locus is required for further recruitment of many other transcription factors and/or paralog pairs of transcription factors such as pan-BET (BRD4/3/2) and pan-CTCF (CTCF/CTCFL), leading to induction of *KREMEN2* gene expression.

Previously, co-immunoprecipitation experiments suggest that homodimer formation by KREMEN1 is suppressed by KREMEN2[35]. In this study, the NanoBiT system used to quantify the homo-interaction between KREMEN1 monomers revealed that KREMEN2 is required for homodimerization of KREMEN1. Thus, reduced expression of KREMEN2 due to inhibition of CBP/p300 triggers apoptosis through monomerization of KREMEN1. These observations are supported by previous studies showing that the N-terminal region of KREMEN1 (i.e., the ECD: Extra Cellular Domain) is required for homodimer formation[42]. Also, overexpression of KREMEN1-WT induces apoptosis, but overexpression of KREMEN1-ΔECDmut induces apoptosis even more strongly[42], indicating that apoptosis is more likely to occur under conditions that prevent homodimerization of KREMEN1. By contrast, as for KREMEN1, the N-terminal ECD domain of KREMEN2 is required for hetero-interaction with KREMEN1[35]. Overexpression of KREMEN1-WT alone induces apoptosis, but simultaneous overexpression of KREMEN1-WT and KREMEN2-WT suppresses induction of apoptosis[35]. However, simultaneous overexpression of KREMEN1-WT and KREMEN2-ΔECDmut (a KREMEN2 mutant that cannot bind to KREMEN1) does not suppress apoptosis[35], indicating that apoptosis is more likely to occur under conditions in which KREMEN1 cannot hetero-interact with KREMEN2 than under conditions in which KREMEN1 and KREMEN2 form heterodimers. In addition, KREMEN1 can also form trimers, which may include proteins other than KREMEN1[35]. Based on these phenomena, we propose the following model for the relationship between KREMEN2 expression and KREMEN1 homo-interactions during apoptosis: when expression of KREMEN2 is high, the homodimer formation by KREMEN1 increases, and then induction of apoptosis is suppressed by binding of the multimer to KREMEN2. However, when expression of KREMEN2 is downregulated, the amount of KREMEN1 homodimers decreases (i.e., the amount of KREMEN1 monomers increases), followed by induction of apoptosis. Thus, in SMARCB1-deficient cells, apoptosis is suppressed by high expression of KREMEN2. However, apoptosis can be induced by increasing KREMEN1 monomerization by downregulating *KREMEN2* using CBP/p300 inhibitors.

The signaling pathway downstream of KREMEN1 that triggers apoptosis has not been elucidated. Here, we found that downregulation of KREMEN2 by inhibiting CBP/p300 in SMARCB1-deficient cells may suppress anti-apoptotic pathways such as the IL-6/JAK2/STAT3 and TNFα/NF-kB pathways, which are commonly associated with CSF1 and SOCS3. Importantly, inhibition of CSF1 or SOCS3 induces apoptosis and suppresses the PI3K-AKT pathway[50–55]. In addition, downregulating *KREMEN2* by inhibiting CBP/p300 attenuated phosphorylation of AKT, which was rescued by depleting *KREMEN1*. Thus, it is suggested that KREMEN1 suppresses some anti-apoptotic pathways,

including the IL-6/JAK2/STAT3, TNFα/NF-kB, and PI3K-AKT pathway, to induce apoptosis. KREMEN1 is a transmembrane protein that suppress LRP5 and LRP6, which are themselves transmembrane proteins required for activation of the Wnt/β-catenin pathway[34]. KREMEN1 binds to LRP5/LRP6, resulting in internalization and depletion of the latter from the plasma membrane, followed by blockade of the Wnt/β-catenin pathway. By contrast, PI3K localizes to the plasma membrane where it plays a role in AKT phosphorylation by PDK1[59]. Taking into consideration the internalization of membrane proteins by KREMEN1, it is suggested that monomerized KREMEN1 interacts with PI3K, and then suppresses PI3K-AKT signaling via internalization of PI3K or PDK1.

At the very least, we showed that CBP/p300 inhibitors appear to have little effect on proliferation of normal cells, or on the body weight of mice. However, there are concerns about side effects due to the genome-wide effects of CBP/p300 inhibitors on expression of genes other than KREMEN2. In SMARCB1-deficient cells, suppression of KREMEN2 causes synthetic lethal, so KREMEN2 may also be a promising target. Compared with CBP/p300 inhibition, inhibiting KREMEN2 may have fewer side effects. Further discovery and development of KREMEN2 inhibitors based on studies of the KREMEN2-KREMEN1 interaction is expected.

Previously, only a single synthetic lethal target for a cancer mutation gene was identified by screening combined databases such as the DepMap project and the CCLE project. Advances in technology mean that comprehensive genomic screening can identify a set of two proteins as a synthetic lethal target. However, identification of two proteins by comprehensive genomic screening is difficult because searching for comprehensive combinations of two genes among all human genes generates so much NGS data that analysis and screening will overwhelm current technologies. If a paralog pair is used as the synthetic lethal target, simultaneous inhibition of that paralog pair is possible using a single inhibitor. The number of human proteins with paralogs is quite limited; thus comprehensive genomic screening to identify such paralog pairs is realistic. Therefore, we will continue to develop synthetic lethal screening methods to identify more paralog pairs in the future.

## Methods

### Animal ethics statement

Mouse experiments were approved by the National Cancer Center (NCC) Animal Ethical Committee and the Institutional Animal Care and Use Committee of Sumitomo Pharma Co. Ltd under certificate protocol number T19-013-M06 and AN13340-B02, respectively, and were performed in accordance with the Act on Welfare and Management of Animals. The experiments were carried out according to the Standards relating to the Care and Keeping and Reducing Pain of Laboratory Animals. Mice were checked for clinical indications, tumor size and body weight as specified in the experimental licenses. Mice were killed before reaching the approved humane end points of either tumor size limit of 2000 mm$^3$ or body weight loss of 20% or whenever they showed apparent clinical signs of pain. The maximum tumor size/burden was never exceeded in the studies. Source data are provided for all in vivo experiments.

### Cell lines

Cells were maintained at 37 °C in a humidified incubator containing 5% $CO_2$. The culture medium comprised DMEM/F-12 (Wako, 048-29785) supplemented with 10% fetal bovine serum (FBS; Gibco/Life Technologies), 10% GlutaMAX Supplement (Gibco, 41550021), and 100 U/mL penicillin and 100 μg/mL streptomycin (Wako, 168-23191). The 786-O (CRL-1932), A-204 (HTB-82), ES-2 (CRL-1978), HEK293T (CRL-3216), H1048 (CRL-5853), H2009 (CRL-5911), H2228 (CRL-5935), H358 (CRL-5807), and H460 (HTB-177) cells were obtained from the American Type Culture Collection (ATCC). Caki-1 (JCRB0801), G-401 (JCRB9065), G-402 (JCRB9070), and JMU-RTK-2 (JCRB1484) cells were obtained

from the Japanese Collection of Research Bioresources (JCRB) Cell Bank. HS-ES-1 (RCB2364), HS-ES-2M (RCB2360), and HS-ES-2R (RCB2361) cells were obtained from the Riken Cell Bank (RCB). NEPS cells were provided by Dr. Hiroyuki Kawashima[60]. The 786-O, ES-2, H2228, H358, H460, G-401, G-402, JMU-RTK-2, HS-ES-1, cells were authenticated by PowerPlex 16 STR System (Promega) in 2020. Although other cell lines are not authenticated, these cell lines were used for functional experiments after <2 months of passaging post-receipt. All cell lines tested negative for mycoplasma by MycoAlert (Lonza, LT07-318). In this study, no sex- and gender-based analyses have been performed because we focus on genetic aberrations regardless of sex and gender.

## Materials

Compounds were purchased from Cayman (GSK2801), MedChemExpress (WM-1119, Inobrodib), Santa Cruz Biotechnology (UNC1999), Selleck (CPI-360, SGC-CBP30, tazemetostat, valemetostat), SIGMA (BI-9564, OF-1, UNC0642), and TOCRIS (A-485, GSK-J4, L-Moses, TP-472). CP-C27 was prepared using a method similar to that used for compound 21[29]. The ON-TARGETplus SMARTPool siRNAs specific for each target gene were purchased from Dharmacon (Supplementary Table 1).

## Generation of lentiviruses and virus-infected cells

The cDNA-expressing lentiviral vectors (pLOC-CMV-SMARCB1-Bsd; OHS5897-202619211; Dharmacon); (pLV-hEF1A-KREMEN2-Puro [NM_172229.3]; VB210629-1291rex, pLV-hPGK-KREMEN1-Linker-LgBiT-Neo [NM_032045.5]; VB230820-1166bht, pLV-hPGK-KREMEN1-Linker-SmBiT-Bsd [NM_032045.5]; VB230820-1167cfp; Vector Builder), shRNA-expressing pSMART-inducible lentiviral vectors (hEF1a-GFP-Puro-TRE-shNT; VSC11653, hEF1a-GFP-Puro-TRE-shEP300; V3SH11252-228017400), and packaging plasmids (psPAX2: #12260 and pMD2.G: #12259; Addgene) were used for constitutive lentiviral expression of shRNA or cDNAs. To generate viruses, 293LTV cells were transfected with lentiviral plasmids and packaging plasmids using Lipofectamine 3000 (ThermoFisher Scientific; L3000015). After 16–24 h, the medium was replaced with fresh growth medium and cells were incubated for 48 h. Lentivirus-containing supernatants were harvested and concentrated by centrifugation by using Lenti-X Concentrator (Takara, 631232). To establish cells infected with viral constructs, cells were transduced with lentivirus suspension containing 8 µg/mL polybrene (Nacalai Tesque, 12996-81), and then incubated for 24 h. The growth medium was then replaced with fresh medium. After 24–48 h, the cells were incubated for 3–7 days in growth medium containing 2 µg/mL puromycin (Wako, 160-23151), 500 µg/mL G-418 (Wako, 078-05961), or 20 µg/mL blasticidin (Wako, 029-18701). To establish JMU-RTK-2 +SMARCB1 cells, SMARCB1-deficient JMU-RTK-2 cells were transduced with lentiviruses derived from the pLOC-CMV-SMARCB1-Bsd lentivirus vector. After selection of blasticidin-resistant cells, a clone of JMU-RTK-2 cells expressing the SMARCB1 protein was isolated. To establish HS-ES-2R +KREMEN2 cells, SMARCB1-deficient HS-ES-2R cells were transduced with lentiviruses derived from the pLV-hEF1A-KREMEN2-Puro lentivirus vector. After selection of puromycin-resistant cells, HS-ES-2R cells expressing KREMEN2 protein were isolated. To establish HS-ES-2R, +KREMEN1-SmBiT, and +KREMEN1-LgBiT cells, SMARCB1-deficient HS-ES-2R cells were co-transduced with lentiviruses derived from the pLV-hPGK-KREMEN1-Linker-LgBiT-Neo and pLV-hPGK-KREMEN1-Linker-SmBiT-Bsd lentivirus vectors. After selecting blasticidin-resistant cells with G418, HS-ES-2R cells expressing KREMEN1-SmBiT and KREMEN1-LgBiT protein were isolated. To establish JMU-RTK-2-shNT, JMU-RTK-2-shEP300, H460-shNT, and H460-shEP300 cells, SMARCB1-deficient JMU-RTK-2 and SMARCB1-proficient H460 cell lines were transduced with lentiviruses derived from the hEF1a-GFP-Puro-TRE-shNT or hEF1a-GFP-Puro-TRE-shEP300 vectors. After selection with puromycin, resistant JMU-RTK-2-shNT, JMU-RTK-2-shEP300, H460-shNT, and H460-shEP300 cells were isolated.

## Cell viability assay

To measure cell viability after siRNA transfection, cells were trypsinized, counted, and reseeded in 24-well plates at a density of $1-2 \times 10^5$ cells per well. Next, the cells were transfected with siRNAs (50 nM) using Lipofectamine RNAiMAX (Thermo Fisher Scientific; 13778150). After 48 h, cells were trypsinized, reseeded in 24-well plates, and transfected repeatedly with siRNAs (50 nM) using Lipofectamine RNAiMAX. Cells were trypsinized after a further 48 h, counted, and reseeded in 96-well plates at a density of 250–1,000 cells per well. After 7 days, cell viability was examined by measuring cellular ATP levels in the CellTiter-Glo Luminescent Cell Viability Assay (Promega, G7571). To measure cell viability after treatment with an inhibitor, cells were trypsinized, counted, and reseeded in 96-well plates at a density of 250–1,000 cells per well. After 24 h, cells were treated with the indicated concentrations of inhibitors. After 6 days, cell viability was measured using the CellTiter-Glo Luminescent Cell Viability Assay (Promega, G7571). To measure cell viability after doxycycline treatment, shNT or shEP300 cells were trypsinized, counted, and reseeded in 24-well plates at a density of $1-2 \times 10^5$ cells per well. Next, the cells were treated with 1 µg/mL doxycycline (Selleck; S5159). After 48 h, cells were trypsinized, reseeded in 24-well plates, and treated repeatedly with 1 µg/mL doxycycline. Cells were trypsinized after a further 48 h, counted, and reseeded in 96-well plates at a density of 250–500 cells per well. After 7 days, cell viability was examined by measuring cellular ATP levels in the CellTiter-Glo Luminescent Cell Viability Assay (Promega, G7571). Luminescence was measured using the Nivo plate reader (PerkinElmer). $IC_{50}$ values were calculated using Graphpad Prism 8.

## Quantitation of mRNA

To measure basal mRNA levels, $2 \times 10^4$ cells were plated into 96-well plates and incubated for 24 h. For drug-treated cells, $2 \times 10^4$ cells were plated into 96-well plates and incubated for 24 h. The medium was then replaced with the medium containing (or not) A-485 or CP-C27, and incubated for 24 h. To establish siRNA-transfected cells, $2 \times 10^4$ cells were plated into 96-well plates, transfected with siRNAs (50 nM) using Lipofectamine RNAiMAX (Thermo Fisher Scientific; 13778150), and incubated for 48 or 96 h. First, mRNA was extracted from all cell lines and cDNA was synthesized using the SuperPrep II Cell Lysis & RT Kit for qPCR (TOYOBO; SCQ-401). Aliquots of cDNA were subjected to quantitative PCR using the THUNDERBIRD Probe qPCR Mix (TOYOBO; QPS101) and TaqMan Gene Expression Assays (Thermo Fisher Scientific). Prior to extraction of mRNA, tumor xenograft samples were weighed and washed with PBS. Samples were cut into 3 mm squares (30 mg) and milled in liquid nitrogen prior to extraction of mRNA using the Animal Tissues protocol from the RNeasy Mini Kit (QIAGEN; 74104) and a QIAshredder (QIAGEN; 79654); the final elution volume was 50 µL. Next, cDNA was synthesized using PrimeScript RT Master Mix (Perfect Real Time) (Takara; RR036A). The following gene-specific primer/probe sets were used for TaqMan Gene Expression Assays: *BRD4* (Hs04188087_m1), *CASP6* (Hs00154250_m1), *CASP9* (Hs00962278_m1), *CREBBP* (Hs00932878_m1), *CTCF* (Hs00902016_m1), *EP300* (Hs00914212_m1), *KREMEN1* (Hs00230750_m1), *KREMEN2* (Hs00225867_m1), *KREMEN2* (pre-mRNA) (APGZPTG), *SMARCA1* (Hs00161922_m1), and *GAPDH* (Hs99999905_m1) (Thermo Fisher Scientific). PCR was performed in an ABI StepOnePlus Real-Time PCR System (Applied Biosystems) under the following conditions: denaturation at 95 °C for 15 s, followed by annealing and extension at 60 °C for 30 s (40 cycles). For each sample, the mRNA level of target genes was normalized to that of *GAPDH*. The target/GAPDH ratios were then normalized against those in control samples using the 2-ΔΔCt method.

## Western blot analysis

To extract proteins from whole cells, $5 \times 10^5$ cells were harvested, washed with PBS, and lysed with 150 µL of 1 × SDS sample buffer at 95 °C for 5 min. Chromatin was sonicated on ice (20 cycles of 15 s pulses; high setting; 15 s between pulses) using a Bioruptor (M&S Instruments). To extract protein from tumor xenografts, samples were weighed, washed with PBS, cut into 3 mm squares (30 mg), and milled in liquid nitrogen. The powdered tumor sample was mixed with 50 µL of RIPA buffer (Wako, 188-02453) supplemented with a proteinase inhibitor cocktail (Active Motif, 37490) and a phosphatase inhibitor cocktail (Active Motif, 37492), and then homogenized on ice using a Mini Cordless Grinder (Funakoshi). The homogenized tumor samples were diluted in an additional 450 µL of RIPA buffer and incubated for 30 min on ice. Whole cell lysates were mixed with 250 µL of 3 × SDS sample buffer and boiled at 95 °C for 5 min, and the chromatin was sonicated on ice by application of 20 cycles of 15 s pulses (high setting; 15 s between pulses) using a Bioruptor (M&S Instruments). For tumor xenografts derived from G401 cells, samples were lysed with 500 µL of RIPA buffer (CST, 9806) supplemented with Halt proteinase and phosphatase inhibitor cocktail (Thermo Fisher Scientific, 78440) using Precellys Evolution (M&S Instruments). The lysates were incubated for 5 min on ice, followed by centrifugation at 12,000 rpm for 10 min. The supernatants were then mixed with 250 µL of 3 × SDS sample buffer and boiled at 95 °C for 5 min. The cell lysates were quantified using Pierce 660 nm Protein Assay Reagent (Thermo Fisher Scientific, 22660) and Ionic Detergent Compatibility Reagent for Pierce™ 660 nm Protein Assay Reagent (Thermo Fisher Scientific, 22663). Next, 15 µg of protein was analyzed by immunoblotting. Proteins were separated by SDS-PAGE, transferred to PVDF membranes, and immunoblotted with the indicated antibodies. β-actin was used as a loading control. Membranes were blocked for 1 h at 25 °C with PVDF Blocking Reagent for Can Get Signal (TOYOBO, NYPBR01) and then probed for 1 h at 25 °C with Can Get Signal Solution 1 (TOYOBO, NKB-201) containing primary antibodies. After washing with TBS containing 0.1% Tween 20, the membranes were incubated for 30 min at 25 °C with TBS containing 0.1% Tween 20, 1% BSA, and horseradish peroxidase-conjugated anti-mouse (CST, 7076) or anti-rabbit (CST, 7074) secondary antibodies before visualization using Western Lightning ECL Pro (Perkin Elmer, NEL120001EA) or the ECL Prime Western Blotting System (Cytiva, RPN2232). Chemiluminescence signals were measured using a FUSION Chemiluminescence Imaging System (M&S Instruments) or an Amersham Imager 600 (Cytiva). Antibodies specific for the following proteins were used for immunoblotting: SMARCB1 (1:1000, CST, 91735), CBP (1:1000, CST, 7425), p300 (1:1000, CST, 54062), H3 (1:1000, CST, 4499), H3K27ac (1:1000, CST, 8173), KREMEN2 (1:1000, LSBio, LS-C165609), KREMEN1 (1:1000, LSBio, LS-C97716), β-catenin (1:1000, CST, 8480), AKT1 (1:1000, CST, 4691), AKTpS473 (1:1000, CST, 4060), PRAS40 (1:1000, CST, 2691), PRAS40pT246 (1:1000, CST, 2997), SMARCA1 (1:1000, CST, 12483), and β-actin (1:2000, CST, 4970 or 5125).

## HAT Enzymatic Assay

The ability of CP-C27 to inhibit p300 HAT activity was evaluated using a SensoLyte HAT (p300) Assay Kit (ANASPEC, AS-72172). Briefly, the recombinant p300 was incubated with CP-C27 for 10 min at room temperature. Then, an acetyl-CoA and histone H3 peptide were added, and the mixture was incubated for 30 min at 37 °C. The enzymatic reaction was stopped by addition of stop solution and the fluorogenic reaction was initiated by addition of developer solution. Finally, the mixture was incubated for 30 min at room temperature in the dark. Fluorescence was measured using a multi-plate reader (Ensight, Perkinelmer). To assess CBP HAT activity, recombinant CBP was purchased from Reactive Motif (No 31590) and evaluated using the SensoLyte HAT (p300) assay kit. Based on the measured fluorescence intensity, an $IC_{50}$ value corresponding to the concentration of the compound at which the enzyme reaction inhibition rate was 50% was calculated. Assessment of other HAT activities (shown in Supplemental Fig. 1n) was outsourced to Eurofins Cerep (Celle-Levescault, France).

## Annexin V/propidium iodide (PI) staining assay

The Annexin V−FITC/PI Apoptosis Detection Kit (Roche, 11858777001) was used to detect apoptotic cells. For drug-treated cells, $1 \times 10^5$ cells (JMU-RTK-2 +SMARCB1 cells, JMU-RTK-2 -SMARCB1, 786-O, HS-ES-2R) were plated into 24 well plates, incubated for 24 h, and treated without or with A-485 for 6 days. For siRNA-transfected cells, $2 \times 10^5$ cells (JMU-RTK-2 +SMARCB1 cells, JMU-RTK-2 -SMARCB1) were plated into 12-well plates and then transfected with siRNAs (50 nM) using Lipofectamine RNAiMAX (Thermo Fisher Scientific; 13778150). After 48 h, $5 \times 10^5$ cells were trypsinized, reseeded in 12-well plates, and then transfected repeatedly for 48 h with siRNAs (50 nM) using Lipofectamine RNAi-MAX. The cell pellets were washed with PBS and suspended in 1 × binding buffer, and then incubated in the dark for 10 min at 25 °C with Annexin V−FITC and PI. Fluorescence was analyzed on a Guava easyCyte HT cytometer (Millipore). Gating of live cells was based on SCC and FSC area parameters. Annexin V−FITC positive fractions were detected by assessing fractions whose signal intensity was higher than that of fractions abundant in non-treated samples. The percentage of Annexin V-positive cells was calculated using GuavaSoft software (v. 2.7).

## NanoLuc Binary Technology (NanoBiT) assay

HEK293T ($1 \times 10^4$) cells were plated into a 96-well plate and incubated for 24 h. The cells were transfected for 48 h with 200 ng/well KREMEN1-SmBiT and/or KREMEN1-LgBiT plasmids using 0.6 µL of FuGENE HD (Promega, E2311). To examine siRNA-transfected HS-ES-2R cells expressing KREMEN1-SmBiT and KREMEN1-LgBiT, $1 \times 10^5$ cells in a 24-well plate were transfected for 48 h with siRNAs (50 nM) using Lipofectamine RNAiMAX (ThermoFisher Scientific; 13778150). Next, $2 \times 10^5$ cells were reseeded into a 96-well plate and incubated for 24 h. To examine CP-C27-treated HS-ES-2R cells expressing KREMEN1-SmBiT and KREMEN1-LgBiT, $2 \times 10^5$ cells were plated onto a 96-well plate and incubated for 24 h. The cells were then treated with CP-C27 for 24 h. To examine HS-ES-2R cells expressing KREMEN1-SmBiT and KREMEN1-LgBiT without or with KREMEN2 expression vectors, $2 \times 10^5$ cells were plated onto a 96-well plate and incubated for 24 h. The luminescence generated by NanoBiT was measured using the Nano-Glo Live Cell Assay System (Promega, N2012)[43,44]. Cell viability was measured using the CellTiter-Glo Luminescent Cell Viability Assay (Promega, G7571). Luminescence intensity was measured using a Nivo plate reader (PerkinElmer). NanoBiT activity was normalized to cell viability.

## Phosphorylation profiling using an antibody array

Cells were trypsinized, counted, and reseeded in a 10 cm dish at a density of $4 \times 10^6$ and incubated for 24 h. The cells were treated without or with 2 µM A-485 for 16 h. Then, the cells were harvested and washed with PBS. Antibody array analysis was conducted using the Proteome Profiler Human Phospho-Kinase Array Kit (R&D Systems; ARY003C). Chemiluminescence was measured using a FUSION Chemiluminescence Imaging System (M&S Instruments). Signal intensity was measured using Image J 1.54 g Software[61]. The ratio of the signal intensity in cells treated with A-485 relative to that in untreated cells was calculated.

## RNA-seq analysis

To examine A-485-treated cells, $5 \times 10^5$ cells (JMU-RTK-2 +SMARCB1 cells, JMU-RTK-2 -SMARCB1, 786-O, HS-ES-2R, G402 and NEPS) were treated without or with 2 µM A-485 or 0.2 µM CP-C27 for 24 h. To examine siRNA-transfected cells, $5 \times 10^5$ cells (JMU-RTK-2 + SMARCB1 cells, JMU-RTK-2 -SMARCB1, 786-O, HS-ES-2R) cultured in

6-well plates were transfected with siRNAs (50 nM) using Lipofectamine RNAiMAX (Thermo Fisher Scientific; 13778150) for 48 h. Next, $5 \times 10^5$ cells were trypsinized, reseeded in 6-well plates, and transfected repeatedly for 48 h with siRNAs (50 nM) using Lipofectamine RNAiMAX. The cells were then washed with PBS, and RNA was isolated using the RNeasy Mini Kit (Qiagen). Library preparation for RNA-seq samples was performed in the Rhelixa using the Poly(A) mRNA Magnetic Isolation Module (New England Biolabs) and NEBNext Ultra II Directional RNA Library Prep Kit for Illumina (New England Biolabs). Sequencing was performed on an Illumina Novaseq 6000, with 150 bp paired end reads for RNA-seq.

## ChIP-seq analysis
JMU-RTK-2 +SMARCB1 cells and JMU-RTK-2 -SMARCB1 cells ($1 \times 10^7$) were plated on a 150 mm dish and incubated for 24 h. For the chromatin immunoprecipitation (ChIP) experiments, library preparation and NGS sequencing for ChIP-seq analysis were performed by Active Motif. Cells ($1 \times 10^7$) were fixed for 15 min at 25 °C with 1% formaldehyde and the reaction was stopped by addition of 125 mM glycine. The fixed cells were washed twice with PBS, and chromatin was isolated by addition of lysis buffer, followed by disruption using a Dounce homogenizer. Lysates were sonicated and the DNA was sheared to an average length of 300–500 bp. Genomic DNA (Input) was prepared by treating aliquots of chromatin with RNase and proteinase K, and heated for de-crosslinking, followed by ethanol precipitation. Pellets were resuspended and the resulting DNA was quantified in a NanoDrop spectrophotometer (Thermo Scientific). Extrapolation to the original chromatin volume allowed quantitation of the total chromatin yield. An aliquot of chromatin (30 μg) was precleared with protein A agarose beads (Invitrogen). Regions of interest in genomic DNA were isolated using 4 μg of antibody specific for p300 (Santa Cruz, sc-585). Complexes were washed, eluted from the beads using SDS buffer, and treated with RNase and proteinase K. Crosslinks were reversed by incubating overnight at 65 °C, and ChIP DNA was purified by phenol-chloroform extraction and ethanol precipitation. Illumina sequencing libraries were prepared from the ChIP and Input DNAs using the standard consecutive enzymatic steps of end-polishing, dA-addition, and adapter ligation. After a final PCR amplification step, the resulting DNA libraries were quantified and sequenced on Illumina's NextSeq 500, with 75 bp single end reads.

## CUT&RUN-seq analysis
To assess proliferating cells, JMU-RTK-2 +SMARCB1 cells and JMU-RTK-2 -SMARCB1 cells ($5 \times 10^5$) were plated on a 6-well plate and incubated for 24 h. For drug-treated cells, $5 \times 10^5$ cells were plated on a 6-well plate, incubated for 24 h, and then treated for 24 h without or with 2 μM A-485 or 0.2 μM CP-C27. CUT&RUN analysis was performed using the CUT&RUN Assay Kit (CST, 86652). The cells were trypsinized, and $1 \times 10^5$ cells were harvested, suspended in 1× Wash Buffer, and attached to concanavalin A beads. Cells were then permeabilized with 100 μL of Antibody Binding Buffer containing digitonin, and incubated at 4 °C for 2 h with 2 μL of an antibody specific for H3K27ac (CST, 8173), H3K27me3 (CST, 9733), H3K4me1 (CST, 5326), H3K4me3 (CST, 9751), SMARCB1 (CST, 91735), RNAPII (Rpb1) (CST, 2629), CTCF (CST, 3418), or BRD4 (CST, 13440). Then, cells were washed four times with Digitonin Buffer, incubated at 4 °C for 1 h with pAG-MNase solution, and washed four times with Digitonin Buffer. MNase was activated by addition of 3 μL of $CaCl_2$ solution at 4 °C for 30 min prior to addition of 1 x Stop Buffer. In addition, genomic DNA (Input) was prepared by treating $1 \times 10^5$ cells with DNA Extraction Buffer containing RNase and proteinase K, followed by heating at 55 °C for 1 h. The chromatin was sonicated to yield fragments with an average length of 150–500 bp (40 cycles of 15 s pulses on ice [30 s between pulses]) using a Bioruptor (M&S Instruments). Chromatin and input DNA were purified using DNA Purification Buffers and Spin Columns (CST, 14209). Libraries were

created by Takara Bio using the ThruPLEX DNA-Seq Kit (Takara, R400674) and amplified for 16 cycles under the following conditions: denaturation at 98 °C for 20 s, followed by annealing and extension at 72 °C for 50 s. The samples were sequenced by Takara Bio using the NovaSeq 6000 platform (Illumina), with 150 bp paired end reads.

## CUT&RUN-qPCR (quantitative PCR)
For proliferating cells, JMU-RTK-2 +SMARCB1 cells and JMU-RTK-2 -SMARCB1 cells ($5 \times 10^5$) were plated on a 6-well plate and incubated for 24 h. For drug-treated cells, $5 \times 10^5$ cells were plated on a 6-well plate, incubated for 24 h, and then treated for 24 h without or with 2 μM A-485 or 0.2 μM CP-C27. Following trypsinization, $1 \times 10^5$ cells were harvested and subjected to CUT&RUN using 2 μL of an antibody specific for H3K4me3 (CST, 9751), H3K4me1 (CST, 5326), H3K27ac (CST, 8173), H3K27me3 (CST, 9733), CBP (Abcam, ab253202), p300 (CST, 54062), EZH2 (CST, 5246), SMARCB1 (CST, 91735), SMARCA4 (Abcam, ab110641), ARID1A (Abcam, ab182560), PBRM1 (CST, 89123), SS18 (CST, 21792), GLTSCR1 (CST, 45441), or SMARCA1 (CST, 12483), or normal IgG (CST, 66362), as described in the section "CUT&RUN-seq analysis". CUT&RUN-qPCR was performed as follows: 2 μL of chromatin DNA (sample immunoprecipitated by antibody targeting a protein or by negative control normal IgG antibody) or input DNA were mixed with 10 μL of THUNDERBIRD Next SYBR qPCR Mix (TOYOBO; QPX-201) and 0.3 μM of each primer mix (final reaction volume, 20 μL). Quantitative PCR was performed in the StepOnePlus Real-Time PCR System (Applied Biosystems) using the following protocol: 95 °C for 30 s, followed by 40 cycles at 95 °C for 5 s, and 60 °C for 30 s. The primer sequences used are shown in the Supplementary Table 2. A standard curve was produced by performing qPCR with a primer set targeting input DNA (in triplicate). The immunoprecipitated samples and negative control normal IgG samples were run alongside the dilution series of the input DNA standards. The quantity of immunoprecipitated DNAs and normal IgG DNAs was calculated from the standard curve, and then the fold enrichment of the immunoprecipitated samples relative to the normal IgG sample was calculated.

## ATAC-seq analysis
JMU-RTK-2 + SMARCB1 cells and JMU-RTK-2 -SMARCB1 cells ($5 \times 10^5$) were treated without or with 2 μM A-485 or 0.2 μM CP-C27 for 24 h. Next, $5 \times 10^5$ cells were cultured in 6-well plates and transfected for 48 h with siRNAs (50 nM) using Lipofectamine RNAiMAX (Thermo Fisher Scientific; 13778150). ATAC-seq analysis was performed by DNAFORM. Cells ($1 \times 10^5$) were lysed and the transposition reaction was performed with Tn5 Transposase (Illumina, FC121-1030) at 37 °C for 30 min. The reaction liquid was purified using the MinElute PCR Purification Kit (Qiagen, 28004). Next, five cycles of PCR were conducted using custom Nextera PCR primers[62] and NEBNext Q5 Hot Start HiFi PCR Master Mix (NEB, M0543S). The number of additional PCR cycles was determined by qPCR of the partly amplified products. The PCR products were purified using Agencourt AMPure XP beads (Beckman Coulter, A63881), using double size selection (left ratio: 1.4 ×; right ratio: 0.5 ×), according to the manufacturer's protocol. The samples were sequenced using the MGI Tech DNBSEQ-G400R sequencer, with 150 bp paired end reads.

## Processing of NGS data
Raw sequencing data from RNA-seq were trimmed using trim-galore version 0.6.5-1 and mapped to the hg38 genome using HISAT2 version 2.2.1[63]. TPM values were calculated using Strand NGS ver 4.0 (TOMY). Genes showing significant changes in expression (i.e., $P < 0.05$ and a 2-fold change $|log2FC| > 1$) were identified by Strand NGS ver 4.0 (TOMY). For Venn diagram analysis and Wikipathway analysis of RNA-seq data, the log2-fold change values were plotted using Strand NGS ver 4.0 (TOMY). BigWig files were generated using the bamCoverage command from deepTools version 3.5.1, with parameters

--normalizeUsing CPM --binSize 10 --smoothLength 30[64], and then visualized by the Integrative Genomics Viewer (IGV) version 2.13.2[65]. Raw sequencing data from ChIP-seq, CUT&RUN-seq, and ATAC-seq were trimmed using fastp version 0.12.4[66] and mapped to the human reference genome (hg38) using Bowtie2 version 2.4.5, with parameters -k 1 --no-mixed --no-discordant -X 2000[67]. Prior to all downstream analyses, duplicate reads were removed using the MarkDuplicates command in picard-tools version 2.26.11 [http://broadinstitute.github.io/picard]. From the ChIP-seq and CUT&RUN data, CPM values in the genome tracks were calculated by subtracting those in the input tracks as the background value for each cell. BigWig files were generated using the bamCompare command from deepTools version 3.5.1, with parameters --operation subtract --normalizeUsing CPM --scaleFactorsMethod None --binSize 10 --smoothLength 30[64], and then visualized by the Integrative Genomics Viewer version 2.13.2[65].

### Gene set enrichment analysis (GSEA)
FeatureCounts version 2.0.6[68] was used to count mapped reads for GSEA 4.3.2[46]. The expression dataset was analyzed using the Molecular Signatures Database (MSigDB)[46]. GSEA identified significant gene sets at the top or bottom of the ranked gene sets that were differentially expressed between the compared gene sets. In the present study, the MSigDB hallmark gene sets were used. Gene sets with an FDR (False Discovery Rate) $q$-value < 0.25 and a Normal $p$-value < 0.05 were considered to be enriched significantly.

### Mouse xenograft model
Cells were counted and resuspended in a 1:1 mixture of PBS/Matrigel (Corning, 354234) (100 μL: 100 μL or 25 μL: 25 μL) on ice. Thereafter, cells (JMU-RTK-2, H460, and G401 [$5 \times 10^5$ cells/mouse]) were injected subcutaneously into the flank of 5–6 week-old female BALB/c-nu/nu mice (CLEA or Jackson Laboratory). When the tumors were palpable (about 14–21 days after implantation), mice were divided randomly into two groups. In the drug treatment group, mice were injected intraperitoneally with either vehicle (PBS or 0.5 w/v% Methyl Cellulose 400 Solution (FUJIFILM Wako, 133-17815)) or CP-C27 (1–100 mg/kg) once daily or twice daily for 14–28 days. In the doxycycline (Dox) treatment study, cells (JMU-RTK-2-shNT and JMU-RTK-2-shEP300 = $5 \times 10^5$ cells/mouse; H460-shNT = $2 \times 10^5$ cells/mouse; and H460-shEP300 = $5 \times 10^5$ cells/mouse) were injected into the flanks of 6-week-old female BALB/c-nu/nu mice. Once the tumors were palpable (about 14 days after implantation), mice were divided randomly into two groups and fed either a diet containing Dox (625 ppm) or a control diet. In other experiments, tumor growth was measured every few days using calipers. The volume of implanted tumors was calculated using the formula $V = L \times W^2/2$, where V is volume (mm³), L is the largest diameter (mm), and W is the smallest diameter (mm). At the end of the experiment, mice were sacrificed in accordance with standard protocols. For pharmacokinetics (PK) analysis, blood and tumor tissue were harvested 4 h after the final administration. Plasma was prepared from blood by centrifugation at 3000 g for 10 min at 4 °C. The plasma samples were stored at −80 °C until measurement. The tumor homogenate was prepared by addition of methanol (a volume of four times the tumor weight) to the collected tumor followed by crushing with a bead homogenizer (TOMY) under cooling at 6000 g for 20 s. The concentration of the plasma samples and the tumor homogenates was measured by liquid chromatography mass spectrometry (LC-MS) (SCIEX).

### Immunohistochemistry
Xenografts were fixed immediately in 10% neutral buffered formalin solution. After 24 h, the xenografts were set in tissue processing cassettes measuring 3.5 × 2.5 × 0.4 cm. The specimens were dehydrated by passage through a series of ethanol solutions, beginning with 70% ethanol and finishing with 100%. Next, the ethanol in the tissue was

replaced by xylene, which is miscible with paraffin. Finally, the tissue specimens were infiltrated and embedded in paraffin. The steps from dehydration to paraffin infiltration were fully automated and performed by Tissue-Tek VIP 6 AI (Sakura Finetek, Japan). Then, the paraffin blocks were sectioned (4 μm thick) prior to H&E and IHC staining. Tissue sections were stained using antibodies specific for cleaved caspase-3 (Asp175) (clone 5A1E, #9664, 1:200, citrate buffer, CST), cleaved PARP (Asp214) (clone D64E10, #5625, 1:100, citrate buffer, CST), H3K27ac (clone EP16602, 1:2000, tris-EDTA buffer pH 9.0, Abcam), and AKT pS473 (clone D9E, #4060, 1:200, citrate buffer, CST). All IHC staining was performed using a Dako autostainer Link48 (Agilent Technologies). The percentage of H3K27ac-positive tumor cells within the total tumor cell population on each slide was calculated. Cleaved caspase-3 was evaluated by avoiding necrotic areas, and the percentage of cytoplasmic-positive tumor cells within the total tumor cell population was calculated. Cleaved PARP-positive cells within the total number of viable cells per slide were evaluated, and the mean number of positive tumor cells in three high-power fields (× 400) in a hotspot of the tumor tissue was reported. Immunohistochemical staining for AKT pS473 was evaluated using a semiquantitative approach, and a histological score (H-score) was assigned to tumor samples (Hirsch et al., 2003). First, membrane staining intensity (0, 1+, 2+, or 3+) was determined for each cell in a fixed field. The H-score was calculated using the following formula: [1 × (% cells 1+) + 2 × (% cells 2+) + 3 × (% cells 3+)]. The final score, ranging from 0–300, assigns more relative weight to higher-intensity membrane staining within a given tumor sample.

### Statistical analysis
Statistical analyses were performed using Microsoft Excel or Graphpad Prism 8. Data are expressed as the mean ± SD or as the mean ± SEM, as indicated in the figure legends. The sample size ($n$) is indicated in the figure legends, and represents the number of biological or technical replicates. Statistical significance was evaluated using a two-tailed Student's $t$-test. Statistically significant differences are indicated by asterisks as follows: *$p < 0.05$, **$p < 0.01$, and ***$p < 0.001$.

### Reporting summary
Further information on research design is available in the Nature Portfolio Reporting Summary linked to this article.

## Data availability
All raw NGS data files, as well as RNA-seq, ATAC-seq, CUT&RUN-seq, and ChIP-seq data, have been deposited as paired end fastq files, and all mapped data have been deposited as bigWig files in the NCBI Gene Expression Omnibus (GEO). The accession numbers for the data reported in this paper are [GSE237043]. Gene expression data from kidney-derived malignant rhabdoid tumor (MRT), cellular mesoblastic nephromas (CMN), and clear cell sarcomas of the kidney (CCSK) have been deposited in the Gene Expression Omnibus (GEO) database [GSE11482]. These downloaded data were normalized and counted using Transcriptome Analysis Console version 4.0 (Applied Biosystems). ChIP-seq, ATAC-seq, and RNA-seq datasets for TTC1240 were obtained from publicly available NCBI GEO datasets [GSE90634, GSE124903][36,69]. These downloaded data were analyzed as described for NGS data processing. Mutation, copy number (CN), and gene expression datasets were obtained from the Cancer Cell Line Encyclopedia (CCLE) database and downloaded from the DepMap website (data version 23Q2, [http://www.depmap.org/]). These downloaded data were analyzed as follows. To examine expression levels of the KREMEN2 gene among cell lines with different genetic abnormalities, mutation and CN data were combined to infer genotypes; this was done because tumor suppressor genes, including SMARCB1, can be lost by a combination of mutation and CN deletion. Specifically, the genotype of SMARCB1 or SMARCA4 was determined as the number of

altered allele(s) with mutations and/or CN deletions using the following methods. To obtain the number of mutated alleles, values (0, 1 or 2) stored in the table of damaging mutations from DepMap were used. The number of deleted alleles was defined as follows: when the measured CN was <0.5, the gene was defined as being homozygously deleted (deleted alleles 2); when the measured CN was <1.5 but ≥0.5, the gene was defined as being heterozygously deleted (deleted alleles 1). The number of mutated and deleted alleles was then summed to calculate the number of altered alleles. For a fraction of cell lines for which the calculated altered allele numbers exceeded 2, the numbers were set to 2. Cell lines with two altered alleles were considered to be SMARCB1- or SMARCA4-deficient. Since this assignation correctly determined SMARCB1 deficiency in seven of nine reported SMARCB1-deficient cell lines[70], this assignation was applied to all cell lines in CCLE. Finally, expression of the KREMEN2 gene was compared between deficient and non-deficient cell lines. Source data are provided with this paper.

## Code availability

No custom codes were used. The sources of the codes used in this study can be found in corresponding method sections. In this study, we used the following software: HISAT2 version 2.2.1[63], Strand NGS ver 4.0 (TOMY), deepTools version 3.5.1[64], Integrative Genomics Viewer (IGV) version 2.13.2[65], fastp version 0.12.4[66], Bowtie2 version 2.4.5[67], picard-tools version 2.26.11 [http://broadinstitute.github.io/picard], FeatureCounts version 2.0.6[68], and GSEA 4.3.2[46].

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

## Acknowledgements

We thank Rie Komatsuzaki, Harumi Hirano, Keiko Nakasato, Hinako Masuda, and Yuko Saida for technical assistance. We also thank Dr. Hitoshi Ban, Dr. Seiji Kamioka, Dr. Shunichiro Uesugi, Dr. Manabu Watanabe, Dr. Ryosaku Inagaki, Dr. Yusuke Sawayama, and Yudai Furuta for providing reagents and project support. We thank Dr. Hiroyuki Kawashima for providing cell lines. This study was supported by grants from JSPS KAKENHI (JP21H02796 to H.O.), an AMED grant (JP19cm0106162 to H.O.), a Sumitomo Pharma grant (C2019-095 to H.O.), the Princess Takamatsu Cancer Research Fund (19-25106 to H.O.), the Naito Foundation (2019 to H.O.), the Takeda Science Foundation (2019 to H.O.), the Japan Foundation for Pediatric Research (R1-3 to H.O.), and the Daiichi Sankyo Foundation of Life Science (2019 to H.O.).

## Author contributions

Conceptualization: H.O.; Data curation: M.S., D.K., K.M., S.T.; Formal analysis: D.K., K.M., H.Y., T.O., H.O.; Funding acquisition: H.O.; Investigation: M.S., D.K., K.M., H.Y., S.T., T.O., H.O.; Methodology: M.S., D.K., H.Y., S.T., H.O.; Project administration: M.S., T.O., H.O.; Resources: M.S., S.T., T.O., H.O.; Supervision: M.S., T.O., H.O.; Validation: M.S., D.K., K.M., H.O.; Visualization: M.S., H.Y., H.O.; Writing – review & editing: M.S., D.K., K.M., H.Y., S.T., T.O., H.O.

## Competing interests

H.O. received grants from Sumitomo Pharma during the study. D.K., K.M., T.O. are an employee at Sumitomo Pharma during the study. The other authors declare no potential conflicts of interest.
