## [Peer Review File · Nature Communications]

Targeting Dependency on a Paralog Pair of CBP/p300 against De-repression of KREMEN2 in SMARCB1-Deficient CancersReviewers' Comments:

Reviewer #1:

Remarks to the Author:

In this manuscript, the authors identified the paralog pair CBP/p300 as synthetic lethal target in SMARCB1-deficient cancers. They showed that CBP/p300 in SMARCB1-deficient cells can transcriptionally activate the expression of KREMEN2. Inhibition of CBP/p300 using siRNA knockdown or dual inhibitors can reduce KREMEN2 levels and induce apoptosis. Mechanistically, KREMEN2 can inhibit the homodimerization of KREMEN1, leading to AKT phosphorylation and activation. Overall, the story is interesting. However, I have quite some concerns on the rationales of identifying downstream mechanisms and the absence of experimental evidence about the impact of CBP/p300 to KREMEN1 homodimerization and KREMEN1-KREMEN2 heterodimerization, as well as certain writing issues that affect the logic and flow of the content, diminishing the overall persuasiveness of the data.

Major Concerns:

1. The main finding of this study is the identification of CBP/p300 paralog pair. However, the rationale on selecting the group of paralog pairs for screening and the reason for choosing the CBP/p300 paralog pair were missing. Firstly, the authors state "we selected paralog pairs of chromatin regulators that may also be a druggable target". How were this group of paralog pairs selected? Secondly, after the screening, the authors directly chose CBP/p300 paralog pair for follow-up studies without giving the reason why this pair was chosen. In Fig 1B, it might indicate the siCREBBP +siEP300 group showing the most significant change as the authors listed it at the first, which however was not described in the text. Therefore, it's not clear why the authors focused on the CBP/p300 paralog pair.
2. The authors state "We also screened existing inhibitors to identify potential paralog pairs". There is no reference on this statement. How were these inhibitors selected? Further descriptions are needed.
3. The authors state "CP-C27 selectively inhibited the HAT activity of both CBP and p300, but had no effect on other HATs". It's a bit confusing to show IC50 of the drug in the referenced figure. The authors should directly show the HAT activity with or without CP-C27 treatment and label the IC50 in the figure.
4. In Fig 2I, there is no significant difference of SMARCB1 peaks between 786-O and NEPS cell lines. As NEPS is a SMARCB1- cell line, the peaks of SMARCB1 should be negative just like the RNA-seq peak for 786-O. Are the peaks noise? The authors need to clarify this.
5. The authors conducted diverse types of sequencing to indicate that "SMARCB1 plays a role in transcriptional suppression of the KREMEN2 gene by localizing directly at the promotor region". However, the authors didn't show direct evidence to confirm this conclusion. Similarly, the evidence of the transcriptional regulation of CBP/p300 on KREMEN2 expression are also weak. For example, the authors should examine the pre-mRNA level of KREMEN2 after inhibiting SMARCB1 and CBP/p300 respectively and conduct ChIP-PCR or CUT&RUN-PCR to confirm the binding of SMARCB1 and CBP/p300 to the upstream of KREMEN2 gene.
6. Why did the authors focus on transcription factors BRD4 and CTCF? How were they selected? Do BRD4 and CTCF transcriptionally regulate KREMEN2? Any previous studies report this regulation? Comprehensive clarification needs to be done here.
7. In Fig 4C, why only upregulated genes were selected for pathway analysis as shown FC>2? And why were apoptosis-related pathways picked up?
8. Why did the authors focus on testing the impact of CBP/p300 dual inhibition on Wnt/ β -catenin and KREMEN1? It looks like the authors randomly pick up these two downstream mechanisms of KREMEN2 rather than evidence-based study.
9. The authors should show the impact of CBP/300 inhibition on KREMEN1 homodimerization and KREMEN1-KREMEN2 heterodimerization. This mechanism clarification was missing.
10. The authors state "we identified several common molecular pathways associated with suppression of CBP/p300 or KREMEN2 in SMARCB1-deficient cells". It is a very vague statement. What are "common molecular pathways"? And why the authors focus on cell proliferation-related pathways involved in phosphorylation and signal transduction? This is conflicted with previous conclusion that Simultaneous inhibition of CBP/p300 in SMARCB1-deficient cells induces apoptosis.
11. The rationale to conduct protein microarrays to detect phosphorylated proteins is weak. And the

reason for choosing AKT is also missing.

12. The authors state "Next, we investigated whether suppressing the AKT pathway by inhibiting CBP/p300 in SMARCB1-deficient cells affects the KREMEN2-KREMEN1 axis". AKT is the downstream of KREMEN2-KREMEN1 axis according to the results. Did the authors intend to say whether suppressing the AKT pathway by inhibiting CBP/p300 in SMARCB1-deficient cells is mediated by the KREMEN2-KREMEN1 axis?

13. In Fig 5H&I, KREMEN2 overexpression should increase AKT phosphorylation as it suppresses the KREMEN1 homodimer. However, the results showed here are opposite. Moreover, in Fig 5I, KREMEN2 overexpression increased the AKT protein level, thus the increased AKT pS473 may be due to the increased the AKT protein rather than enhanced phosphorylation.

14. Throughout the paper, statistical significance was not shown in a number of bar charts especially in supplementary figures.

15. The authors should show protein levels as the knockdown efficiency rather than RNA levels throughout the paper, such as CBP, p300, KREMEN1, KREMEN2. Moreover, is the SFig 3K is the overexpression efficiency for SFig 3L-N? If so, the authors should show the protein efficiency for each figure as L, M, and N are different experiments, one efficiency cannot represent all.

16. The manuscript and figure legends were not well written, which affects the logic and flow of the content and is hard to follow. For example, "Furthermore, synthetic lethality induced by CBP/p300 inhibition in SMARCB1-deficient cells via de-suppression of KREMEN1 and subsequent downregulation of KREMEN2 meant that synthetic lethality could be avoided via suppression of KREMEN1 through overexpression of KREMEN2 (Supplementary Fig. S4Y)". The shortening sentence would be "Synthetic lethality ... meant that synthetic lethality could..." what's the meaning of this sentence? Moreover, specific dataset IDs need to be clearly written including previous studies and this studies such as the legend for SFig 4F "RNA microarray data were obtained from the Gene Expression Omnibus database repository (GEO) used in a previous study". The authors should carefully revise the manuscript and probably rewrite the manuscript to make it clear, no bias, easy to follow and comprehensive.

Minor Concerns:

1. Please check that all abbreviations are spelled out the first time they are used, such as SMARCB1 and EZH2 in page 4.
2. Please check that references are properly cited throughout the manuscript. For example, the reference for "Paralog proteins are two very similar proteins that in general have a redundant function" in page 5 is missing.
3. The meaning of Fig 1C didn't match with the description in the text in page 6.
4. What's the difference between Fig 1K and the first two bars in Fig 1J and similarly between Fig 1L and the third and fourth bars in Fig 1J. Are they same experiment?
5. "we isolated the gene cluster whose expression fell specifically in SMARCB1-deficient cells, but not SMARCB1-deficient cells, after treatment with a CBP/p300 inhibitor". It should be "not SMARCB1-proficient cells".
6. "ND" needs to be spelled out in the legend. Please check abbreviations in all figure legends (e.g. Fig. S2F).
7. In Fig. S4B, actin is not even. Better quality figure should be provided.
8. In Fig. S6J, EP300 mRNA was also reduced by 20-30% in shNT groups. Is it significant? Protein levels are needed.

Reviewer #2:

Remarks to the Author:

This manuscript describes targeting paralog proteins CBP/p300 for synthetic lethality in SMARCB1-deficient cancers. The author identified the gene KREMEN2 which is differentially upregulated in SMARCB1-deficient cancers but decreased expression upon treating a CBP/p300 HAT inhibitor A-485. The chromatin state of both enhancer and promoter KREMEN2 is affected by the SMARCB1 expression level and inhibition of CBP/p300. The author also investigated the mechanism of synthetic lethality by KREMEN2 by showing that inhibition of KREMEN2 increases Caspase 6 and Caspase 9 by de-

suppression of KREMEN1 (although I have some confusion on this point described in detail below) and the PI3K/AKT pathway is identified. However, in contrast to the known function of KREMEN2, which inhibits canonical Wnt signaling, decreased KREMEN2 by inhibiting CBP/p300 in SMARCB1-deficient cancer function independent from the Wnt pathway. Lastly, the simultaneous inhibition of CBP/p300 significantly inhibited the SMARCB1-deficient cancer growth in vivo, concluding that CBP/p300 is the effective target of synthetic lethality of SMARCB1-deficient cancer. The majority of the work described simultaneous inhibition of a paralog pair method. Since most inhibitors in this study inhibit the paralog pair, the author suggests that simultaneous inhibition of a paralog pair method may be more effective than the traditional synthetic lethality, which inhibits a single target allowing redundancy to prevail.

Overall, I have great enthusiasm for this study. While it could use additional editing for clarity, it is overall well-written and conveys a very difficult topic well. The authors take an innovative approach and consider the often-overlooked role of SWI/SNF in transcriptional repression (many consider only transcriptional activation). Below I suggest some specific areas where clarity is required. I may have gotten confused in the double-negatives (which might still suggest that many readers would appreciate greater clarity), but there are a few spots where the authors may have confused themselves while writing. I also propose some experiments that would greatly strengthen and enhance the proposed model mechanistically. However, I give discretion to the editor and the authors as to the necessity of these experiments for publication in Nature Communications.

Major points for consideration: Not all of these need to be addressed by additional experiments. Some can be addressed in the writing or figures but are still major needs of clarification.

(1) In figure1, the author mentioned that CP-C27 was generated as an optimized A-485 analog, selectively inhibiting the HAT activity of CBP/p300 but having no effect on other HATs. Therefore, Figure 3E would have been stronger with CP-C27 instead of A-485.

(2) Since SMARCB1 is deficient in the cancer JMU-RTK-2 line, mapping of SMARCB1 ChIP alone cannot show the SWI/SNF's genome localization in SMARCB1-deficient cells. Therefore, investigating SWI/SNF distribution on genome targeting another subunit, such as Brg1 is necessary to compare how the complex distribution differs between SMARCB1-deficient and SMARCB1-rescued JMU-RTK-2 lines. Figure S3P of Mashlatir et al., 2018, shows that BAF and PBAF complexes are largely left intact without SMARCB1. This suggests two possible models in your system (1) There are residual complexes at KREMEN2 in SMARCB1- cells that are activating transcription, or (2) there are no SWI/SNF complexes at the locus at all, and WT SWI/SNF only acts to repress KREMEN2 transcription in a SMARCB1+ context? To answer this, one could start with a Brg1 or Baf155 ChIP (or another shared subunit), and if there are residual SWI/SNF complexes present in SMARCB1- cells, then one could use unique subunits (PHF10, ARID1s, etc.) to determine cBAF versus PBAF. If after these experiments are done and the model is one of repression in the SMARCB1+ WT context, what subcomplex is doing the repression? Is it cBAF or PBAF? One might think it is more likely to be PBAF based on other findings, but the authors have the ability to determine definitively which it is in this case, and it would greatly strengthen their model and this paper to do so. (Figure 2H-I, and Figure S2J)

(3) The first paragraph of page 11, "These results indicate that simultaneous inhibition of CBP and p300 in SMARCB1-deficient cells causes the synthetic lethality via induction of apoptosis" A knockdown of KREMEN2 would be more direct testing of the model proposed in this study than CBP/p300 inhibition here.

(4) The authors state, "... then induction of synthetic lethality by inhibition of CBP/p300 could be avoided by suppression of KREMEN1." This sentence confuses me, and it may be because of a typo. Do the authors mean KREMEN2? Because wouldn't the opposite of the sentence as written be true for KREMEN 1? KREMEN1 is proposed to be pro-apoptotic, so suppressing it would not be synthetic lethal. Do the authors mean activation or over-expression of KREMEN1? Side note: What will happen if KREMEN1 is overexpressed?

(5) I am confused as to the model of KREMEN1. Is the model an increase in the transcription (and therefore protein levels of KREMEN1) in the absence of KREMEN2, or is it that existing KREMEN1 can homodimerize and lead to a signaling cascade that results in apoptosis? I thought it was the latter, but then the authors suggest the former on page 13. If it is an increase in homodimerization (an elegant model), can the authors show an increase in homodimerization when KREMEN2 is knocked down or in CBP/p300 inhibition? These data would greatly strengthen the mechanism of the proposed model.

(6) Comparing Figure 5J and 5K, the control lanes of siNT (KREMEN2 -) have very different levels of AKT pS473 making interpretation and comparison harder. In J, there is more AKT-pS473 in the KREMEN2 + condition than KREMEN2 - condition, but the opposite is true in Fig 5K. It is unclear to me why this would be. It seems the model figure in Figure 5F suggests that KREMEN2 overexpression increases the phosphorylation of AKT. What makes the expression of KREMEN2 different in Figures 5K, 5H, and 5I (lanes 1 and 3)? And how does the phosphorylation of PRAS40 change in both conditions of Figure 5J and 5K? There are a lot of differences in AKT pS473 in siNT conditions that make it really hard to interpret and make conclusions from. It would be appreciated if the cell line names were included in Figure 5G-K, similar to Figure 5D-F. Additionally Figure 5 would benefit from the cell lines being in the figure panels and not just the figure legend.

Minor points:

I think "de-repression" over "de-suppression" is the more accurate and widely-used term, but the authors are welcome to use the one they think is best.

While described in the methods, it would be useful to briefly explain how SMARCB1 was reintroduced in the JMU-RTK-2 cells (lentivirus, right?) How does this expression compare to WT cells? While this is difficult to say due to imperfect direct comparisons, it is important to know if this brings SMARCB1 to roughly WT levels or is more akin to over-expression. (page 6)

Does Inobrodib inhibit other bromodomain-containing proteins (there are bromodomains in BAF complex subunits, for example), or does it only inhibit CBP/p300 (page 6)?

While the authors describe what the role of KREMEN2 protein is later in the manuscript, a brief description at the end of page 7 (after the text of "(Fig. 2A)", would be a good place).

Cancer treatment with synthetic lethality aims to selectively impose lethality on cancer cells without significantly affecting "non-cancer" cells. Therefore, I think you should move the HEK293T IC50 data from S1 to Figure 1J-L.

Page 8, the paragraph that begins with "To further investigate transcriptional regulation at the KREMEN2 gene locus, we observed the chromatin condition at the locus," I suggest replacing the phrase "observed the chromatin condition" with something else like "characterized the chromatin state." Observed and condition aren't quite the right words for this sentence.

While I pointed out some specifics in lack of clarity on page 11, I want to reiterate that there are other points I may not have directly pointed out, so this part of the manuscript should be carefully read and edited to make the language as precise as possible.

The basal level of beta-catenin is lower in SMARCB1-deficient cancer (Figure S4B). Is the Beta-catenin expression level also increased by overexpressing SMARCB1 (JMU-RTK +SMARCB1)?

For additional discussion: Is KREMEN2 a feasible drug target for synthetic lethality? As the authors suggest, this would reduce off-target inhibition effects of other CBP/p300 target genes during CBP+p300 inhibition.

Reviewer #3:

Remarks to the Author:

The study conducted by Sasaki et al. presents a synthetic lethal screening of chromatin regulator paralogs for SMARCB1-deficient tumors. The researchers discovered that the dual inhibition of CBP/p300 is lethal in SMARCB1-deficient tumors and performed in-depth molecular analysis. They show an involvement of KREMEN2 and KREMEN1, as well as the activation of AKT.

Given the highly aggressive nature of SMARCB1-deficient tumors and the lack of effective therapeutics for patients with this condition, the exploration of novel therapeutic targets is of utmost importance. The discovery of the synthetic lethal capacity of CBP/p300 dual inhibition in SMARCB1-deficient tumors is, therefore, relevant. Moreover, the screening approach using simultaneous inhibition of chromatin regulator paralogs is a novel and innovative aspect of the study.

However, despite the potential clinical significance of the findings, the data and strength of the molecular studies presented in the paper are somewhat limited. The selection criteria for the targets chosen for further studies, namely KREMEN2, remain unclear, and many of the validations rely solely on RT-PCRs. This includes the effects of KREMEN2 on KREMEN1 homodimerization and promotion of apoptosis. Additionally, the use of western-blot analysis to detect endogenous protein levels, especially for KREMEN2, did not yield any conclusive results across the samples.

In summary, while the study provides valuable insights into promising therapeutic targets for SMARCB1-deficient tumors, the molecular studies conducted are not sufficiently robust, thus limiting the overall strength of the findings.

Specific comments/concerns

-In their screening for synthetic lethal interactions in SMARCB1-deficient cells, the authors selected candidates based on siRNAs that affect cell growth in cells lacking SMARCB1 compared to isogenic cells with restored SMARCB1. However, it's important to note that the restitution of SMARCB1 itself has a potent effect on cell growth inhibition. How did the authors account for the effect of SMARCB1 restitution in their analysis?

-In Figure 1, a Western blot depicting the decrease in CREBBP and EP300 after depletion by siRNA should be included.

-It is challenging to comprehend the specific information presented in Figure 2A and Supplementary Fig 2A. The figure legends do not provide sufficient clarification either. On the other hand, the experimental design, and the approach for selecting KREMEN2 might not be the most optimal strategy. From what this reviewer understands, the researchers have chosen genes that exhibit common up/down-regulation in SMARCB1-deficient cells compared to SMARCB1wt cells. However, their initial comparison involves selecting genes that are commonly altered in cells after SMARCB1-restitution and in one SMARCB1-deficient cell line compared to one SMARCB1-proficient cell line from different tumor types. A more appropriate approach could have been restoring SMARCB1 in two or three different deficient cancer cell lines, identifying common genes that are affected and, subsequently, validating these candidate genes in a panel of SMARCB1-deficient and SMARCB1wt cells.

-The authors have performed different wide genome screenings such as bulk-RNAseq, ChIPseq, ATAC-seq. However, it is not clear which exact cell lines/cell models/conditions have been used in each case. This information could be included in the methods section. A more detailed analysis (genes Up/DOWN, for RNAseq, correlation RNAseq-ChIPseq, ATACseq, among others) is also lacking.

-What is the point of using siKREMEN2 when there is almost no mRNA in the SMARCB1-deficient cells (Suppl Fig 2C)?

-There was no detectable KREMEN2 protein observed in any of the western blots conducted to measure endogenous protein levels. The authors must provide an explanation for how a protein that is either not expressed or shows very low levels in all the cell lines can play a role in mediating cell death in SMARCB1-mutant tumors treated with EP300/CREBBP inhibitors. This is a crucial concern as it forms the foundation of their work and requires thorough explanation and addressing.

-The authors should utilize databases and previous datasets of RNAseq and ChIP-seq to validate their

observations. For example, they could examine a large panel of cancer cell lines that are SMARCB1-mutant and compare their KREMEN2 levels to those of SMARCB1-wt expressing cell lines. Additionally, they should investigate whether KREMEN2 levels decrease in other experimental datasets where SMARCB1 expression has been restored. To gain further insights, the authors should explore whether SMARCB1 directly binds to the KREMEN2 promoter in other available ChIP-seq data. Moreover, it would be valuable to analyze if there is any correlation between increased levels of KREMEN2 and mutations in other members of the SWI/SNF complex, such as SMARCA4.

-The ChIP-seq snapshots shown appear to be quite noisy. It is common for ChIP-seq data to exhibit some unspecific background signals. To address this issue, it is advisable to include positive controls, such as snapshots of regions that are well-known to recruit SMARCB1, P300, or are associated with the histone modifications being tested in this study. Incorporating snapshots of these positive controls will help to validate the specificity of the observed signals and ensure the reliability of the results.

-Why did the authors perform ChIP-seq of BRD4 and CTCF? The reason behind this needs to be explained.

-Again, WB of KREMEN1, not only RT-QPCR, should be included in the siKREMEN1 experiments (various panels on figure 4).

-On figure 5A, please include a list with the 8 common genes.

-Please, expand the number of SMARCB1 wt tested for pAKT inhibition following p300/CBP inhibition. It seems that only 1 has been tested (Fig 5D) and thus, a cell-specific effect cannot be discarded.

Other comments

-In Figure 1K and L the cell lines that have been used for these experiments should be indicated.

-For many of the experiments it is not indicated how many replicates have been used, ex: Fig 2B-F; Fig 3A-C, F-K, among others.

-Please indicate, for each of the figure panels, whether cancer cell lines are SMARCB1 wild type or mutant. Sometimes this has been indicated but not always, making the manuscript difficult to read.

-The manuscript will benefit from some English editing because, again, some sections/paragraphs are difficult to follow. For example, on page 7, lines 31-32 the paragraph: " (...) in SMARCB1-deficient cells, but not SMARCB1-deficient cells, after treatment with (...)" . Either there is a mistake, or the comma is misplaced.

-Please correct: SAMRCB1 (page 6 line 21); proficient (page 6, line 27); deificient (page 6, line 33); Matrials (Page 18)

Response Letter

REVIEWER COMMENTS

Reviewer #1 (Remarks to the Author):

In this manuscript, the authors identified the paralog pair CBP/p300 as synthetic lethal target in SMARCB1-deficient cancers. They showed that CBP/p300 in SMARCB1-deficient cells can transcriptionally activate the expression of KREMEN2. Inhibition of CBP/p300 using siRNA knockdown or dual inhibitors can reduce KREMEN2 levels and induce apoptosis. Mechanistically, KREMEN2 can inhibit the homodimerization of KREMEN1, leading to AKT phosphorylation and activation. Overall, the story is interesting. However, I have quite some concerns on the rationales of identifying downstream mechanisms and the absence of experimental evidence about the impact of CBP/p300 to KREMEN1 homodimerization and KREMEN1-KREMEN2 heterodimerization, as well as certain writing issues that affect the logic and flow of the content, diminishing the overall persuasiveness of the data.

Response

Thank you for your very detailed review. I am sorry that parts of the manuscript were not explained adequately. We have tried to explain the purpose and rationale in a more logical way in the revised version. In addition, we believe that the paper has become more convincing after inclusion of data from additional experiments that we conducted based on your suggestions.

Major Concerns:

1. The main finding of this study is the identification of CBP/p300 paralog pair. However, the rationale on selecting the group of paralog pairs for screening and the reason for choosing the CBP/p300 paralog pair were missing.

Firstly, the authors state “we selected paralog pairs of chromatin regulators that may also be a druggable target”. How were this group of paralog pairs selected?

Response

In this study, we searched for a promising synthetic lethal target for SMARCB1-deficient cancers from among chromatin regulators; we did this because

SMARCB1 is involved in chromatin regulation, and there are cases in which functionally related chromatin regulators show synthetic lethal properties. We selected 30 paralog pairs based on published molecular phylogenetic trees, as well as the protein structures of chromatin regulators such as histone acetylase, histone methylase, histone demethylase, and chromatin remodeling factor. These points have been clarified in the text, along with inclusion of additional supporting citations. (Lines 139-146)

KAT: *Biomolecules* 2021, 11(3), 455; doi.org/10.3390/biom11030455

KMT: *Clin Epigenet* 8, 102 (2016). doi.org/10.1186/s13148-016-0268-4

KDM: *Epigenetics & Chromatin* 10, 9 (2017). doi.org/10.1186/s13072-017-0116-6

CR: *Annu Rev Biochem.* 2009; 78:273-304.

doi.org/10.1146/annurev.biochem.77.062706.153223

Secondly, after the screening, the authors directly chose CBP/p300 paralog pair for follow-up studies without giving the reason why this pair was chosen. In Fig 1B, it might indicate the siCREBBP +siEP300 group showing the most significant change as the authors listed it at the first, which however was not described in the text. Therefore, it's not clear why the authors focused on the CBP/p300 paralog pair.

Response

"The siRNA screening shown in Figure 1b identified CREBBP+EP300, KDM3A+KDM3B, KMT2C+KMT2D, BRPF1+BRPF3, KDM6A+KDM6B, and PRDM8+PRDM13 as new paralog pair candidates that are synthetic lethal to SMARCB1-deficient cells. EZH2+EZH1 was also identified as an existing synthetic lethal target 25,26 during screening (**Fig. 1b**). We revalidated these candidates using other SMARCB1+ cells (786-O, H460) and SMARCB1-cells (G402, HS-ES-2R) to narrow them down (**Fig. 1c**). The data showed that the siRNA pair siCREBBP+siEP300 had little effect on the viability of SMARCB1-proficient cells but was lethal to SMARCB1-deficient cells, i.e., it was synthetically lethal. In addition, we used six SMARCB1-proficient cell lines (SMARCB1+) and six SMARCB1-deficient cell lines (SMARCB1-) (**Fig. 1d**) to confirm that simultaneous depletion of paralog pair CREBBP+EP300 (**Supplementary Fig. 1b, c, d**) decreased the viability of SMARCB1-deficient cell lines, but not that of SMARCB1-proficient cell lines (**Fig. 1e**). Therefore, we focused on CREBBP+EP300 as a

promising synthetic lethal target pair in SMARCB1-deficient cells. " (lines 150-165).

In addition, the original Fig S1A has been moved to the main manuscript (now cited as **Fig. 1c**).

2. The authors state "We also screened existing inhibitors to identify potential paralog pairs". There is no reference on this statement. How were these inhibitors selected? Further descriptions are needed.

Response

Originally, we intended to consider drug screening to reaffirm the siRNA screening. Therefore, we conducted drug susceptibility screening using existing inhibitors of chromatin regulators. As the Reviewer points out, there was little evidence in the original manuscript to support the use of the selected drug. Therefore, we have clarified the rationale for selecting the drug. In line with a logical progression from siRNA screening, we re-selected existing inhibitors from among the chromatin regulator paralog pairs selected in siRNA screening. Drug-susceptibility screening using these inhibitors identified the CBP/p300 inhibitor A-485 as the most selective for SMARCB1-deficient cancers (**Supplementary Fig. 1m**).

We have now explained this more clearly in the revised manuscript (page 6-7, lines 195-199).

3. The authors state "CP-C27 selectively inhibited the HAT activity of both CBP and p300, but had no effect on other HATs". It's a bit confusing to show IC₅₀ of the drug in the referenced figure. The authors should directly show the HAT activity with or without CP-C27 treatment and label the IC₅₀ in the figure.

Response

We agree.

We have revised **Fig. 1i** and **Supplementary Fig. 1n** to show HAT activity in the presence/absence of CP-C27. We have also labeled the IC₅₀ in the figure.

4. In Fig 2i, there is no significant difference of SMARCB1 peaks between 786-O and NEPS cell lines. As NEPS is a SMARCB1- cell line, the peaks of SMARCB1

should be negative just like the RNA-seq peak for 786-O. Are the peaks noise? The authors need to clarify this.

Response

The same point was made by Reviewer 3, who pointed out that ChIP-seq tends to emit non-specific background signals, so it would be better to show the positive region.

Reviewer 3 stated that “The ChIP-seq snapshots shown appear to be quite noisy. It is common for ChIP-seq data to exhibit some unspecific background signals. To address this issue, it is advisable to include positive controls, such as snapshots of regions that are well-known to recruit SMARCB1, P300, or are associated with the histone modifications being tested in this study. Incorporating snapshots of these positive controls will help to validate the specificity of the observed signals and ensure the reliability of the results.”

Therefore, we added snapshots of CUT&RUN-seq or ChIP-seq signals at the ANKRD1 and CDKN1A regions as “positive regions” in which binding of SMARCB1 outside of the KREMEN2 locus has been reported (Cell. 2019 179(6):1342-1356.e23. <https://doi.org/10.1016/j.cell.2019.10.044>; Nat Genet. 2017 49(11):1613-1623. <https://doi.org/10.1038/ng.3958>). In addition to the original data derived from the JMU-RTK-2 +SMARCB1 and JMU-RTK-2 -SMARCB1 pairs, and the 786-O (SMARCB1+) and NEPS (SMARCB1-) pairs, reported in this study, we examined publicly available data related to the TTC1240 +SMARCB1 and TTC1240 -SMARCB1 pairs.

At the ANKRD1 locus, SMARCB1 co-localized with H3K27ac, particularly at regions upstream of the ANKRD1 transcription start site (TSS) in 786-O SMARCB1+ cells (**Supplementary Fig. 3g**). By contrast, SMARCB1 signals at these regions decreased markedly in NEPS SMARCB1- cells (**Supplementary Fig. 3g**). At the CDKN1A locus, SMARCB1 co-localized with H3K27ac around the TSS, and across the gene body regions, in 786-O SMARCB1+ cells (**Supplementary Fig. 3h**). By contrast, SMARCB1 signals decreased in NEPS SMARCB1- cells (**Supplementary Fig. 3h**).

In SMARCB1 isogenic models on a JMU-RTK-2 and TTC1240 background,

SMARCB1 signals around TSS regions were detected in +SMARCB1 cells, but they were markedly weaker in -SMARCB1 cells (**Supplementary Fig. 3c–f**). Localization of CUT&RUN-seq and ChIP-seq signals at the KREMEN2 gene regions of SMARCB1 reported in this study was reconfirmed by published data for TTC1240. Localization of SMARCB1 to KREMEN2 regions was also confirmed by CUT&RUN-qPCR (**Fig. 3b**). It can be argued therefore that the data regarding localization to KREMEN2 are reliable.

However, this does not mean that SMARCB1 signals in SMARCB1- cells disappear completely. By incorporating the positive control snapshots, we found that the SMARCB1 CUT&RUN signals do generate some non-specific background noise (**Supplementary Fig. 3c,d,g,h**), as pointed out by Reviewer 3; this is even true for the published ChIP-seq data (**Supplementary Fig. 3e,f**). Therefore, we consider the SMARCB1 signals at the KREMEN2 locus in NEPS cells to be non-specific background signals.

We have now revised the manuscript text accordingly, and included additional data for the positive control snapshots (page12, lines 384-410).

5. The authors conducted diverse types of sequencing to indicate that “SMARCB1 plays a role in transcriptional suppression of the KREMEN2 gene by localizing directly at the promotor region”. However, the authors didn’t show direct evidence to confirm this conclusion. Similarly, the evidence of the transcriptional regulation of CBP/p300 on KREMEN2 expression are also weak.

For example, the authors should examine the pre-mRNA level of KREMEN2 after inhibiting SMARCB1 and CBP/p300 respectively and conduct ChIP-PCR or CUT&RUN-PCR to confirm the binding of SMARCB1 and CBP/p300 to the upstream of KREMEN2 gene.

Response

Thank you for pointing this out. We have now examined expression of pre-mRNA as well as mature mRNA. Expression of pre-mRNA was higher in JMU-RTK-2 - SMARCB1 than in JMU-RTK-2 +SMARCB1 (**Supplementary Fig. 2h**). In addition, expression of pre-mRNA in SMARCB1-deficient cell lines were higher than that in SMARCB1-proficient cell lines (**Supplementary Fig. 2i**). All of this provides further evidence that SMARCB1 suppresses KREMEN2 transcription via low

expression of pre-mRNA as well as mRNA.

On the other hand, treatment with CBP/p300 inhibitors A-485 and CP-C27 reduced expression of pre-mRNA in SMARCB1-deficient cell lines (**Supplementary Fig. 4c, d**). Therefore, CBP/p300 inhibition was thought to reduce mRNA with pre-mRNA attenuation.

In addition, we examined localization of SMARCB1 and CBP/p300 to the KREMEN2 gene regions using CUT&RUN-qPCR. First, the histone H3K4me3 (a marker of the promoter region) showed a localization peak near to the TSS (**Fig. 2h**). In addition, H3K4me1 (a marker of the enhancer region) localized to a region distant from the TSS (**Fig. 2i**). Furthermore, H3K27ac also localized to a region distant from the TSS (**Fig. 2j**). Thus, similar data were obtained by CUT&RUN-qPCR and CUT&RUN-seq. Furthermore, we confirmed that CBP and p300 localized upstream of the KREMEN2 gene region, including the vicinity of TSS (**Fig. 2l, m**). In addition, we confirmed that SMARCB1 localized to a region upstream of the KREMEN2 gene (**Fig. 3b**). These data confirm the results obtained by CUT&RUN-seq using CUT&RUN-qPCR.

We have now included these additional data in the revised manuscript and revised the text accordingly (page9, lines 284-293)(page14, lines 473-477) (page11, lines 342-355) (page12, lines 402-410).

6. Why did the authors focus on transcription factors BRD4 and CTCF? How were they selected? Do BRD4 and CTCF transcriptionally regulate KREMEN2? Any previous studies report this regulation? Comprehensive clarification needs to be done here.

Response

Localization of BRD4, a member of the bromo and extra-terminal (BET) family, to the KREMEN2 was investigated because BRD4 is a transcription factor that binds to acetylated histones, and may be involved in transcriptional regulation of the region localized by H3K27ac, which is targeted by CBP/p300 (Mol Cell. 2021 May 20;81(10):2166-2182.e6. doi: 10.1016/j.molcel.2021.03.008.). CTCF is a transcription factor that localizes upstream of its target gene loci (Cell. 2017 May 18;169(5):930-944.e22. doi: 10.1016/j.cell.2017.05.004.). In addition, CTCF colocalizes with the residual SWI/SNF complex in the absence of SMARCB1 (Nat Cell Biol. 2018 Dec;20(12):1410-1420. doi: 10.1038/s41556-018-0221-1.),

suggesting that it may be involved in transcription of KREMEN2, expression of which is promoted by SMARCB1 deficiency. Therefore, we asked whether BRD4 and CTCF localize to the KREMEN2 locus in SMARCB1-deficient cell lines; the results show that BRD4 and CTCF do localize to KREMEN2 regions (**Fig. 4g**). In addition, localization of BRD4 and CTCF was attenuated by inhibition of CBP/p300 (**Fig. 4g**). (page15, lines 496-508)

These reasons are now made clear in the revised manuscript (pages, lines).

7. In Fig 4C, why only upregulated genes were selected for pathway analysis as shown $FC > 2$? And why were apoptosis-related pathways picked up?

Response

In **Fig. 5c** (Fig. 4C in the original manuscript), both upregulated and downregulated genes were selected for pathway analysis. We intended that " $FC > 2$ " would include both upregulated and downregulated genes showing a change in expression $FC > 2$. We appreciate that this may have caused a misunderstanding. Therefore, " $FC > 2$ " has been changed to " $FC-UP > 2$ " and " $FC-DOWN > 2$ " in **Figure 5c**.

We also showed that inhibition or depletion of CBP/p300 induced apoptosis. To further investigate the relationship between CBP/p300 inhibition and apoptosis, we performed gene expression analysis using RNA-seq to isolate the molecular pathways involved in altered expression of gene clusters specifically in SMARCB1-deficient cells treated with a CBP/p300 dual inhibitor (**Fig. 5c**). We isolated 3135 genes whose expression was upregulated or downregulated specifically in JMU-RTK-2 -SMARCB1 cells, but not in JMU-RTK-2 +SMARCB1 cells, treated with a CBP/p300 dual inhibitor (**Fig. 5c**). We also isolated 1163 genes whose expression was upregulated or downregulated specifically in SMARCB1-deficient HS-ES-2R cells, but not in SMARCB1-proficient 786-O cells, treated with a CBP/p300 dual inhibitor (**Fig. 5c**). Then, we identified a set of 332 genes that showed overlap between these two gene sets (**Fig. 5c**). WikiPathway analysis identified 112 molecular pathways that were significantly associated with the 332 genes (**Fig. 5c**). To identify apoptotic markers that respond to treatment with CBP/p300 dual inhibitors, we focused on two apoptosis-related pathways: Hs-Apoptosis-WP254-106302 and Hs-Apoptosis-Modulation-and-Signaling-WP1772-107525.

The reason for focusing on apoptosis-related pathways was to identify apoptosis-related factors that fluctuate when CBP/p300 is suppressed.

We have revised the manuscript to clarify our reasons for examining apoptotic pathways (page17, lines 567-582).

8. Why did the authors focus on testing the impact of CBP/p300 dual inhibition on Wnt/ β -catenin and KREMEN1? It looks like the authors randomly pick up these two downstream mechanisms of KREMEN2 rather than evidence-based study.

Response

Only two KREMEN2 functions have been reported in the literature; one relates to the Wnt/ β -catenin pathway and the other to the apoptotic pathway. Therefore, we examined whether synthetic lethality due to suppression of KREMEN2 expression by CBP/p300 inhibition is associated with these two pathways.

We have now clarified the reasons for examining the connection between these two downstream pathways in the revised manuscript (lines 539-543).

9. The authors should show the impact of CBP/300 inhibition on KREMEN1 homodimerization and KREMEN1-KREMEN2 heterodimerization. This mechanism clarification was missing.

Response

In the original manuscript, we intended to propose that interaction between KREMEN2 and KREMEN1 regulates a signaling cascade that results in apoptosis. Thus, we investigated the effect of the KREMEN1 homo-interaction on downregulation of KREMEN2 or on CBP/p300 inhibition.

We have revised the manuscript text to include the following description of KREMEN1 and KREME2, their functions, and how they interact with each other, as well as other molecules. We also describe an assay system developed to examine the interaction between KREMEN1 molecules:

"KREMEN1 and KREMEN2 are single-transmembrane proteins; however, they have opposite effects on induction of apoptosis: KREMEN1 triggers apoptosis ⁴¹,

whereas KREMEN2 suppresses it ³⁴. Co-immunoprecipitation experiments suggest that KREMEN1 forms homodimers; however, formation of these homodimers is suppressed by overexpression of KREMEN2 ³⁴. There are no studies reporting quantitative analyses of the effect of KREMEN2 on homodimer formation by KREMEN1. Therefore, we investigated the relationship between KREMEN1 and KREMEN2 with respect to protein–protein interactions. To do this, we constructed an assay system based on quantitative measurement of homodimer formation by KREMEN1 using the NanoBiT system. This system can analyze protein–protein interactions, such as the KREMEN1-KREMEN1 homo-interaction, in real-time in living cells. It does this by detecting luminescence signals generated when the individual components of NanoLuc luciferase, i.e., the Small BiT (SmBiT) comprising 13 amino acid residues, and the Large BiT (LgBiT) comprising 156 amino acid residues, complement each other to form an active luciferase molecule ^{42,43}. We constructed vectors in which SmBiT and LgBiT were fused to the C-terminal region of KREMEN1. The NanoBiT system emits luminescence when proteins tagged with SmBiT and LgBiT are co-expressed in cells, and each protein binds in close proximity in the cell. It is possible to measure the amount of binding between proteins based on the amount of luminescence emitted at this time, and express it as an index. Therefore, we introduced KREMEN1 harboring a SmBiT tag and a LgBiT tag at the C-terminus into HEK293T cells and measured NanoBiT activity as a readout of the KREMEN1 homo-interaction (**Supplementary Fig. 5u**). No NanoBiT activity was observed when SmBiT alone or LgBiT alone was introduced into cells (**Supplementary Fig. 5v**). However, NanoBiT activity was observed when both KREMEN1-SmBiT and KREMEN1-LgBiT were introduced (**Supplementary Fig. 5v**). Thus, KREMEN1-SmBiT and KREMEN1-LgBiT appear to homo-interact inside the cell. Therefore, we used the SMARCB1-deficient cell line HS-ES-2R to establish a KREMEN1-SmBiT/KREMEN1-LgBiT stably transduced cell line, called HS-ES-2R +KREMEN1-NanoBiT (**Supplementary Fig. 5t**). We then used this cell line to investigate the effects of KREMEN2 expression on the KREMEN1 homo-interaction. Treatment with CBP/p300 inhibitors attenuated NanoBiT activity in the HS-ES-2R +KREMEN1-NanoBiT cell line (**Fig. 5o**). In addition, NanoBiT activity was reduced by knockdown of KREMEN2 (**Fig. 5p, Supplementary Fig. 5w**). By contrast, NanoBiT activity was increased by overexpression of KREMEN2 (**Fig. 5q, Supplementary Fig. 5w**). These results indicate that KREMEN2 increases the homo-interaction between KREMEN1 monomers. Thus, decreased expression of

KREMEN2 via CBP/p300 inhibition promotes monomerization of KREMEN1, followed by induction of apoptosis.” (page19, lines 652-691).

Thus, we used the NanoBit system to measure the KREMEN1 homo-interaction, and found that this interaction is promoted by KREMEN2. Thus, a reduction in expression of KREMEN2 in response to CBP/p300 inhibition triggers apoptosis through monomerization of KREMEN1.

We have also added the following text (explaining this further) to the revised manuscript in discussion section:

“Previously, co-immunoprecipitation experiments suggest that homodimer formation of KREMEN1 is suppressed by KREMEN2³⁴. In this study, the NanoBit system used to quantify the homo-interaction between KREMEN1 monomers revealed that KREMEN2 is required for homodimerization of KREMEN1. Thus, reduced expression of KREMEN2 due to inhibition of CBP/p300 triggers apoptosis through monomerization of KREMEN1. These observations are supported by previous studies showing that the N-terminal region of KREMEN1 (i.e., the ECD: Extra Cellular Domain) is required for homodimer formation⁴¹. Also, overexpression of KREMEN1-WT induces apoptosis, but overexpression of KREMEN1- Δ ECDmut induces apoptosis even more strongly⁴¹, indicating that apoptosis is more likely to occur under conditions that prevent homodimerization of KREMEN1. On the other hand, as for KREMEN1, the N-terminal ECD domain of KREMEN2 is required for hetero-interaction with KREMEN1³⁴. Overexpression of KREMEN1-WT alone induces apoptosis, but simultaneous overexpression of KREMEN1-WT and KREMEN2-WT suppresses induction of apoptosis³⁴. However, simultaneous overexpression of KREMEN1-WT and KREMEN2- Δ ECDmut (a KREMEN2 mutant that cannot bind to KREMEN1) does not suppress apoptosis³⁴, indicating that apoptosis is more likely to occur under conditions in which KREMEN1 cannot hetero-interact with KREMEN2 than under conditions in which KREMEN1 and KREMEN2 form heterodimers. In addition, KREMEN1 can also form trimers, which may include proteins other than KREMEN1³⁴. Based on these phenomena, we propose the following model for the relationship between KREMEN2 expression and KREMEN1 homo-interactions during apoptosis: when expression of KREMEN2 is high, the homodimer formation by KREMEN1 increases and then induction of apoptosis is suppressed by binding of the multimer to

KREMEN2. However, when expression of KREMEN2 is downregulated, the amount of KREMEN1 homodimers decreases (i.e., the amount of KREMEN1 monomers increases), followed by induction of apoptosis. Thus, in SMARCB1-deficient cells, apoptosis is suppressed by high expression of KREMEN2. However, apoptosis can be induced by increasing KREMEN1 monomerization by downregulating KREMEN2 using CBP/p300 inhibitors." (page26-27, lines 900-930).

10. The authors state "we identified several common molecular pathways associated with suppression of CBP/p300 or KREMEN2 in SMARCB1-deficient cells". It is a very vague statement. What are "common molecular pathways"? And why the authors focus on cell proliferation-related pathways involved in phosphorylation and signal transduction? This is conflicted with previous conclusion that simultaneous inhibition of CBP/p300 in SMARCB1-deficient cells induces apoptosis.

Response

What are "common molecular pathways"?

Figure 5A in the original manuscript refers to molecular pathways in which expression of a common group of genes fluctuates due to CBP/p300 inhibition, CBP/p300 knockdown, and KREMEN2 knockdown.

Why the authors focus on cell proliferation-related pathways involved in phosphorylation and signal transduction?

We agree that this conflicts with the previous conclusion that simultaneous inhibition of CBP/p300 in SMARCB1-deficient cells induces apoptosis.

Therefore, we used Gene Set Enrichment Analysis (GSEA) instead of the Wikipathway analysis (depicted in the original Figure 5A). As shown in the revised version (now **Fig. 6**), and as described below, we focused on the TNF α /NF- κ B and IL-6/JAK2/STAT3 signaling pathways (which are involved in apoptosis) as molecular pathways associated with induction of apoptosis in SMARCB1-deficient cells treated with a CBP/p300 inhibitor.

The manuscript text has been revised as follows to explain these points:

"Here, we found that downregulation of KREMEN2 by CBP/p300 in SMARCB1-deficient cells triggered apoptotic cell death via KREMEN1. A previous report shows that KREMEN1 is involved in induction of apoptosis³⁴; however, the molecular pathways downstream of KREMEN1 that trigger apoptosis are still unclear. Therefore, we performed gene expression analysis in two SMARCB1-deficient cell lines (JMU-RTK-2 and HS-ES-2R) using RNA-seq after treatment with the CBP/p300 inhibitor A-485, after CBP/p300 depletion, and after KREMEN2 depletion. To better understand the molecular pathways impacted by inhibition of CBP/p300, depletion of CBP/p300, and depletion of KREMEN2, we carried out Gene Set Enrichment Analysis (GSEA)^{44,45}, and also made use of the Molecular Signatures Database Hallmark Gene Set collection, each of which can be used to identify a specific biological state or process and to identify genes involved in these signatures⁴⁶. We then isolated significantly enriched signatures (i.e., $p < 0.05$, $q < 0.25$) among downregulated genes affected by treatment with A-485 (**Fig. 6a**), CBP/p300 knockdown (**Fig. 6b**), and KREMEN2 knockdown (**Fig. 6c**). On the other hand, there were no significantly enriched signatures among upregulated genes. Notably, we identified overlapping five downregulated gene signatures that correlated negatively with CBP/p300 inhibitor treatment, depletion of CBP/p300, and depletion of KREMEN2 (**Fig. 6d**). Of the five, we focused on two: TNFA-SIGNALING-VIA-NFKB and IL6-JAK-STAT3-SIGNALING (**Supplementary Fig. 6a–c**). This is because these signatures are associated with regulation of apoptosis^{47,48}. Inhibition of the TNF α (tumor necrosis factor- α)/NF- κ B (nuclear factor-kappa B) or IL-6 (Interleukin 6)/JAK2 (Janus Kinase 2)/STAT3 (signal transducer and activator of transcription 3) signaling pathways induced apoptosis^{47,48}, suggesting that downregulation of KREMEN2 by CBP/p300 inhibition triggers apoptosis by suppressing the TNF α /NF- κ B and IL-6/JAK2/STAT3 signaling pathways." (pages 20-21, lines 695-721).

11. The rationale to conduct protein microarrays to detect phosphorylated proteins is weak. And the reason for choosing AKT is also missing.

Response

Related to comment 10 above, we focused on TNF α /NF- κ B and IL-6/JAK2/STAT3 signaling pathways as molecular pathways that induce apoptosis in SMARCB1-deficient cells upon simultaneous inhibition of CBP/p300. In addition, we identified the genes within TNF α /NF- κ B and IL-6/JAK2/STAT3 signaling pathways

that are associated with apoptosis induction. Notably, core enrichment genes, which were the subset of genes that contributes most to the enrichment result, in each signature from TNFA-SIGNALING-VIA-NF-KB or IL6-JAK-STAT3-SIGNALING, overlapped with 11 genes and 3 genes, respectively, in these pathways (**Fig. 6e**), indicating that multiple genes in the gene sets of the TNF α /NF-kB or IL-6/JAK2/STAT3 signaling pathways are associated with apoptosis via CBP/p300-KREMEN2-KREMEN1 axis. We also found that each of the gene sets overlapped with two genes, CSF1 (colony-stimulating factor-1) and SOCS3 (suppressors of cytokine signaling 3) (Fig. 6e). Importantly, suppression of CSF1 or SOCS3 is involved in induction of apoptosis⁴⁹⁻⁵¹. CSF1 is a cytokine and binds to its receptor CSF1R (colony-stimulating factor-1 receptor), which is a receptor tyrosine kinase, and then induces tyrosine phosphorylation CSF1R, leading to activation of RAS (rat sarcoma)-ERK (extracellular signal-regulated kinase), PI3K (phosphatidylinositol 3-kinase)-AKT (protein kinase B), and JAK2-STAT3 phosphorylation signaling⁵²⁻⁵⁴. By contrast, suppression of SOCS3 induces hyper activation of STAT3 phosphorylation signaling and reduces activation of PI3K-ATK phosphorylation signaling⁵⁰. Thus, downregulation of KREMEN2 upon inhibition of CBP/p300 in SMARCB1-deficient cells could impact phosphorylation signaling pathways such as RAS-ERK, PI3K-AKT or JAK2-STAT3. Therefore, to investigate the phosphorylation signaling pathways affected by CBP/p300 inhibition, we performed phospho-protein microarray analysis to identify phosphorylation proteins affected by CBP/p300 inhibition in SMARCB1-deficient cells.

In addition, we found that treatment of JMU-RTK-2 -SMARCB1 cells with a CBP/p300 inhibitor markedly attenuated phosphorylation of AKT and its downstream protein PRAS40 (proline-rich AKT substrate of 40 kDa), but did not affect that of other TNF α /NF-kB or IL-6/JAK2/STAT3 signaling-related proteins such as STATs and ERKs (**Fig. 6f, g**). Therefore, we focused on the PI3K-AKT phosphorylation signaling pathway as a downstream signaling pathway affected by treatment with CBP/p300 inhibitors.

We have now included all of the above information in the revised version of the manuscript (page21, lines 722-750).

12. The authors state "Next, we investigated whether suppressing the AKT pathway by inhibiting CBP/p300 in SMARCB1-deficient cells affects the KREMEN2-KREMEN1 axis". AKT is the downstream of KREMEN2-KREMEN1 axis

according to the results. Did the authors intend to say whether suppressing the AKT pathway by inhibiting CBP/p300 in SMARCB1-deficient cells is mediated by the KREMEN2-KREMEN1 axis?

Response

Yes, we intended to ask whether suppression of the AKT pathway via inhibition of CBP/p300 in SMARCB1-deficient cells is mediated by the KREMEN2-KREMEN1 axis.

Therefore, we have revised the manuscript text to explain the findings of Figure 6 (i.e., that downregulation of KREMEN2 in SMARCB1-deficient cells upon CBP/p300 inhibition induces apoptosis through KREMEN1) in detail. Based on this, Next, we investigated whether suppression of the AKT signaling pathway by CBP/p300 inhibition is mediated by KREMEN1 (page22, lines 766-767).

13. In Fig 5H&I, KREMEN2 overexpression should increase AKT phosphorylation as it suppresses the KREMEN1 homodimer. However, the results showed here are opposite. Moreover, in Fig 5I, KREMEN2 overexpression increased the AKT protein level, thus the increased AKT pS473 may be due to the increased the AKT protein rather than enhanced phosphorylation.

Response

As the Reviewer points out, overexpression of KREMEN2 should increase AKT phosphorylation if downregulation of KERMEN2 or CBP/p300 inhibition reduces AKT phosphorylation through de-suppression KREMEN1. Thus, we conducted repeat experiments to examine whether KREMEN2 overexpression increases AKT phosphorylation. However, overexpression of KREMEN2 attenuated AKT phosphorylation rather than increasing it (see below).

On the other hand, KREMEN1 knockdown should increase AKT phosphorylation if downregulation of KERMEN2 or CBP/p300 inhibition reduces AKT phosphorylation through de-suppression of KREMEN1. Therefore, we confirmed

that KREMEN1 knockdown recovered (partially) AKT phosphorylation (see **Fig. 6k** and **Fig. 6l**).

Although it is not clear why results for KREMEN2 overexpression are opposite to those expected, the results regarding KREMEN1 knockdown are more consistent. Therefore, we have removed data related to AKT phosphorylation upon overexpression of KREMEN2.

Instead, we investigated whether KREMEN1 knockdown prevents the reduction of AKT phosphorylation mediated by KREMEN2 knockdown. As observed for CBP/p300 inhibition (**Fig. 6k**), KREMEN2 knockdown reduced AKT phosphorylation; however, the reduction in AKT phosphorylation was ameliorated partially by knockdown of KREMEN1 (**Fig. 6l**). These results indicate that downregulation of KREMEN2 by inhibition of CBP/p300 suppresses the PI3K-AKT signaling pathway mediated by KREMEN1, followed by induction of apoptosis.

We have now explained this in the text of the revised manuscript (page22, lines 766-773).

14. Throughout the paper, statistical significance was not shown in a number of bar charts especially in supplementary figures.

Response

Thank you. We have indicated statistical significance in the bar charts (except for Figures showing knockdown efficiency).

15. The authors should show protein levels as the knockdown efficiency rather than RNA levels throughout the paper, such as CBP, p300, KREMEN1, KREMEN2. Moreover, is the SFig 3K is the overexpression efficiency for SFig 3L-N? If so, the authors should show the protein efficiency for each figure as L, M, and N are different experiments, one efficiency cannot represent all.

Response

Thank you. We have now expressed CBP and p300 protein levels in terms of knockdown efficiency in **Supplementary Fig. 1d, g, j**.

In original manuscript, we tried to detect expression of intracellular KREMEN2.

We have expanded and explained this in more detail in the revised version of the manuscript as follows:

“We were unable to detect intracellular KREMEN2 protein using a commercially available anti-KREMEN2 antibody (**Supplementary Fig. 2j, k, l**). Transfection of KREMEN2 siRNA into SMARCB1-deficient cells reduced expression of KREMEN2 mRNA, as detected by qPCR (**Supplementary Fig. 2j**). The commercially available anti-KREMEN2 antibody did bind to several protein bands on western blots, but the intensity of these bands was not reduced after transfection of KREMEN2 siRNA (**Supplementary Fig. 2k**). However, the KREMEN2 antibody did detect ectopically overexpressed KREMEN2 because the intensity of a band representing overexpressed KREMEN2 was reduced after transfection of KREMEN2 siRNA (**Supplementary Fig. 2l**). Although expression of KREMEN2 mRNA in SMARCB1-deficient cells was higher than that in SMARCB1-proficient cells, expression of endogenous KREMEN2 proteins was not detected by commercially available antibodies. However, expression of overexpressed KREMEN2 proteins was detected by introducing cDNA encoding exogenous KREMEN2. Since this overexpressed KREMEN2 protein was knocked down by siKREMEN2, we believe that the antibody does recognize KREMEN2. Therefore, although the currently available antibodies were not able to detect expression of endogenous (intracellular) KREMEN2 protein, the lethality induced by KREMEN2 knockdown can be avoided by overexpression of KREMEN2, as shown in Figure 2d. Thus, we consider that endogenous KREMEN2 proteins (although they may be expressed at low levels) are functional. Therefore, CBP/p300 inhibitors may attenuate expression of KREMEN2 protein and induce cell death.” (page9, lines 294-315).

In the revised manuscript, we also tried to detect intracellular expression of KREMEN1.

We have explained the process in detail as follows:

“Next, we tried to examine expression of KREMEN1 protein. Although we could detect KREMEN1 protein expressed via the overexpression vector, we could not detect endogenous intracellular KREMEN1 protein using a commercially available anti-KREMEN1 antibody (**Supplementary Fig. 5r-t**) (as was the case for KREMEN2) (**Supplementary Fig. 2j-l**). We confirmed knockdown of KREMEN1

mRNA by KREMEN1 siRNA (**Supplementary Fig. 5r**); however, western blot analysis using commercially available KREMEN1 antibodies did not detect KREMEN1 proteins (as was also the case for KREMEN2). Many bands were detected by the anti-KREMEN1 antibodies, but none showed reduced intensity or disappeared after treatment with KREMEN1 siRNA, even after long exposure to chemical luminescence agents (**Supplementary Fig. 5s**). In addition, in cells transduced with an KREMEN1 overexpression vector, we detected bands of KREMEN1 that disappeared upon treatment with KREMEN1 siRNA (**Supplementary Fig. 5t**). The current commercially available KREMEN1 antibody detected KREMEN1 when expressed at high levels, but not at basal levels. This may have something to do with the specificity/affinity of the KREMEN1 antibody. However, we consider that endogenous KREMEN1 proteins are expressed and are functional in the cells." (page19, lines 634-651).

In **Supplementary Fig. 4o-q** (Original Fig. 3L-N), we used the five clone cells (#1-5) of SMARCB1-deficient HS-ES-2R cells stably introduced KREMEN2 overexpression vectors. Thus, we showed the protein efficiency for Fig. 4o-q in **Fig. 4n.**" (page16, lines 530-532).

16. The manuscript and figure legends were not well written, which affects the logic and flow of the content and is hard to follow. For example, "Furthermore, synthetic lethality induced by CBP/p300 inhibition in SMARCB1-deficient cells via de-suppression of KREMEN1 and subsequent downregulation of KREMEN2 meant that synthetic lethality could be avoided via suppression of KREMEN1 through overexpression of KREMEN2 (Supplementary Fig. S4Y)". The shortening sentence would be "Synthetic lethality ... meant that synthetic lethality could..." what's the meaning of this sentence?

Moreover, specific dataset IDs need to be clearly written including previous studies and this studies such as the legend for SFig 2F "RNA microarray data were obtained from the Gene Expression Omnibus database repository (GEO) used in a previous study". The authors should carefully revise the manuscript and probably rewrite the manuscript to make it clear, no bias, easy to follow and comprehensive.

Response

"Furthermore, synthetic lethality induced by CBP/p300 inhibition in SMARCB1-

deficient cells via de-suppression of KREMEN1 and subsequent downregulation of KREMEN2 meant that synthetic lethality could be avoided via suppression of KREMEN1 through overexpression of KREMEN2 (Supplementary Fig. S4Y)".

We apologize that this sentence is confusing. We intended to state the following:

Overexpression of KREMEN2 suppressed KREMEN1-mediated apoptosis. Thus, downregulation of KREMEN2 in SMARCB1-deficient cells by CBP/p300 inhibition induces apoptosis via de-suppression of KREMEN1.

As requested, we have tried to revise the manuscript to make it more logical and easier to follow.

Minor Concerns:

17-1. Please check that all abbreviations are spelled out the first time they are used, such as SMARCB1 and EZH2 in page 4.

Response

Thank you, we have checked this as requested.

18-2. Please check that references are properly cited throughout the manuscript. For example, the reference for "Paralog proteins are two very similar proteins that in general have a redundant function" in page 5 is missing.

Response

We have checked that references are properly cited throughout the manuscript, and we have included additional references where appropriate.

Nat Rev Genet. 2008 Dec;9(12):938-50. doi: 10.1038/nrg2482.

Cell. 2009 Jul 10;138(1):198-208. doi: 10.1016/j.cell.2009.04.029.

19-3. The meaning of Fig 1C didn't match with the description in the text in page 6.

Response

Thank you. We have revised the manuscript text accordingly (pages 5-6, lines 159-163).

20-4. What's the difference between Fig 1K and the first two bars in Fig 1J and similarly between Fig 1L and the third and fourth bars in Fig 1J. Are they same experiment?

Response

We apologize for the poor explanation. In the original manuscript, Fig. 1J, 1K, and 1L (**Fig 1j, k, l**) refer to different experiments. Fig. 1J (**Fig. 1j**) shows drug sensitivity data for JMU-RTK-2 +SMARCB1 and JMU-RTK-2 -SMARCB1 cells, whereas Fig. 1K (**Fig. 1k**) and Fig. 1L (**Fig. 1l**) show drug sensitivity data for CP-C27 and A-485, respectively, in ten SMARCB1-proficient cell lines and eight of SMARCB1-deficient cell lines (corresponding data are shown in **Supplementary Fig. 1o, 1p**).

We have revised the manuscript text to try to make everything as clear as possible (page 7, lines 215-218, 223-227).

21-5. "we isolated the gene cluster whose expression fell specifically in SMARCB1-deficient cells, but not SMARCB1-deficient cells, after treatment with a CBP/p300 inhibitor". It should be "not SMARCB1-proficient cells".

Response

We have corrected "not SMARCB1-deficient cells" to "not SMARCB1-proficient cells".

22-6. "ND" needs to be spelled out in the legend. Please check abbreviations in all figure legends (e.g. Fig. S2F).

Response

We have checked the abbreviations in all figure legends and spelled them out accordingly.

23-7. In Fig. S4B, actin is not even. Better quality figure should be provided.

Response

We repeated the western blotting analyses shown in Fig. S4B. The new data are presented in **Supplementary Fig. 5a**.

24-8. In Fig. S6J, EP300 mRNA was also reduced by 20-30% in shNT groups. Is it significant? Protein levels are needed.

Response

We repeated the qPCR analysis and western blot analyses shown in Fig. S6J. We confirmed that treatment with doxycycline did not affect expression of EP300 mRNA (**Supplementary Fig. 7j**) or p300 protein (**Supplementary Fig. 7k**) in shNT-treated cells.

Reviewer #2 (Remarks to the Author):

This manuscript describes targeting paralog proteins CBP/p300 for synthetic lethality in SMARCB1-deficient cancers. The author identified the gene KREMEN2 which is differentially upregulated in SMARCB1-deficient cancers but decreased expression upon treating a CBP/p300 HAT inhibitor A-485. The chromatin state of both enhancer and promoter KREMEN2 is affected by the SMARCB1 expression level and inhibition of CBP/p300. The author also investigated the mechanism of synthetic lethality by KREMEN2 by showing that inhibition of KREMEN2 increases Caspase 6 and Caspase 9 by de-suppression of KREMEN1 (although I have some confusion on this point described in detail below) and the PI3K/AKT pathway is identified. However, in contrast to the known function of KREMEN2, which inhibits canonical Wnt signaling, decreased KREMEN2 by inhibiting CBP/p300 in SMARCB1-deficient cancer function independent from the Wnt pathway. Lastly, the simultaneous inhibition of CBP/p300 significantly inhibited the SMARCB1-deficient cancer growth in vivo, concluding that CBP/p300 is the effective target of synthetic lethality of SMARCB1-deficient cancer. The majority of the work described simultaneous inhibition of a paralog pair method. Since most inhibitors in this study inhibit the paralog pair, the author suggests that simultaneous inhibition of a paralog pair method may be more effective than the traditional synthetic lethality, which inhibits a single target allowing redundancy to prevail.

Overall, I have great enthusiasm for this study. While it could use additional editing for clarity, it is overall well-written and conveys a very difficult topic well. The authors take an innovative approach and consider the often-overlooked role of SWI/SNF in transcriptional repression (many consider only transcriptional activation). Below I suggest some specific areas where clarity is required. I may have gotten confused in the double-negatives (which might still suggest that many readers would appreciate greater clarity), but there are a few spots where the authors may have confused themselves while writing. I also propose some experiments that would greatly strengthen and enhance the proposed model mechanistically. However, I give discretion to the editor and the authors as to the necessity of these experiments for publication in Nature Communications.

Response

Thank you for your very detailed review. I apologize for the lack of clarity in some

sections. We have revised the text, and have tried to explain the purpose and rationale more logically. In addition, we have conducted additional experiments based on your suggestions, which we believe clarify the mechanism underlying transcriptional repression not only by SMARCB1 alone but also as part of the SWI/SNF complex. As a result, I believe that the paper is much more persuasive. Thank you.

Major points for consideration: Not all of these need to be addressed by additional experiments. Some can be addressed in the writing or figures but are still major needs of clarification.

(1) In figure 1, the author mentioned that CP-C27 was generated as an optimized A-485 analog, selectively inhibiting the HAT activity of CBP/p300 but having no effect on other HATs. Therefore, Figure 3E would have been stronger with CP-C27 instead of A-485.

Response

We investigated whether treatment with CP-C27 and A-485 affected the localization of histone markers and transcription factors to the KREMEN2 loci. CP-C27, as with A-485, reduced ATAC-seq and RNA-seq signals (**Fig. 4e**) and attenuated localization of H3K4me3, H3K4me1, H3K27ac, RNAPII, BRD4, and CTCF at the KREMEN2 loci (**Fig. 4g**). The additional data have been added to the revised manuscript (page 15, lines 486-508).

(2) Since SMARCB1 is deficient in the cancer JMU-RTK-2 line, mapping of SMARCB1 ChIP alone cannot show the SWI/SNF's genome localization in SMARCB1-deficient cells. Therefore, investigating SWI/SNF distribution on genome targeting another subunit, such as Brg1 is necessary to compare how the complex distribution differs between SMARCB1-deficient and SMARCB1-rescued JMU-RTK-2 lines. Figure S3P of Mashlatir et al., 2018, shows that BAF and PBAF complexes are largely left intact without SMARCB1. This suggests two possible models in your system (1) There are residual complexes at KREMEN2 in SMARCB1- cells that are activating transcription, or (2) there are no SWI/SNF complexes at the locus at all, and WT SWI/SNF only acts to repress KREMEN2 transcription in a SMARCB1+ context? To answer this, one could start with a Brg1 or Baf155 ChIP (or another shared subunit), and if there are residual SWI/SNF

complexes present in SMARCB1- cells, then one could use unique subunits (PHF10, ARID1s, etc.) to determine cBAF versus PBAF. If after these experiments are done and the model is one of repression in the SMARCB1+ WT context, what subcomplex is doing the repression? Is it cBAF or PBAF? One might think it is more likely to be PBAF based on other findings, but the authors have the ability to determine definitively which it is in this case, and it would greatly strengthen their model and this paper to do so. (Figure 2H-I, and Figure S2J)

Response

To investigate transcriptional regulation of the KREMEN2 gene region of the SWI/SNF complex containing SMARCB1, we examined localization of SWI/SNF factors using CUT&RUN-qPCR.

This is now explained in detail in the revised manuscript, along with the inclusion of the new CUT&RUN data, as follows:

“To further confirm localization of histone markers and CBP/p300 at the KREMEN2 locus, we conducted CUT&RUN-qPCR. H3K4me3, a marker of transcriptional promotion, was localized in the vicinity of the TSS in JMU-RTK-2 -SMARCB1 cells, but not in JMU-RTK-2 +SMARCB1 cells (**Fig. 2h**). H3K4me1, a marker of transcriptional enhancement, was localized in a region distant from the TSS in JMU-RTK-2 -SMARCB1 cells, but not in JMU-RTK-2 +SMARCB1 cells (**Fig. 2i**). H3K27ac, a marker of both the promoter and enhancer regions, was localized across the regions also localized by H3K4me3 and H3K4me1 in JMU-RTK-2 -SMARCB1 cells, but not in JMU-RTK-2 +SMARCB1 cells (**Fig. 2j**). In contrast to H3K27ac, H3K27me3 (a marker of transcriptional repression) was localized in the region intermediate between the promoter and enhancer regions in JMU-RTK-2 +SMARCB1 cells, but not in JMU-RTK-2 -SMARCB1 cells (**Fig. 2k**). Moreover, in JMU-RTK-2 -SMARCB1 cells, the acetyltransferases CBP and p300 of H3K27ac were also localized across regions localized by H3K27ac (**Fig. 2l, m**). However, in JMU-RTK-2 +SMARCB1 cells, the methyltransferase EZH2, a component of PRC2 methyltransferase complex, of H3K27me3 was also localized across regions localized by H3K27me3 (**Fig. 2n**). This result also confirmed published ChIP-seq data (GSE90634) derived from the TTC1240 SMARCB1 isogenic model showing that H3K27me3 co-localized with SUZ12, another component of the PRC2 complex (**Supplementary Fig. 2n**). Thus, CUT&RUN-qPCR yielded results similar

to those of CUT&RUN-seq and ChIP-seq.” (page11, lines 342-362).

We have also revised the manuscript text as follows to include new CUT&RUN data regarding transcriptional regulation of the KREMEN2 gene region of the SWI/SNF complex containing SMARCB1:

“To further investigate transcriptional regulation of the KREMEN2 gene region of the SWI/SNF complex containing SMARCB1, we examined localization of SWI/SNF factors using CUT&RUN-qPCR. We confirmed that SMARCB1 binds to the region between the promoter region and the enhancer region in JMU-RTK-2 +SMARCB1 cells (**Fig. 3b**), as did H3K27me3 (**Fig. 2k**) and EZH2 (**Fig. 2n**); however, localization was absent in JMU-RTK-2 -SMARCB1 cells, indicating that SMARCB1 localizes directly at the promoter region of the KREMEN2 locus and plays a role in transcriptional repression of the KREMEN2 gene. Therefore, upregulation of KREMEN2 gene expression due to a deficiency of SMARCB1 may underlie the KREMEN2-dependency of SMARCB1-deficient cells.

The SWI/SNF chromatin remodeling complex comprises BAF, PBAF, and ncBAF complexes ⁶. To investigate transcriptional regulation of the KREMEN2 gene by the SWI/SNF complex containing SMARCB1, we examined localization of SWI/SNF factors using CUT&RUN-qPCR. The ATPase factor SMARCA4 of the SWI/SNF complex localized at regions upstream of the KREMEN2 locus in JMU-RTK-2 +SMARCB1 cells (**Fig. 3c**). However, SMARCA4 was still present in JMU-RTK-2 -SMARCB1 cells, and was newly localized to the TSS (**Fig. 3c**). In addition, the constituents of SWI/SNF subtype complexes cBAF (ARID1A) and PBAF (PBRM1), which include SMARCB1, localized at regions upstream of the KREMEN2 locus in JMU-RTK-2 +SMARCB1 cells (**Fig. 3d, e**). However, in JMU-RTK-2 -SMARCB1 cells, cBAF (ARID1A) and PBAF (PBRM1) shifted to sites proximal to the TSS (**Fig. 3d, e**). These results also confirmed the similar localization patterns of SMARCB1, SMARCA4, DPF2 (cBAF), and ARID2 (PBAF) in the published ChIP-seq data (GSE90634, GSE124903) derived from the TTC1240 SMARCB1 isogenic model (**Supplementary Fig. 3i**) ^{27,35}. In addition, SS18, a constituent of cBAF and ncBAF, also localized at the region upstream of the KREMEN2 locus in JMU-RTK-2 +SMARCB1 cells (**Fig. 3f**). In JMU-RTK-2 -SMARCB1 cells, SS18 shifted to sites proximal to the TSS (**Fig. 3f**), as in ARID1A (cBAF) (**Fig. 3d**). In SMARCB1-deficient cell lines, ncBAF complex is substantially more localized to promoter-proximal sites ^{27,36,37}. Therefore, we examined localization of the GLTSCR1, a constituent of

ncBAF. In JMU-RTK-2 +SMARCB1 cells, GLTSCR1 did not localize to the KREMEN2 locus, but it did in JMU-RTK-2 -SMARCB1 cells (**Fig. 3g**), which it was newly localized at sites proximal to the TSS. Thus, as observed for the SWI/SNF complex containing SMARCB1, cBAF and PBAF (as well as the transcriptional repressor EZH2) localized at regions upstream of the KREME2 locus, and are thus considered to repress transcription of KREMEN2.

It is reported that even in cells with SMARCB1 deficiency, cBAF and PBAF complexes have no effect on formation of these complex 6. Our data suggest that residual cBAF and PBAF deficient in SMARCB1 migrate to the TSS of the KREMEN2 gene (**Fig. 3c-f**). In addition, we thought that ncBAF would also be newly recruited to the KREMEN2 gene regions (**Fig. 3g**). Therefore, we examined whether residual cBAF and PBAF deficient in SMARCB1, or newly recruited ncBAFs, promote transcription of KREMEN2. Suppression of cBAF (ARID1A), PBAF (ARID2), and ncBAF (BRD9) did not attenuate expression of KREMEN2 (**Fig. 3h**). Simultaneous suppression of SMARCA4 and SMARCA2, both of which are essential for SWI/SNF function, did not reduce expression of KREMEN2 (**Fig. 3h**). Therefore, we assumed that none of the SMARCB1-deficient residual SWI/SNF complexes were involved in promoting expression of KREMEN2. Apart from the SWI/SNF complex, other chromatin remodeling complexes such as ISWI, CHD, and INO80 family complexes are classified as sub-complexes 24. Therefore, we investigated whether chromatin remodeling complexes other than SWI/SNF complexes are involved in promotion of KREMEN2 transcription. Suppression of SMARCA1, but not that of other complexes, attenuated expression of KREMEN2 (**Fig. 3i**). In addition, we confirmed that depletion of SMARCA1 from SMARCB1-deficient cell lines (**Supplementary Fig. 3j, k**) reduced expression of KREMEN2 mRNA (**Fig. 3i**). Therefore, we considered that the SMARCA1 complex was involved in promotion of KREMEN2 transcription in SMARCB1-deficient cells. SMARCA1 is an ATPase and a subunit of the ISWI family complex, which is involved in transcription when recruited to target gene loci 24. In fact, SMARCA1 was localized widely in the region upstream of the KREMEN2 locus due to a lack of SMARCB1 (**Fig. 3k**). In addition, treatment of JMU-RTK-2 -SMARCB1 cells with the CBP/p300 inhibitor CP-C27 attenuated localization of H3K27ac across the upstream regions of the KREMEN2 locus (**Fig. 3m**), as well as localization of SMARCA1 around the TSS site (**Fig. 3l**). Therefore, in SMARCB1-deficient cells, the SMARCA1 complex may function as a chromatin remodeling complex involved in promoting transcription of KREMEN2." (pages12-14, lines 402-468).

(3) The first paragraph of page 11, "These results indicate that simultaneous inhibition of CBP and p300 in SMARCB1-deficient cells causes the synthetic lethality via induction of apoptosis" A knockdown of KREMEN2 would be more direct testing of the model proposed in this study than CBP/p300 inhibition here.

Response

We agree with the Reviewer. We have now included data on increased expression of CASP6 upon KREMEN2 knockdown or inhibition of CBP/p300 prior to the sentence "These results indicate that simultaneous inhibition of CBP and p300 in SMARCB1-deficient cells causes synthetic lethality via induction of apoptosis".

The text has been revised as follows:

"To confirm involvement of KREMEN2 in suppression of apoptosis, we investigated whether expression of proapoptotic marker gene is altered by knockdown of KREMEN2. The CASP6 gene was upregulated by knockdown of KREMEN2 in SMARCB1-deficient cells, but not in SMARCB1-proficient cells (**Supplementary Fig. 5h**), indicating that suppression of KREMEN2 in SMARCB1-deficient cells induces apoptosis. Taken together, these results indicate that inhibition of CBP/p300 in SMARCB1-deficient cells induces apoptosis by downregulating KREMEN2." (pages 17-18, lines 587-594).

(4) The authors state, "... then induction of synthetic lethality by inhibition of CBP/p300 could be avoided by suppression of KREMEN1." This sentence confuses me, and it may be because of a typo. Do the authors mean KREMEN2? Because wouldn't the opposite of the sentence as written be true for KREMEN1? KREMEN1 is proposed to be pro-apoptotic, so suppressing it would not be synthetic lethal. Do the authors mean activation or over-expression of KREMEN1? Side note: What will happen if KREMEN1 is overexpressed?

Response

A previous study shows that overexpression of KREMEN1 induces apoptosis (Cell Death Differ. 2016 Feb;23(2):323-32. doi: 10.1038/cdd.2015.100); therefore, the Reviewer is correct to point out that since KREMEN1 is a pro-apoptotic factor, it is thought that synthetic lethality will not occur when KREMEN1 is suppressed.

We apologize for the confusion caused by the sentence "... then induction of synthetic lethality by inhibition of CBP/p300 could be avoided by suppression of KREMEN1."

We have revised the text to better explain the relationship between KREMEN1 and KREMEN2 as follows:

"KREMEN2 and KREMEN1 are single-pass transmembrane proteins that interact with each other ³⁴. KREMEN1 is involved in apoptosis induction ⁴¹, whereas KREMEN2 suppresses KREMEN1-mediated apoptosis ³⁴. Therefore, we hypothesized that downregulating KREMEN2 would increase KREMEN1-mediated apoptosis, and then additional depletion of KREMEN1 would inhibit apoptosis." (page 18, lines 595-600).

(5) I am confused as to the model of KREMEN1. Is the model an increase in the transcription (and therefore protein levels of KREMEN1) in the absence of KREMEN2, or is it that existing KREMEN1 can homodimerize and lead to a signaling cascade that results in apoptosis? I thought it was the latter, but then the authors suggest the former on page 13. If it is an increase in homodimerization (an elegant model), can the authors show an increase in homodimerization when KREMEN2 is knocked down or in CBP/p300 inhibition? These data would greatly strengthen the mechanism of the proposed model.

Response

We apologize for the confusion. In the original manuscript, we wanted to say that interaction between KREMEN2 and KREMEN1 regulates a signaling cascade that results in apoptosis. Thus, we investigated the effects of the KREMEN1 homo-interaction on downregulation of KREMEN2 or on CBP/p300 inhibition.

We have revised the manuscript text to include the following description of KREMEN1 and KREME2, their functions, and how they interact with each other, as well as other molecules. We also describe an assay system developed to examine the interaction between KREMEN1 molecules:

“KREMEN1 and KREMEN2 are single-transmembrane proteins; however, they have opposite effects on induction of apoptosis: KREMEN1 triggers apoptosis ⁴¹, whereas KREMEN2 suppresses it ³⁴. Co-immunoprecipitation experiments suggest that KREMEN1 forms homodimers; however, formation of these homodimers is suppressed by overexpression of KREMEN2 ³⁴. There are no studies reporting quantitative analyses of the effect of KREMEN2 on homodimer formation by KREMEN1. Therefore, we investigated the relationship between KREMEN1 and KREMEN2 with respect to protein–protein interactions. To do this, we constructed an assay system based on quantitative measurement of homodimer formation by KREMEN1 using the NanoBiT system. This system can analyze protein–protein interactions, such as the KREMEN1-KREMEN1 homo-interaction, in real-time in living cells. It does this by detecting luminescence signals generated when the individual components of NanoLuc luciferase, i.e., the Small BiT (SmBiT) comprising 13 amino acid residues, and the Large BiT (LgBiT) comprising 156 amino acid residues, complement each other to form an active luciferase molecule ^{42,43}. We constructed vectors in which SmBiT and LgBiT were fused to the C-terminal region of KREMEN1. The NanoBiT system emits luminescence when proteins tagged with SmBiT and LgBiT are co-expressed in cells, and each protein binds in close proximity in the cell. It is possible to measure the amount of binding between proteins based on the amount of luminescence emitted at this time, and express it as an index.” (pages 19-20, lines 652-672).

We have also revised the text as follows to provide an explanation of how we developed an assay to examine the KREMEN1/KREMEN1 and KREMEN1/KREMEN2 interactions under different conditions; these new data are included in the revised manuscript:

“Therefore, we introduced KREMEN1 harboring a SmBiT tag and a LgBiT tag at the C-terminus into HEK293T cells and measured NanoBiT activity as a readout of the KREMEN1 homo-interaction (Supplementary Fig. 5u). No NanoBiT activity was observed when SmBiT alone or LgBiT alone was introduced into cells (Supplementary Fig. 5v). However, NanoBiT activity was observed when KREMEN1-SmBiT and KREMEN1-LgBiT were introduced (Supplementary Fig. 5v). Thus, KREMEN1-SmBiT and KREMEN1-LgBiT appear to homo-interact inside the cell. Therefore, we used the SMARCB1-deficient cell line HS-ES-2R to establish a KREMEN1-SmBiT/KREMEN1-LgBiT stably transduced cell line, called HS-ES-2R

+KREMEN1-NanoBiT (Supplementary Fig. 5t). We then used this cell line to investigate the effects of KREMEN2 expression on the KREMEN1 homo-interaction. Treatment with CBP/p300 inhibitors attenuated NanoBiT activity in the HS-ES-2R +KREMEN1-NanoBiT cell line (Fig. 5o). In addition, NanoBiT activity was reduced by knockdown of KREMEN2 (Fig. 5p, Supplementary Fig. 5w). By contrast, NanoBiT activity was increased by overexpression of KREMEN2 (Fig. 5q, Supplementary Fig. 5w). These results indicate that KREMEN2 increases the homo-interaction between KREMEN1 monomers. Thus, decreased expression of KREMEN2 via CBP/p300 inhibition promotes monomerization of KREMEN1, followed by induction of apoptosis.” (page 20, lines 672-691).

Thus, we used the NanoBit system to measure the KREMEN1 homo-interaction, and found that this interaction is promoted by KREMEN2. Thus, a reduction in expression of KREMEN2 in response to CBP/p300 inhibition triggers apoptosis through monomerization of KREMEN1.

We have also added the following text (explaining this further) to the revised manuscript in discussion section:

“Previously, co-immunoprecipitation experiments suggest that homodimer formation of KREMEN1 is suppressed by KREMEN2³⁴. In this study, the NanoBiT system used to quantify the homo-interaction between KREMEN1 monomers revealed that KREMEN2 is required for homodimerization of KREMEN1. Thus, reduced expression of KREMEN2 due to inhibition of CBP/p300 triggers apoptosis through monomerization of KREMEN1. These observations are supported by previous studies showing that the N-terminal region of KREMEN1 (i.e., the ECD: Extra Cellular Domain) is required for homodimer formation⁴¹. Also, overexpression of KREMEN1-WT induces apoptosis, but overexpression of KREMEN1- Δ ECDmut induces apoptosis even more strongly⁴¹, indicating that apoptosis is more likely to occur under conditions that prevent homodimerization of KREMEN1. However, as for KREMEN1, the N-terminal ECD domain of KREMEN2 is required for hetero-interaction with KREMEN1³⁴. Overexpression of KREMEN1-WT alone induces apoptosis, but simultaneous overexpression of KREMEN1-WT and KREMEN2-WT suppresses induction of apoptosis³⁴. However, simultaneous overexpression of KREMEN1-WT and KREMEN2- Δ ECDmut (a KREMEN2 mutant that cannot bind to KREMEN1) does not suppress apoptosis³⁴, indicating that

apoptosis is more likely to occur under conditions in which KREMEN1 cannot hetero-interact with KREMEN2 than under conditions in which KREMEN1 and KREMEN2 form heterodimers. In addition, KREMEN1 can also form trimers, which may include proteins other than KREMEN1³⁴. Based on these phenomena, we propose the following model for the relationship between KREMEN2 expression and KREMEN1 homo-interactions during apoptosis: when expression of KREMEN2 is high, the homodimer formation by KREMEN1 increases and then induction of apoptosis is suppressed by binding of the multimer to KREMEN2. However, when expression of KREMEN2 is downregulated, the amount of KREMEN1 homodimers decreases (i.e., the amount of KREMEN1 monomers increases), followed by induction of apoptosis. Thus, in SMARCB1-deficient cells, apoptosis is suppressed by high expression of KREMEN2. However, apoptosis can be induced by increasing KREMEN1 monomerization by downregulating KREMEN2 using CBP/p300 inhibitors.” (page 26-27, lines 900-930) .

(6) Comparing Figure 5J and 5K, the control lanes of siNT (KREMEN2 -) have very different levels of AKT pS473 making interpretation and comparison harder. In J, there is more AKT-pS473 in the KREMEN2 + condition than KREMEN2 – condition, but the opposite is true in Fig 5K. It is unclear to me why this would be. It seems the model figure in Figure S5F suggests that KREMEN2 overexpression increases the phosphorylation of AKT. What makes the expression of KREMEN2 different in Figures 5K, 5H, and 5I (lanes 1 and 3)? And how does the phosphorylation of PRAS40 change in both conditions of Figure 5J and 5K? There are a lot of differences in AKT pS473 in siNT conditions that make it really hard to interpret and make conclusions from. It would be appreciated if the cell line names were included in Figure 5G-K, similar to Figure 5D-F. Additionally Figure 5 would benefit from the cell lines being in the figure panels and not just the figure legend.

Response

As the Reviewer points out, overexpression of KREMEN2 should increase AKT phosphorylation if downregulation of KERMEN2 or CBP/p300 inhibition reduces AKT phosphorylation through de-suppression KREMEN1. Thus, we conducted repeat experiments to examine whether KREMEN2 overexpression increases AKT phosphorylation. However, overexpression of KREMEN2 attenuated AKT phosphorylation rather than increasing it (see below).

On the other hand, KREMEN1 knockdown should increase AKT phosphorylation if downregulation of KERMEN2 or CBP/p300 inhibition reduces AKT phosphorylation through de-suppression of KREMEN1. Therefore, we confirmed that KREMEN1 knockdown increased (partially) AKT phosphorylation (see **Fig. 6k** and **Fig. 6l**).

Although it is not clear why results for KREMEN2 overexpression are opposite to those expected, the results regarding KREMEN1 knockdown are more consistent. Therefore, we have removed data related to AKT phosphorylation upon overexpression of KREMEN2.

Instead, we investigated whether KREMEN1 knockdown prevents the reduction of AKT phosphorylation mediated by KREMEN2 knockdown. As observed for CBP/p300 inhibition (**Fig. 6k**), KREMEN2 knockdown reduced ATK and PRAS40 phosphorylation; however, the reduction in AKT and PRAS40 phosphorylation was ameliorated partially by knockdown of KREMEN1 (**Fig. 6l**).

These results indicate that downregulation of KREMEN2 by inhibition of CBP/p300 suppresses the PI3K-AKT signaling pathway mediated by KREMEN1, followed by induction of apoptosis..

We have now explained this in the text of the revised manuscript (pages 22-23, lines766-773).

In addition, cell line names were included in the figure panels and not just the figure legend in not only Figure 5 but also other Figures.

Minor points:

7-I think "de-repression" over "de-suppression" is the more accurate and widely-used term, but the authors are welcome to use the one they think is best.

Response

Thank you, we have made the change as suggested.

8-While described in the methods, it would be useful to briefly explain how SMARCB1 was reintroduced in the JMU-RTK-2 cells (lentivirus, right?) How does this expression compare to WT cells? While this is difficult to say due to imperfect direct comparisons, it is important to know if this brings SMARCB1 to roughly WT levels or is more akin to over-expression. (page 6)

Response

We established the cell line clone JMU-RTK-2 +SMARCB1 cells by stably transducing cells with a SMARCB1-expressing lentivirus vector.

We have now revised the text as follows to include a description of how we established the JMU-RTK-2 +SMARCB1 cell line:

“To establish JMU-RTK-2 +SMARCB1 cells, SMARCB1-deficient JMU-RTK-2 cells were transduced with lentiviruses derived from the pLOC-CMV-SMARCB1-Bsd lentivirus vector. After selection of blasticidin-resistant cells, a clone of JMU-RTK-2 cells expressing the SMARCB1 protein was isolated.” (page 30, lines 1018-1022).

“In addition, we observed that expression levels of SMARCB1 protein in JMU-RTK-2 +SMARCB1 cells were comparable with those in SMARCB1-proficient cell lines HEK293T and 786-O (**Supplementary Fig. 1a**). Therefore, we considered expression of SMARCB1 protein in the JMU-RTK-2 +SMARCB1 cells to be at the wild-type level rather than being overexpressed.” (page 5, lines 129-134).

9-Does Inobrodib inhibit other bromodomain-containing proteins (there are bromodomains in BAF complex subunits, for example), or does it only inhibit CBP/p300 (page 6)?

Response

Inobrodib is a specific inhibitor of both CBP and p300 among the 32 proteins with bromo domains (which include SMARCA4, SMARCA2, and PBRM1 with bromodomains in BAF complex subunits (Cancer Discov. 2021 May;11(5):1118-1137. doi: 10.1158/2159-8290.CD-20-0751.)). Therefore, it is considered that the synthetic lethality caused by inhibition of CBP/p300 in SMARCB1-deficient cells is specific to inhibition of CBP/p300, and not inhibition of SMARCA4/2.

We have revised the manuscript text to clarify this point (page 7, lines 200-207).

10-While the authors describe what the role of KREMEN2 protein is later in the manuscript, a brief description at the end of page 7 (after the text of "(Fig. 2A)", would be a good place).

Response

Thank you. We have made the change as suggested and placed the following text after our explanation of Figure 2a:

"The KREMEN2 gene, which is a single-pass transmembrane protein that plays dual roles in cells (suppression of the Wnt/ β catenin pathway and the apoptosis pathway)^{33,34}, was upregulated specifically in SMARCB1-deficient cells and downregulated specifically in SMARCB1-deficient cells treated with CBP/p300 inhibitors (**Fig. 2a**), indicating that KREMEN2 is a candidate gene that determines synthetic lethality." (page 8, lines 252-260).

11-Cancer treatment with synthetic lethality aims to selectively impose lethality on cancer cells without significantly affecting "non-cancer" cells. Therefore, I think you should move the HEK293T IC50 data from S1 to Figure 1J-L.

Response

As requested, the IC₅₀ data for HEK293T have been moved to **Fig. 1j**.

We have also added the following sentence:

"The IC₅₀ (50% inhibitory concentration) values derived from JMU-RTK-2 - SMARCB1 cells treated with CBP/p300-specific inhibitors CP-C27, A-485, and inobrodib were markedly lower than those from JMU-RTK-2 +SMARCB1 cells and HEK293T non-cancer cells (**Fig. 1j**)." (page 7, lines 215-218).

12-Page 8, the paragraph that begins with "To further investigate transcriptional regulation at the KREMEN2 gene locus, we observed the chromatin condition at the locus," I suggest replacing the phrase "observed the chromatin condition" with something else like "characterized the chromatin state." Observed and

condition aren't quite the right words for this sentence.

Response

As suggested, we replaced "observed the chromatin condition" with "characterized the chromatin state."

13-While I pointed out some specifics in lack of clarity on page 11, I want to reiterate that there are other points I may not have directly pointed out, so this part of the manuscript should be carefully read and edited to make the language as precise as possible.

Response

We have revised the text to try to prevent any misunderstanding. We have also checked through the manuscript and tried to make the language as precise as possible.

14-The basal level of beta-catenin is lower in SMARCB1-deficient cancer (Figure S4B). Is the Beta-catenin expression level also increased by overexpressing SMARCB1 (JMU-RTK +SMARCB1)?

Response

We investigated whether expression of beta-catenin increases after introduction of SMARCB1 into SMARCB1-deficient JMU-RTK-2 cells. Rescuing SMARCB1 expression in SMARCB1-deficient cells had little effect on β -catenin expression (**Supplementary Fig. 5b**).

We have included these data, and revised the manuscript text accordingly (page 16, lines 550-551).

15-For additional discussion: Is KREMEN2 a feasible drug target for synthetic lethality? As the authors suggest, this would reduce off-target inhibition effects of other CBP/p300 target genes during CBP+p300 inhibition.

Response

Thank you for pointing this out. We also think that KREMEN2 may be a feasible drug target for synthetic lethality in SMARCB1-deficient cancers. At the very least,

we showed that CBP/p300 inhibitors appear to have little effect on proliferation of normal cells, or on body weight in mice. However, there are concerns about side effects due to genome-wide effects (i.e., on expression of genes other than KREMEN2) due to inhibition of CBP/p300. Suppression of KREMEN2 also shows synthetic lethality in SMARCB1-deficient cells; thus KREMEN2 is a promising synthetic lethal target. Compared with CBP/p300 inhibition, KREMEN2 inhibition may be expected to have fewer side effects. Further use of KREMEN2-KREMEN1 interaction inhibitors to discover new drugs, and to develop KREMEN2 inhibitors, is expected.

We have made these points in the "Discussion" section of the revised manuscript (page 28, lines 950-958).

Reviewer #3 (Remarks to the Author):

The study conducted by Sasaki et al. presents a synthetic lethal screening of chromatin regulator paralogs for SMARCB1-deficient tumors. The researchers discovered that the dual inhibition of CBP/p300 is lethal in SMARCB1-deficient tumors and performed in-depth molecular analysis. They show an involvement of KREMEN2 and KREMEN1, as well as the activation of AKT.

Given the highly aggressive nature of SMARCB1-deficient tumors and the lack of effective therapeutics for patients with this condition, the exploration of novel therapeutic targets is of utmost importance. The discovery of the synthetic lethal capacity of CBP/p300 dual inhibition in SMARCB1-deficient tumors is, therefore, relevant. Moreover, the screening approach using simultaneous inhibition of chromatin regulator paralogs is a novel and innovative aspect of the study.

However, despite the potential clinical significance of the findings, the data and strength of the molecular studies presented in the paper are somewhat limited. The selection criteria for the targets chosen for further studies, namely KREMEN2, remain unclear, and many of the validations rely solely on RT-PCRs. This includes the effects of KREMEN2 on KREMEN1 homodimerization and promotion of apoptosis. Additionally, the use of western-blot analysis to detect endogenous protein levels, especially for KREMEN2, did not yield any conclusive results across the samples.

In summary, while the study provides valuable insights into promising therapeutic targets for SMARCB1-deficient tumors, the molecular studies conducted are not sufficiently robust, thus limiting the overall strength of the findings.

Response

Thank you for your very detailed review. We apologize for the lack of clarity in some areas. We have revised the manuscript to try to explain the purpose and rationale underlying our experiments. In particular, we tried to provide detailed explanations of how we selected and narrowed down our targets. In addition, we have conducted additional experiments and provided more detailed explanations about protein expression, interactions between KREMEN2 and KREMEN1, and revalidation of our data using published datasets; we believe that the paper is much more convincing as a result. Thank you.

Specific comments/concerns

1-In their screening for synthetic lethal interactions in SMARCB1-deficient cells, the authors selected candidates based on siRNAs that affect cell growth in cells lacking SMARCB1 compared to isogenic cells with restored SMARCB1. However, it's important to note that the restitution of SMARCB1 itself has a potent effect on cell growth inhibition. How did the authors account for the effect of SMARCB1 restitution in their analysis?

Response

As pointed out by the reviewer, reports suggest that restoring expression of SMARCB1 genes in SMARCB1-deficient cells slows cell proliferation (Nat Genet. 2017 Nov;49(11):1613-1623. doi: 10.1038/ng.3958).

We have conducted new experiments to examine the effect of SMARCB1 on cell proliferation. These experiments, and the results, are explained in the revised manuscript as follows:

"It is reported that restoring SMARCB1 genes in SMARCB1-deficient cells slows cell proliferation²⁷. Indeed, growth of JMU-RTK-2 +SMARCB1 cell lines was slower than that of JMU-RTK-2 -SMARCB1 cell lines (**Supplementary Fig. 1k**). Therefore, to consider the impact of differences in proliferation of these cell lines on the results of siRNA screening, we also examined SMARCB1-proficient HEK293T immortalized cell lines and five SMARCB1-proficient cancer cell lines. We confirmed that the siRNA targeting CREBBP and EP300 that we used for screening had almost no effect on cell proliferation (**Fig. 1b, e**). In addition, we examined synthetic lethality in six SMARCB1-proficient cell lines and six SMARCB1-deficient cell lines and found little difference in the degree of cell proliferation overall (**Supplementary Fig. 1l**). This suggests that not only isogenic cells harboring SMARCB1 (i.e., JMU-RTK-2 +SMARCB1 cells), but also the six SMARCB1-proficient cell lines, did not show an effect on CBP/p300 inhibition due to the differences of cell proliferation." (page 6, lines 181-194).

2-In Figure 1, a Western blot depicting the decrease in CREBBP and EP300 after depletion by siRNA should be included.

Response

Thank you, we have now included a western blot conducted after knockdown of

CBP and p300, which confirmed a decrease in protein expression (**Supplementary Fig. 1d, g, j**).

3-It is challenging to comprehend the specific information presented in Figure 2A and Supplementary Fig 2A. The figure legends do not provide sufficient clarification either. On the other hand, the experimental design, and the approach for selecting KREMEN2 might not be the most optimal strategy. From what this reviewer understands, the researchers have chosen genes that exhibit common up/down-regulation in SMARCB1-deficient cells compared to SMARCB1wt cells. However, their initial comparison involves selecting genes that are commonly altered in cells after SMARCB1-restitution and in one SMARCB1-deficient cell line compared to one SMARCB1-proficient cell line from different tumor types. A more appropriate approach could have been restoring SMARCB1 in two or three different deficient cancer cell lines, identifying common genes that are affected and, subsequently, validating these candidate genes in a panel of SMARCB1-deficient and SMARCB1wt cells.

Response

As the reviewer points out, it would have been better to use multiple SMARCB1 isogenic cell lines to identify common genes. When we first tried to perform such an analysis, we tried to establish SMARCB1 isogenic cell JMU-RTK-2 and other SMARCB1-deficient cell lines; however, we were unable to achieve this. In addition, most rhabdoid tumors and epithelioid sarcomas are SMARCB1-deficient, and there are no commercially available SMARCB1-proficient rhabdoid tumor- or epithelioid sarcoma-derived cell lines.

Therefore, to compare multiple pairs of SMARCB1-proficient and SMARCB1-deficient cell lines, we searched for common influential genes by examining pairs of SMARCB1-deficient HS-ES-2R and SMARCB1-proficient 786-O cell lines in addition to the SMARCB1 isogenic cell line JMU-RTK-2. Although this analytical approach might not be ideal, the effects of KREMEN2 expression identified using these cell lines could then be validated using SMARCB1-deficient and SMARCB1-proficient cell line panels.

We have tried to improve our explanation of the analysis method, as well as the legends to Figure 2a and Supplementary Figure 2a, to make them easier to

understand.

The manuscript text has been revised as follows:

“To investigate this hypothesis, we performed gene expression analyses using RNA-seq. First, we identified a set of 471 genes that were concordantly upregulated in SMARCB1-deficient cells (JMU-RTK-2, HS-ES-2R) but not in SMARCB1-proficient (JMU-RTK-2 +SMARCB1, 786-O) cells (**Fig. 2a**). Next, we identified a set of 50 genes that were concordantly downregulated in SMARCB1-deficient cells (JMU-RTK-2, HS-ES-2R) treated with A-485, but not in SMARCB1-proficient cells (JMU-RTK-2 +SMARCB1, 786-O) treated with A-485 (**Fig. 2a**). Then, we identified a set of 22 genes that showed overlap between these two gene sets (**Fig. 2a**). To further narrow down these genes, we identified a set of 54 genes that were concordantly downregulated in two other SMARCB1-deficient cell lines (G402, NEPS) treated with A-485 (**Fig. 2a**). Finally, we identified only the KREMEN2 (Kringle containing transmembrane protein 2) gene as overlapping between the 22 gene and the 54 gene sets (**Fig. 2a**). The KREMEN2 gene, which is a single-pass transmembrane protein that plays dual roles in cells (suppression of the Wnt/ β catenin pathway and the apoptosis pathway)^{33,34}, was upregulated specifically in SMARCB1-deficient cells and downregulated specifically in SMARCB1-deficient cells treated with CBP/p300 inhibitors (**Fig. 2a**), indicating that KREMEN2 is a candidate gene that determines synthetic lethality. We were unable to identify a gene downregulated specifically in SMARCB1-deficient cells, or downregulated specifically in SMARCB1-deficient cells treated with CBP/p300 inhibitors (**Supplementary Fig. 2a**).” (page 8, lines 240-260).

Please see also the revised text for the legends to Figure 2a and Supplementary Figure 2a.

4-The authors have performed different wide genome screenings such as bulk-RNAseq, ChIP seq, ATAC-seq. However, it is not clear which exact cell lines/cell models/conditions have been used in each case. This information could be included in the methods section. A more detailed analysis (genes Up/DOWN, for RNAseq, correlation RNAseq-ChIP seq, ATACseq, among others) is also lacking.

Response

We have now revised the Methods section to better explain/describe all of the cell lines/cell models/conditions used for RNAseq, ChIPseq, CUT&RUNseq/qPCR, and ATAC-seq.

5-What is the point of using siKREMEN2 when there is almost no mRNA in the SMARCB1-deficient cells (Suppl Fig 2C)?

Response

Expression of KREMEN2 mRNA is lower in SMARCB1-proficient cells than in SMARCB1-deficient cells. Knockdown experiments confirmed the efficacy of siKREMEN2 (**Fig. 2c**); however, to show that SMARCB1-proficient cells were transfected successfully with KREMEN2 siRNA (as were SMARCB1-deficient cells) even though expression of KREMEN2 is low in SMARCB1-proficient cells, we have conducted an experiment that demonstrates the knockdown efficiency of siKREMEN2 (see **Supplementary Fig. 2c**).

6-There was no detectable KREMEN2 protein observed in any of the western blots conducted to measure endogenous protein levels. The authors must provide an explanation for how a protein that is either not expressed or shows very low levels in all the cell lines can play a role in mediating cell death in SMARCB1-mutant tumors treated with EP300/CREBBP inhibitors. This is a crucial concern as it forms the foundation of their work and requires thorough explanation and addressing.

Response

Thank you. We conducted the experiments to address this issue, and explain the results in the manuscript as follows:

“We were unable to detect intracellular KREMEN2 protein using a commercially available anti-KREMEN2 antibody (**Supplementary Fig. 2j, k, l**). Transfection of KREMEN2 siRNA into SMARCB1-deficient cells reduced expression of KREMEN2 mRNA, as detected by qPCR (**Supplementary Fig. 2j**). The commercially available anti-KREMEN2 antibody did bind to several protein bands on western blots, but the intensity of these bands was not reduced after transfection of KREMEN2 siRNA (**Supplementary Fig. 2k**). However, the KREMEN2 antibody did detect ectopically overexpressed KREMEN2 because the intensity of a band representing overexpressed KREMEN2 was reduced after transfection of KREMEN2 siRNA

(Supplementary Fig. 2I). Although expression of KREMEN2 mRNA in SMARCB1-deficient cells was higher than that in SMARCB1-proficient cells, expression of endogenous KREMEN2 proteins was not detected by commercially available antibodies. However, expression of overexpressed KREMEN2 proteins was detected by introducing cDNA encoding exogenous KREMEN2. Since this overexpressed KREMEN2 protein was knocked down by siKREMEN2, we believe that the antibody does recognize KREMEN2. Therefore, although the currently available antibodies were not able to detect expression of endogenous (intracellular) KREMEN2 protein, the lethality induced by KREMEN2 knockdown can be avoided by overexpression of KREMEN2, as shown in **Figure 2d**. Thus, we consider that endogenous KREMEN2 proteins (although they may be expressed at low levels) are functional. Therefore, CBP/p300 inhibitors may attenuate expression of KREMEN2 protein and induce cell death." (pages 9-10, lines 294-315).

7-The authors should utilize databases and previous datasets of RNAseq and ChIP-seq to validate their observations. For example, they could examine a large panel of cancer cell lines that are SMARCB1-mutant and compare their KREMEN2 levels to those of SMARCB1-wt expressing cell lines. Additionally, they should investigate whether KREMEN2 levels decrease in other experimental datasets where SMARCB1 expression has been restored. To gain further insights, the authors should explore whether SMARCB1 directly binds to the KREMEN2 promoter in other available ChIP-seq data. Moreover, it would be valuable to analyze if there is any correlation between increased levels of KREMEN2 and mutations in other members of the SWI/SNF complex, such as SMARCA4.

Response

Thank you for the suggestion. We have done this as requested and revised the manuscript text accordingly, as follows:

"To further examine expression levels of the KREMEN2 gene among cell lines with different genetic abnormalities, we used mutation, copy number, and gene expression data from the CCLE database in DepMap (data version 23Q2). Expression of KREMEN2 mRNA was compared with gene expression data from SMARCB1/SMARCA4-proficient, SMARCB1-deficient, and SMARCA4-deficient cell lines. The results showed that expression of KREMEN2 in not only SMARCB1-

deficient cell lines, but also in SMARCA4-deficient cell lines, was significantly higher than that in SMARCB1/SMARCA4-proficient cell lines (**Supplementary Fig. 2f**).” (page 9, lines 272-281).

“This result was also confirmed by published ATAC-seq (GSE124903) and RNA-seq (GSE124903) data derived from another SMARCB1 isogenic cell line model (SMARCB1-deficient TTC1240 cells), which showed that the ATAC-seq and RNA-seq signals in TTC1240 -SMARCB1 cells were higher than those in TTC1240 +SMARCB1 cells (**Supplementary Fig. 2m**)” (page 10-11, lines 337-341).

“Similar results were obtained by analyzing published ChIP-seq data (GSE90634, GSE124903) derived from another SMARCB1 isogenic (TTC1240) cell line model (**Supplementary Fig. 3a**)^{27,35}.” (page 12, lines 386-388).

8-The ChIP-seq snapshots shown appear to be quite noisy. It is common for ChIP-seq data to exhibit some unspecific background signals. To address this issue, it is advisable to include positive controls, such as snapshots of regions that are well-known to recruit SMARCB1, P300, or are associated with the histone modifications being tested in this study. Incorporating snapshots of these positive controls will help to validate the specificity of the observed signals and ensure the reliability of the results.

Response

Thank you, we agree that there is quite a lot of background noise.

To address the point made by the Reviewer, we have included snapshots of ChIP-seq and CUT&RUN-seq signals at the ANKRD1 and CDKN1A regions; these act as positive regions in which SMARCB1 binding to loci outside of the KREMEN2 locus has been reported.

Cell. 2019 179(6):1342-1356.e23. <https://doi.org/10.1016/j.cell.2019.10.044>

Nat Genet. 2017 49(11):1613-1623. <https://doi.org/10.1038/ng.3958>

We have explained this in the revised manuscript as follows:

“CUT&RUN-seq and ChIP-seq tend to generate non-specific background signals. Therefore, we added snapshots of ChIP-seq and CUT&RUN-seq signals at the ANKRD1 and CDKN1A regions as “positive regions” in which SMARCB1 binding

to loci outside the KREMEN2 locus has been reported ^{27,35}. The localization signals of H3K4me3, H3K4me1, H3K27ac, p300 ChIP-seq in addition to ATAC-seq and RNA-seq were detected in the ANKRD1 gene region in JMU-RTK-2 +SMARCB1 cells, whereas that of H3K27me3 was detected in JMU-RTK-2 -SMARCB1 cells (**Supplementary Fig. 2o**). The localization signals of H3K4me3, H3K4me1, H3K27ac, p300 ChIP-seq, ATAC-seq, and RNA-seq were detected in the CDKN1A region in both JMU-RTK-2 +SMARCB1 cells and JMU-RTK-2 -SMARCB1 cells, but that of H3K27me3 ChIP-seq was not detected in either JMU-RTK-2 +SMARCB1 or JMU-RTK-2 -SMARCB1 (**Supplementary Fig. 2p**). These snapshots support the data showing that the histone markers and p300 localize at the KREMEN2 locus." (pages 11, lines 363-375).

"It is common for epigenetic data obtained by NGS technologies such as ChIP-seq and CUT&RUN-seq data to exhibit some non-specific background signals. To address this, we incorporated positive control snapshots of the ANKRD1 and CDKN1A loci, both of which recruit the SMARCB1-containing SWI/SNF complex ^{27,35}. At both the ANKRD1 and CDKN1A gene loci, SMARCB1 co-localized with H3K27ac and H3K4me3 at regions upstream of the TSS in JMU-RTK-2 +SMARCB1 cells, but not JMU-RTK-2 -SMARCB1 cells (**Supplementary Fig. 3c, d**). Similar results were obtained for another SMARCB1 isogenic (TTC1240) cell line model (**Supplementary Fig. 3e, f**) ^{27,35,36}, and for a pair of SMARCB1-proficient 786-O cell lines and SMARCB1-deficient NEPS cells (**Supplementary Fig. 3g, h**). These snapshots support the data showing that the SMARCB1 localizes at the KREMEN2 locus." (page 12, lines 391-402).

9-Why did the authors perform ChIP-seq of BRD4 and CTCF? The reason behind this needs to be explained.

Response

Localization of BRD4, a member of the bromo and extra-terminal (BET) family, to KREMEN2 was investigated because BRD4 is a transcription factor that binds to acetylated histones, and may therefore be involved in transcriptional regulation at the localized region of H3K27ac, which is targeted by CBP/p300 (Mol Cell. 2021 May 20;81(10):2166-2182.e6. doi: 10.1016/j.molcel.2021.03.008).

CTCF is a transcription factor localized upstream of its target gene loci (Cell. 2017 169(5):930-944.e22. doi: 10.1016/j.cell.2017.05.004). In addition, CTCF colocalizes with the residual SWI/SNF complex in the absence of SMARCB1 (Nat Cell Biol. 2018 Dec;20(12):1410-1420. doi: 10.1038/s41556-018-0221-1), suggesting that it may be involved in transcription of KREMEN2, expression of which is promoted by SMARCB1 deficiency.

Therefore, we examined whether BRD4 and CTCF localize to the KREMEN2 locus in SMARCB1-deficient cell lines. We found that BRD4 and CTCF did localize to KREMEN2 regions (**Fig. 4g**), and that localization of BRD4 and CTCF was attenuated by inhibition of CBP/p300 (**Fig. 4g**).

These data are included in the revised manuscript (page 15, lines 497-508).

10-Again, WB of KREMEN1, not only RT-QPCR, should be included in the siKREMEN1 experiments (various panels on figure 4).

Response

Unfortunately, we were not able to obtain data on the knockdown efficiency of KREMEN1 by western blotting. We have explained the reasons for this in the revised manuscript as follows:

“Next, we tried to examine expression of KREMEN1 protein. Although we could detect KREMEN1 protein expressed via the overexpression vector, we could not detect endogenous intracellular KREMEN1 protein using a commercially available anti-KREMEN1 antibody (**Supplementary Fig. 5r-t**) (as was the case for KREMEN2) (**Supplementary Fig. 2j-l**). We confirmed knockdown of KREMEN1 mRNA by KREMEN1 siRNA (**Supplementary Fig. 5r**); however, western blot analysis using commercially available KREMEN1 antibodies did not detect KREMEN1 proteins (as was also the case for KREMEN2). Many bands were detected by the anti-KREMEN1 antibodies, but none showed reduced intensity or disappeared after treatment with KREMEN1 siRNA, even after long exposure to chemical luminescence agents (**Supplementary Fig. 5s**). In addition, in cells transduced with an KREMEN1 overexpression vector, we detected bands of KREMEN1 that disappeared upon treatment with KREMEN1 siRNA

(**Supplementary Fig. 5t**). The current commercially available KREMEN1 antibody detected KREMEN1 when expressed at high levels, but not at basal levels. This may have something to do with the specificity/affinity of the KREMEN1 antibody. However, we consider that endogenous KREMEN1 proteins are expressed and are functional in the cells.” (pages 19, lines 634-651).

11-On figure 5A, please include a list with the 8 common genes.

Response

Figure 5A of the original manuscript showed common molecular pathways associated with each of gene sets identified as showing altered expression after treatment with CBP/p300 inhibitors, after CBP/p300 knockdown, or after KREMEN2 knockdown. We identified eight common pathways. However, because Reviewer 1 pointed out that there was little evidence to support the use of a phosphorylated antibody array to identify signaling pathways from the data in Figure 5A, we decided to perform GSEA analysis. Therefore, we have removed the original Figure 5A from the revised manuscript and instead included the results of GSEA analysis (see the new **Figure 6**).

12-Please, expand the number of SMARCB1 wt tested for pAKT inhibition following p300/CBP inhibition. It seems that only 1 has been tested (Fig 5D) and thus, a cell-specific effect cannot be discarded.

Response

In addition to 786-O, we examined three SMARCB1-proficient cell lines. We observed no phosphorylation of AKT in the three additional SMARCB1-proficient cell lines, but we did detect phosphorylation of PRAS40 in both. In addition, CBP/p300 inhibition did not affect phosphorylation of either AKT or PRAS40. These data have been included in the manuscript (page 22, lines 754-757) and are presented in **Supplementary Figure 6f**.

Other comments

13-In Figure 1K and L the cell lines that have been used for these experiments should be indicated.

Response

We apologize for the confusion. In the revised manuscript, the cell lines (SMARCB1-proficient (10 cell lines) and SMARCB1-deficient (8 cell lines)) used in **Fig. 1k, l, and m** are compatible with those in **Supplementary Fig. 1o, p, and q**. The text has been revised to make this clear.

14-For many of the experiments it is not indicated how many replicates have been used, ex: Fig 2B-F; Fig 3A-C, F-K, among others.

Response

The number of replicates per experiment is now specified in the Figure Legends.

15-Please indicate, for each of the figure panels, whether cancer cell lines are SMARCB1 wild type or mutant. Sometimes this has been indicated but not always, making the manuscript difficult to read.

Response

The name of the cell lines, and whether they are SMARCB1 wild-type or mutant lines, has now been stated in the figure panels and not just the figure legends.

16-The manuscript will benefit from some English editing because, again, some sections/paragraphs are difficult to follow. For example, on page 7, lines 31-32 the paragraph: “ (...) in SMARCB1-deficient cells, but not SMARCB1-deficient cells, after treatment with (...)”. Either there is a mistake, or the comma is misplaced.

Response

We have revised the sentence “ (...) in SMARCB1-deficient cells, but not SMARCB1-deficient cells, after treatment with (...)” to “The KREMEN2 gene, which is a single-pass transmembrane protein that plays dual roles in cells (suppression of the Wnt/ β catenin pathway and the apoptosis pathway)^{33,34}, was upregulated specifically in SMARCB1-deficient cells and downregulated specifically in SMARCB1-deficient cells treated with CBP/p300 inhibitors (**Fig. 2a**), indicating that KREMEN2 is a candidate gene that determines synthetic lethality”. We have also made other careful corrections overall.

17-Please correct: SAMRCB1 (page 6 line 21); proficient (page 6, line 27); deificent

(page 6, line 33); Materials (Page 18)

Response

We have revised the text to avoid any misunderstanding. We have also edited the paper to try to make the language as precise as possible.

Reviewers' Comments:

Reviewer #1:

Remarks to the Author:

The authors have done a really thorough revision of the paper and it's much improved. I do not have any further concerns from a reviewer's viewpoint.

Reviewer #2:

Remarks to the Author:

Overall, this version of the manuscript is improved with careful consideration of each reviewer's comments and suggestions. While not every concern could be met, most reviewer concerns were addressed. In particular, data mapping BAF and other chromatin regulator subunits at the KREMIN2 gene strengthen the mechanism for how BAF functions at this locus in SMARCB1 proficient and deficient cells. The synthetic lethality is convincing, and the evidence of it is more robust. The writing has improved, and the paper is now clearer.

The role of KREMINs remains confusing, but I appreciate the authors' efforts to address our collective concerns. It is too bad that antibodies are not great and only detect high protein levels. While the authors could have used CRISPR to knock-in an epitope tag for better KREMIN detection this would have been an ambitious undertaking, but may be required for further KREMIN studies. It is possible that low (below detection) levels of KREMIN are enough to drive the phenotype, but the result is still a bit surprising and mysterious. The AKT data remains perplexing, and I hope the authors will be able to build models in the future to understand it better. Even with these two unresolved issues, I am enthusiastic about this work and think it will be of interest to the BAF and epigenetic cancer fields.

Reviewer #3:

Remarks to the Author:

The authors have made revisions aimed at addressing all questions and doubts. However, two significant concerns remain, particularly regarding the soundness of the mechanistic explanations, which is challenging to be solved, especially the first one.

1.-Despite their comments, it remains still perplexing that the reduction in levels of a protein scarcely expressed in their cancer cell models is attributed to causing the observed effects. Their rationale regarding the lack of sensitivity of the commercial antibody is not entirely convincing, as the KREMEN2 band is prominently observed in cells overexpressing the protein and in the same cells after siRNA treatment, which markedly reduces protein levels. They posit that endogenous KREMEN2 protein, albeit expressed at low levels, is functional, but this assertion appears overstated. Similar considerations apply to KREMEN1.

2.- Regarding the ChIP-seq data of SMARCB1, the authors have now included some positive controls demonstrating the recruitment of SMARCB1 in wild-type cells. However, there is a notable discrepancy in the scale of the snapshots presented: the scale is higher for the positive controls compared to that for KREMEN2. This incongruity raises concerns about the possibility of nonspecific peaks, which cannot be ruled out.

Finally, in their initial screening, they also identify additional potential candidates for inhibition in SMARCB1-deficient cells. A recent publication discusses the inhibition of KDM6s in SMARCA4-deficient tumors, which the authors may find relevant to their work (PMID: 34262032).

Response Letter R2

REVIEWER COMMENTS

Reviewer #1 (Remarks to the Author):

The authors have done a really thorough revision of the paper and it's much improved. I do not have any further concerns from a reviewer's viewpoint.

Response

Thank you for reviewing the revised version of our manuscript, which is now much improved.

Reviewer #2 (Remarks to the Author):

Overall, this version of the manuscript is improved with careful consideration of each reviewer's comments and suggestions. While not every concern could be met, most reviewer concerns were addressed. In particular, data mapping BAF and other chromatin regulator subunits at the KREMIN2 gene strengthen the mechanism for how BAF functions at this locus in SMARCB1 proficient and deficient cells. The synthetic lethality is convincing, and the evidence of it is more robust. The writing has improved, and the paper is now clearer.

Response

Thank you for reviewing the revised manuscript. Thanks to your suggestion regarding the relationship between SWI/SNF complexes at the KREMEN2 locus, we were able to propose a mechanism underlying the functional relationship between SWI/SNF complexes involved in transcriptional regulation of KREMEN2 genes.

The role of KREMINs remains confusing, but I appreciate the authors' efforts to address our collective concerns. It is too bad that antibodies are not great and only detect high protein levels. While the authors could have used CRISPR to knock-in an epitope tag for better KREMIN detection this would have been an ambitious undertaking, but may be required for further KREMIN studies. It is possible that low (below detection) levels of KREMIN are enough to drive the phenotype, but the result is still a bit surprising and mysterious. The AKT data remains perplexing, and I hope the authors will be able to build models in the future to understand it better. Even with these two unresolved issues, I am enthusiastic about this work and think it will be of interest to the BAF and epigenetic cancer fields.

Response

We examined the phenotypes resulting from knockdown and overexpression of KREMEN2 and KREMEN1 genes; however, we could not detect endogenous KREMEN2 protein or endogenous KREMEN1 protein using commercially available antibodies. Therefore, we cannot prove that endogenous KREMEN2 and KREMEN1 proteins are expressed and functional. In future, we would like to investigate synthetic lethal therapies that target KREMEN2 in SMARCB1-deficient cancers by constructing a knock-in model with epitope tags, or by creating anti-

KREMEN2 antibodies with high specificity for endogenous KREMEN2.

In addition, previous reports suggest that KREMEN1 is involved in induction of apoptosis. Here, we suggest that multiple signaling pathways are involved in induction of apoptosis downstream of KREMEN1. It is possible that the AKT pathway is involved, but there is still a possibility that multiple pathways induce apoptosis. We intend to conduct further research that explains this mechanism in more a simple way.

Reviewer #3 (Remarks to the Author):

The authors have made revisions aimed at addressing all questions and doubts. However, two significant concerns remain, particularly regarding the soundness of the mechanistic explanations, which is challenging to be solved, especially the first one.

Response

Thank you for reviewing the revised manuscript. We have tried to address all of the concerns as best we can; however, we are aware that important points have not been resolved.

1.-Despite their comments, it remains still perplexing that the reduction in levels of a protein scarcely expressed in their cancer cell models is attributed to causing the observed effects. Their rationale regarding the lack of sensitivity of the commercial antibody is not entirely convincing, as the KREMEN2 band is prominently observed in cells overexpressing the protein and in the same cells after siRNA treatment, which markedly reduces protein levels. They posit that endogenous KREMEN2 protein, albeit expressed at low levels, is functional, but this assertion appears overstated. Similar considerations apply to KREMEN1.

Response

We could detect overexpressed KREMEN2 proteins using commercially available anti-KREMEN2 antibodies (**Supplementary Fig. 2l**); however, the anti-KREMEN2 antibody bound to several proteins on western blots, and the intensity of these bands was not reduced after transfection of KREMEN2 siRNA into four SMARCB1-deficient cell lines (**Supplementary Fig. 2k**). By contrast, we were able to obtain genetic data suggesting that the lethality induced by knockdown of KREMEN2 genes can be avoided by overexpression of KREMEN2 genes (and thus KREMEN2 proteins). We believe that these data show that the endogenous KREMEN2 proteins are functional; however, as you pointed out, we have not been able to prove that endogenous KREMEN2 is functional. We agree that stating that endogenous KREMEN2 proteins are functional is an exaggeration. Therefore, the text has been revised as follows:

“Therefore, although the currently available antibodies were not able to detect

expression of endogenous KREMEN2 protein, the lethality induced by KREMEN2 knockdown can be avoided by overexpression of KREMEN2, as shown in **Figure 2d**. Thus, CBP/p300 inhibitors may attenuate gene expression of KREMEN2 and induce cell death. However, we have not been able to prove that endogenous KREMEN2 proteins are expressed and functional. In the future, this issue will be resolved by inserting an epitope tag into endogenous KREMEN2 or by creating anti-KREMEN2 antibodies with high specificity and affinity." (**Page 10, lines 312–319**)

Similarly, we were able to detect overexpressed KREMEN1 proteins using commercially available anti-KREMEN1 antibodies (**Supplementary Fig. 5t**); however, many bands were detected by the anti-KREMEN1 antibodies in six cell lines, but none showed reduced intensity or disappeared after treatment with KREMEN1 siRNA, even after long exposure to chemical luminescence agents (**Supplementary Fig. 5s**). By contrast, we were able to obtain genetic data suggesting that the lethality induced by knockdown of KREMEN2 genes can be avoided by knocking down KREMEN1. As with KREMEN2, we believe that these data reflect the function of the endogenous KREMEN1 protein. However, as you point out, since we have not been able to prove that endogenous KREMEN1 proteins are functional, it is an exaggeration to state that endogenous KREMEN1 proteins are functional. Therefore, I have corrected the text as follows:

"The current commercially available KREMEN1 antibody detected KREMEN1 when expressed at high levels, but not at basal levels. This may have something to do with the specificity and affinity of the KREMEN1 antibody; however, we have not been able to prove that endogenous KREMEN1 proteins are expressed and functional. In the future, this issue will be resolved by inserting an epitope tag into endogenous KREMEN1, or by creating anti-KREMEN1 antibodies with high specificity and affinity." (**Page 19, lines 651–658**)

2.- Regarding the ChIP-seq data of SMARCB1, the authors have now included some positive controls demonstrating the recruitment of SMARCB1 in wild-type cells. However, there is a notable discrepancy in the scale of the snapshots presented: the scale is higher for the positive controls compared to that for KREMEN2. This incongruity raises concerns about the possibility of nonspecific peaks, which cannot be ruled out.

Response

As pointed out, in the JMU-RTK-2 isogenic model, the CUT&RUN-seq scale (0–0.19) (**Fig. 3a**) of SMARCB1 at the KREMEN2 locus was lower than that at the other loci (ANKRD1, 0–0.24 (**Supplementary Fig. 3c**), and CDKN1A, 0–0.29 (**Supplementary Fig. 3d**)). However, the CUT&RUN-seq scale (0–0.43) (**Fig. 3a**) of H3K27ac at the KREMEN2 locus was also lower than that at other loci (ANKRD1, 0–1.70 (**Supplementary Fig. 3c**), and CDKN1A, 0–2.47 (**Supplementary Fig. 3d**)).

In addition, in the TTC1240 isogenic model using the published ChIP-seq data, which was analyzed under the same conditions as the JMU-RTK-2 isogenic model, the scale (0–0.16) (**Supplementary Fig. 3a**) of SMARCB1 at the KREMEN2 locus was lower than that at other loci (ANKRD1, 0–0.66 (**Supplementary Fig. 3e**), and CDKN1A, 0–0.29 (**Supplementary Fig. 3f**)). In addition, the scale (0–0.34; **Supplementary Fig. 3a**) of the ChIP-seq data for H3K27ac at the KREMEN2 locus was lower than that at other loci (ANKRD1, 0–0.67 (**Supplementary Fig. 3e**), and CDKN1A, 0–1.17 (**Supplementary Fig. 3f**)).

These comparative data suggest that the scale of the SMARCB1 signal values, as well as those of H3K27ac, varies at each locus. Different cell types may also have different signal values.

Importantly, as noted by the Reviewer, by presenting snapshots of loci other than KREMEN2, we were able to show that the SMARCB1 protein localizes not only at the KREMEN2 locus but also at other loci. In addition, published data suggest a similar trend in another isogenic cell line model derived from a SMARCB1-deficient TTC1240 cell line.

In addition, we investigated CUT&RUN-qPCR to support the localization signal of the SMARCB1 at the KREMEN2 locus (**Fig. 3b**). As a result, we clarified that not only SMARCB1, but also other SWI/SNF complex factors such as SMARCA4, ARID1A, PBRM1, and SS18, are localized upstream of the KREMEN2 locus in a SMARCB1-proficient cell line (**Fig. 3b–f**). Thus, SMARCB1 proteins are localized at the KREMEN2 locus.

Finally, in their initial screening, they also identify additional potential candidates for inhibition in SMARCB1-deficient cells. A recent publication discusses the inhibition of KDM6s in SMARCA4-deficient tumors, which the authors may find relevant to their work (PMID: 34262032).

Response

Thank you for highlighting the paper on KDM6A and KDM6B, which have been identified as existing synthetic lethal targets in SMARCA4-deficient cancers. In addition to existing targets EZH1 and EZH2, we identified KDM6A and KDM6B as potential targets during screening of synthetic lethal targets for SMARCB1-deficient cancers, which supports the reliability of the screening process used in this study. The manuscript text has been revised as follows:

"In addition, it has been reported that KDM6A and KDM6B are synthetic lethal targets for cancers deficient in SMARCA4, which is another subunit of the SWI/SNF complex²⁷." (**Page 5, lines 155–157**).

Reviewers' Comments:

Reviewer #2:

Remarks to the Author:

I have no further comments regarding this study, as both this referee and the authors agree that while further studies will be required to elucidate the mechanism in more detail, that should not prevent the publication of the current study.

Reviewer #3:

Remarks to the Author:

The authors have now addressed all the concerns prompted by this reviewer.

Response Letter R3

REVIEWERS' COMMENTS

Reviewer #2 (Remarks to the Author):

I have no further comments regarding this study, as both this referee and the authors agree that while further studies will be required to elucidate the mechanism in more detail, that should not prevent the publication of the current study.

Response

Thank you for reviewing the revised version of our manuscript. In future, we would like to elucidate the mechanism of synthetic lethality between CBP/p300-KREMEN2 axis and SMARCB1 in more detail.

Reviewer #3 (Remarks to the Author):

The authors have now addressed all the concerns prompted by this reviewer.

Response

Thank you for reviewing the revised version of our manuscript, which is now much improved.